# A CD26$^+$ tendon stem progenitor cell population contributes to tendon repair and heterotopic ossification

Siwen Chen[1,2,8], Yingxin Lin [3,4,5,8], Hao Yang[6,8], Zihao Li[1,2], Sifang Li[1,2], Dongying Chen[7], Wenjun Hao[1,2], Shuai Zhang[1,2], Hua Chao[1,2], Jingyu Zhang[1,2], Jianru Wang [1,2], Zemin Li[1,2], Xiang Li[1,2], Zhongping Zhan[7] & Hui Liu [1,2] ✉

Inadequate tendon healing and heterotopic bone formation result in substantial pain and disability, yet the specific cells responsible for tendon healing remain uncertain. Here we identify a CD26$^+$ tendon stem/progenitor cells residing in peritendon, which constitutes a primitive stem cell population with self-renewal and multipotent differentiation potentials. CD26$^+$ tendon stem/progenitor cells migrate into the tendon midsubstance and differentiation into tenocytes during tendon healing, while ablation of these cells led to insufficient tendon healing. Additionally, CD26$^+$ tendon stem/progenitor cells contribute to heterotopic ossification and Tenascin-C-Hippo signaling is involved in this process. Targeting Tenascin-C significantly suppresses chondrogenesis of CD26$^+$ tendon stem/progenitor cells and subsequent heterotopic ossification. Our findings provide insights into the identification of tendon stem/progenitor cells and illustrate the essential role of CD26$^+$ tendon stem/progenitor cells in tendon healing and heterotopic bone formation.

Tendon is a unique form of connective tissue that links muscle to bone, possessing specific mechanical properties that enable it to adapt to loading transmitted by muscles[1,2]. Injury of tendon can be highly debilitating, resulting in substantial pain, disability, and health-care costs[2,3]. Healed tendons rarely attain the mechanical properties of the undamaged state and associated complications usually worsen clinical outcomes[1].

Identifying the cells responsible for tendon healing is crucial for elucidating the underlying mechanism and development of therapeutic strategies to improve tendon repair[1,3]. However, the specific cells responsible remain elusive and debated. Tendon stem/progenitor cells (TSPCs) served as promising target for therapeutic strategies to improve tendon repair[1,4–6]. TSPCs were identified by Bi et al. through cell isolation from murine and human tendons[7]. These cells exhibited

the common properties of stem cells in vitro[7]. Subsequent studies also confirmed the existence of TSPCs with triple-differential potentials through tendon isolation and cell culture[4,5,8]. However, since the acquisition of TSPCs in these investigations relied solely on enzymatic isolation of tendons, these TSPCs constituted a heterogeneous pool of diverse cell types. Further studies attempted to characterize TSPCs with markers of tenocytes, mesenchymal stromal cells or other stem/progenitor cells, including Scleraxis (Scx), CD90, Sca-1, CD146 and Nestin[4,7,9]. Although the tenogenesis capacity of these cells have been confirmed in vitro, these markers are also expressed in a variety of other type of cells, which raises the questions about the rationale of the identity of TSPCs with these markers. In addition, whether these cells contribute to tenogenesis in endogenous tendon injury repair remains uncertain. Among these population of TSPCs, the most

[1]Department of Spine Surgery, The First Affiliated Hospital, Sun Yat-sen University, Guangdong, PR China. [2]Guangdong Province Key Laboratory of Orthopaedics and Traumatology, Guangdong, PR China. [3]School of Mathematics and Statistics, The University of Sydney, Sydney, NSW, Australia. [4]Charles Perkins Centre, The University of Sydney, Sydney, NSW, Australia. [5]Laboratory of Data Discovery for Health Limited (D24H), Science Park, Hong Kong SAR, PR China. [6]Pediatric Orthopaedics, Beijing Jishuitan Hospital, Peking University, Beijing, PR China. [7]Department of Rheumatology and Immunology, The First Affiliated Hospital, Sun Yat-sen University, Guangdong, PR China. [8]These authors contributed equally: Siwen Chen, Yingxin Lin, Hao Yang. ✉e-mail: liuhui58@mail.sysu.edu.cn

convincing in vivo evidences in tendon repair comes from the studies of Scx[+] cells, osteocalcin (Ocn)[+] cells, Nestin[+] cells, αSMA[+] cells and Tppp3[+] cells with rigorous experimental design[9–13]. Nonetheless, these studies showed controversial results. Howell et al. found Scx[+] cells were able to regenerate transected Achilles tendon in neonates but not in adults[11]. Wang et al. identified that peritendon derived Ocn[+] cells contributed to tendon repair in vivo, whereas Pdgfrα[+]Tppp3[+] tendon stem cells (TSCs) identified by Harvey et al. were Ocn[-10,12]. Dyment et al. identified αSMA[+] cells contribute to tendon healing but did not give rise to tenocytes with longitudinally aligned collagen matrix second harmonic generation (SHG) signals[13]. Collectively, the identity of TSPCs remains incompletely understood.

Heterotopic ossification (HO) is a pathological process that occurs after injury, in which bone forms in nonskeletal tissues such as muscles and tendons, causing pain and restricting range of motion[14,15]. Rather than scarring, adhesions and failed repairs, HO is one of the most severe complications after injury of tendon, leading to severe pain, deformities and joint contractures[15,16]. HO of tendon is common in populations with high risk of tendon injury such as athletes and workers with repetitive tendon overuse or sever tendon injury[3,15,17–19]. Additionally, HO formation has been reported in 14-62% of cases following surgical repair after tendon injury[20,21]. Identification of the specific cells responsible for HO during tendon healing is crucial for elucidating the cellular and molecular mechanisms underlying HO, which could provide insights for therapeutic strategies. Extensive studies exist on heterotopic ossification in tendon injury, but the cells type responsible for HO remains a topic of debate. Previous studies suggest that Prrx1[+] mesenchymal progenitor cells and Scx[+] tendon progenitor cells contribute to genetic HO, including fibrodysplasia ossificans progressive (FOP) and progressive osseous heteroplasia (FOH), but whether they contribute to tramatic HO is under debated[22,23]. Scx[+] tendon progenitor cells were found contribute less to trauma induced HO than Scx[-] progenitor cells, whose identification remains to be clarified[11]. Therefore, further investigation into the identification of the specific cells responsible for HO during tendon healing is essential for understanding the underlying mechanisms and developing effective therapeutic strategies.

In this work, we utilize single-cell RNA sequencing (scRNA-seq) to identify a population of CD26 (Dipeptidyl peptidase−4, Dpp4)[+] tendon stem/progenitor cells (CD26[+] TSPCs) resident within tendon tissue. These TSPCs exhibit high proliferative potential, display multipotent differentiation capacities, and are capable of differentiating into tenocytes. Through both in vitro studies and in vivo lineage tracing, we confirm that CD26[+] TSPCs contribute to both tendon healing and HO. Furthermore, our mechanistic investigation uncovers a critical role for Tenascin-C (TNC)-Hippo signaling in the chondrogenic differentiation of CD26+ TSPCs and the formation of HO. Collectively, our findings offer insights into the identification of tendon stem/progenitor cells and their underlying cellular and molecular mechanisms, emphasizing their pivotal roles in tendon healing and heterotopic bone formation.

## Results

### Single-cell RNA-seq identifies CD26[+] mesenchymal stromal cells

To investigated the cell populations in Achilles tendon at adulthood, single-cell RNA-sequencing (scRNA-seq) was applied to identify the cell populations in an unbiased manner of the Achilles tendon from 9-week-old shamed mice (n = 25). Unsupervised clustering was performed and gene expression data were aligned to projected in a 2-dimensional space through uniform manifold approximation and projection (UMAP) and 9 clusters were identified (Fig. 1a). Clusters include mesenchymal stromal cells (MSCs), tenocyte, immune cell, endothelial cell, epithelial cell, lymphatic cell, smooth muscle cell, and muscle stem cell (Fig. 1a, b).

To investigate the populations within MSCs and its relationship with tenocytes, MSCs and tenocyte cluster were extracted and

projected through UMAP. Three transcriptionally distinct mesenchymal stromal cell (MSC) subpopulations (clusters 1–3, namely MSC-1, MSC-2 and MSC-3) were identified based on canonical MSC markers *Prrx1, Pdgfrα* and *Ly6a (Sca1)* (Fig. 1a, b)[24–27]. MSC-1 cells represent "Ly6a[+]Cd26[+]Sema5a[-] MSCs" (CD26[+] MSCs) with their high expression of *Cd26, Pi16, Mfap5, Stmn4* and *Il33* (Fig. 1c-d, S1a-e). Cd26/Dpp4 is expressed as a type II transmembrane protein, with a short six amino acid cytoplasmic tail[28]. It is active as a dimer with a mono-mer molecular weight of 110 kDa. which is involved in processes such as nutrition, nociception, cell-adhesion, psycho-neuroendocrine regulation, immune response and cardiovascular adaptation[29]. In contrast, MSC-2 cells were classified as "Ly6a[+]Cd26[-]Apoe[+] MSCs" (Apoe[+] progenitors), since they had few expressions of both *Cd26* and *Sema5a*, but expressing higher level of *C4b, Apoe, Gpx3, Cxcl14* and *Lpl* (Fig. 1c, d, S1a-b). MSC-3 cells, were classified as "Ly6a[+]CD26[-]Sema5a[+] progenitors" (Sema5a[+] progenitors), since they had few expressions of *Cd26* and uniquely expressed *Sema5a* along with *Ingba* and *Hmcn1*. Lastly, a related cluster 4 cells were marked by typical tenocytes markers, *Scx, Fmod* and *Tnmd*, likely representing tenocytes (Fig. S1a-b). To investigate the uniqueness of CD26[+] MSCs relative to previously identified tendon stem/progenitor cells (TSPCs), we detected the signature genes expression level of Tppp3[+]Pdgfra[+] TSCs[10], classic TSPCs[7], Nestin[+] TSPCs[9], OCN[+] TSPCs[12] and αSMA[+] TSPCs[13] in CD26[+] MSCs, Apoe[+] progenitors, Sema5a[+] progenitors and tenocytes from the scRNA-seq dataset. The results showed that CD26[+] MSCs were partially enriched in signature genes of Tppp3[+]Pdgfra[+] TSPC (*Tppp3, Pdgfra, Ly6e, Plin2, Lama4*) and classic TSPC (*Ly6a, Cd44* and *Cd90* instead of *Scx* and *Comp*). Nonetheless, about 25.6% of CD26[+] MSCs did not expressed *Tppp3*, indicating that CD26[+] MSCs were not identical to Tppp3[+]Pdgfra[+] TSPC. In addition, marker genes of Nestin[+] TDPC (*Nestin* and *Mcam*), OCN[+] TDPC (*Bglap*) and αSMA[+] TPC (*Acta2*) were not highly expressed in CD26[+] MSCs, suggesting that CD26[+] MSCs were different form Nestin[+] TDPCs, OCN[+] TDPCs and αSMA[+] TSPCs (Fig. S2a).

Next, we inferred the potential precursor-product relationships between the identified mesenchymal populations using pseudotime analysis with CytoTRACE and Monocle. CytoTRACE analysis on CD26[+] MSCs, Apoe[+] proenitors, Sema5a[+] progenitors and Tenocytes showed that CD26[+] MSCs presented the highest cytotrace score, indicating that CD26[+] MSCs were least differentiated among these populations (Fig. 1e). Pseudotime analysis with Monocle revealed that CD26[+] MSCs might develop towards Apoe[+] progenitors/Sema5a[+] progenitors (Fig. S1d–f). Tracking differential gene expression across CD26[+] MSCs and Sema5a[+]/Apoe[+] progenitors showed CD26[+] MSCs were presented at the beginning of trajectory and enriched expression of previously identified TSPCs genes (*Tppp3, Cd34* and *CD90*) (Fig. 1f)[7,10]. Altogether, we identified a CD26[+] MSCs population which display TSPCs signature. Therefore, we named this cell population as CD26[+] TSPC.

### CD26[+] TSPCs reside within peritendon

To investigate the anatomic localization of CD26[+] TSPCs, gene expression analysis of these cells was performed with the scRNA-seq dataset. The result showed that peritendon markers including *Lama4, Ly6a, Pdgfrα* and *Plin2* were enrich in CD26[+] TSPCs, compared to tenocytes (Fig. 2a). To confirm the anatomic localization of these cells, Achilles tendons were collected from adult mice for histological analysis (Fig. 2b). Immunofluorescent analysis confirmed that CD26[+] TSPCs expressed peritendon marker *Lama4* and *Ly6a* and resided within peritendon (Fig. 2c–e). Similarly, we found CD26[+] TSPCs resided within tail tendon sheath (Fig. S2b). We also observed Apoe[+] progenitors and Sema5a[+] progenitors located within peritendon (Figs. 2f, g). To investigate whether CD26[+] TSPCs were conservative in human, scRNA-seq was performed on the entheseal ligament tissues from patients with scoliosis correction (Fig. 2h). We identified similar cell types as found in the mouse data, including

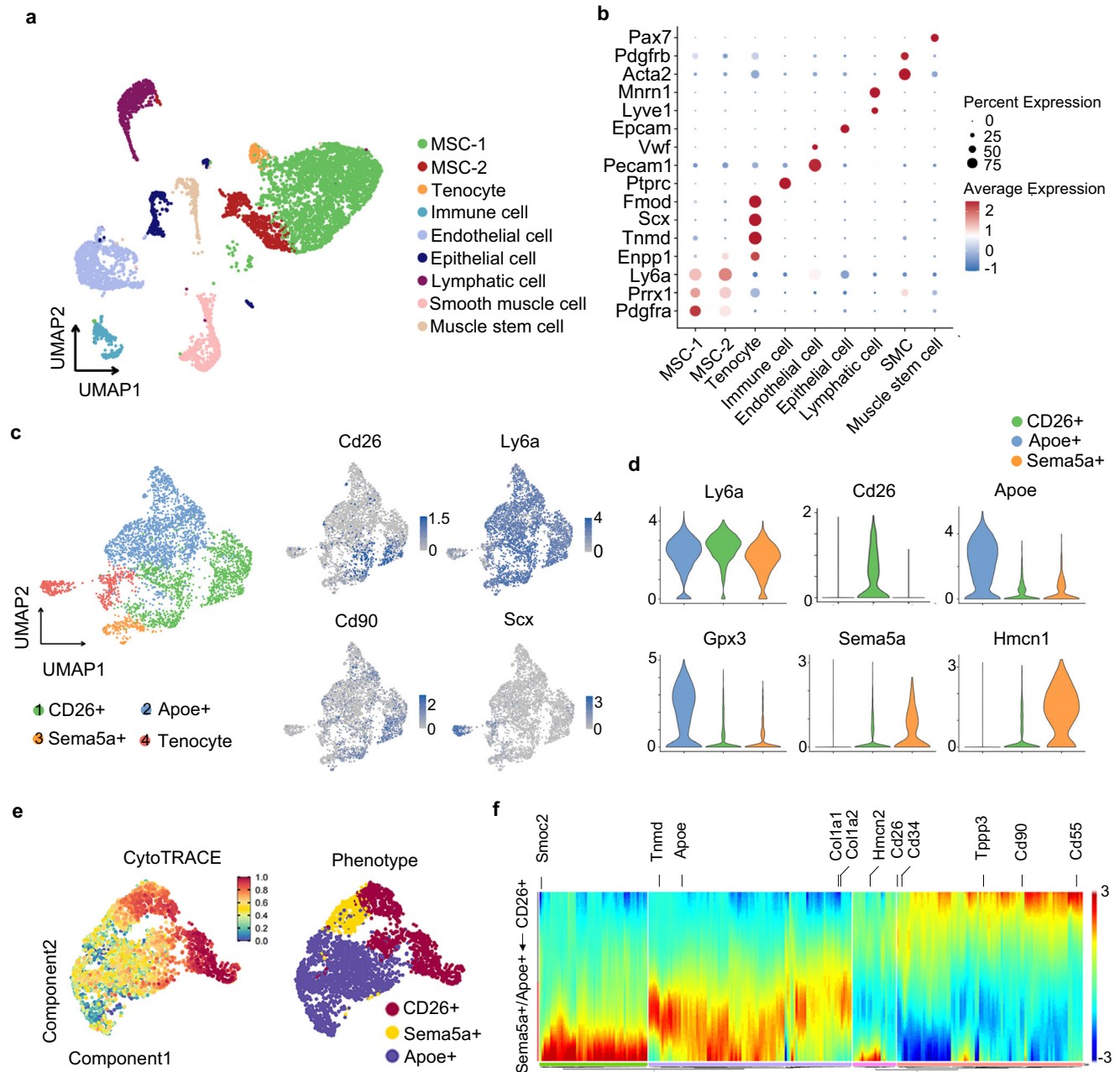

**Fig. 1 | Single-cell RNA-seq and cell trajectory analysis delineate the lineage hierarchy of CD26+ mesenchymal stromal cells. a** UMAP plot and keys to the clustering and cell types are shown to the right. MSC, mesenchymal stromal cell. **b** Bubble diagram of key markers of each cluster. **c** UMAP plot of re-clustering of MSC and tenocyte in 1a and representative markers (*Cd26, Ly6a, Cd90*, and *Scx*). **d** Violin map of key genes of CD26+ MSCs, Apoe+ progenitors, Sema5a+ progenitors.

**e** CytoTRACE analysis of CD26+ MSCs, Apoe+ progenitors, Sema5a+ progenitors. **f** Heatmap of differentially expressed genes ordered based on their common kinetics through pseudotime displayed at each trajectory branch point as defined in the Monocle trajectory analysis of the CD26+ MSCs, Apoe+ progenitors and Sema5a+ progenitors.

mesenchymal cells, chondrocytes, endotheliocytes, smooth muscle cells, muscle cell and immune cells (Fig. S2c). Mesenchymal cell clusters were extracted and project to UMAP and 3 MSC clusters, osteoblast and chondrocyte/tenocyte cluster were identified (Fig. 2i). Gene expression analysis showed that MSC-1 expressed high level of *CD26/Dpp4, CD55, Pi16, CD248* and *Sema3c*, which was also highly expressed in CD26+ TSPC in mice scRNAseq data (Fig. 2j, k). These results indicate that CD26+ TSPC cluster is analogus between mouse and human species. Nonetheless, MSC-2 and MSC-3 showed few expressions of markers of Apoe+ progenitors and Sema5a+ progenitors. Immunofluorescent analysis of Achilles tendons and spinal ligament tissues confirmed that CD26+ TSPCs located within

peritendon of Achilles tendon and spinal ligament enthesis tissue (Fig. 2l and S2d). These results suggest that CD26+ TSPCs reside within peritendon tissue.

## CD26+ TSPCs display self-renewal and triple-lineage differentiation capacities

To investigate the stemness of CD26+ TSPCs, we examined the self-renewal and triple-lineage differentiation capacity of CD26+ TSPCs and Sema5a+ progenitors collected from adult mice aged 9 weeks through fluorescence activated cell sorting (FACS) (Fig. 3a). Proliferation and CCK8 assay showed that CD26+ TSPCs proliferated at a higher rate than Sema5a+ progenitors (Fig. 3b, c). CD26+ TSPCs could differentiated into

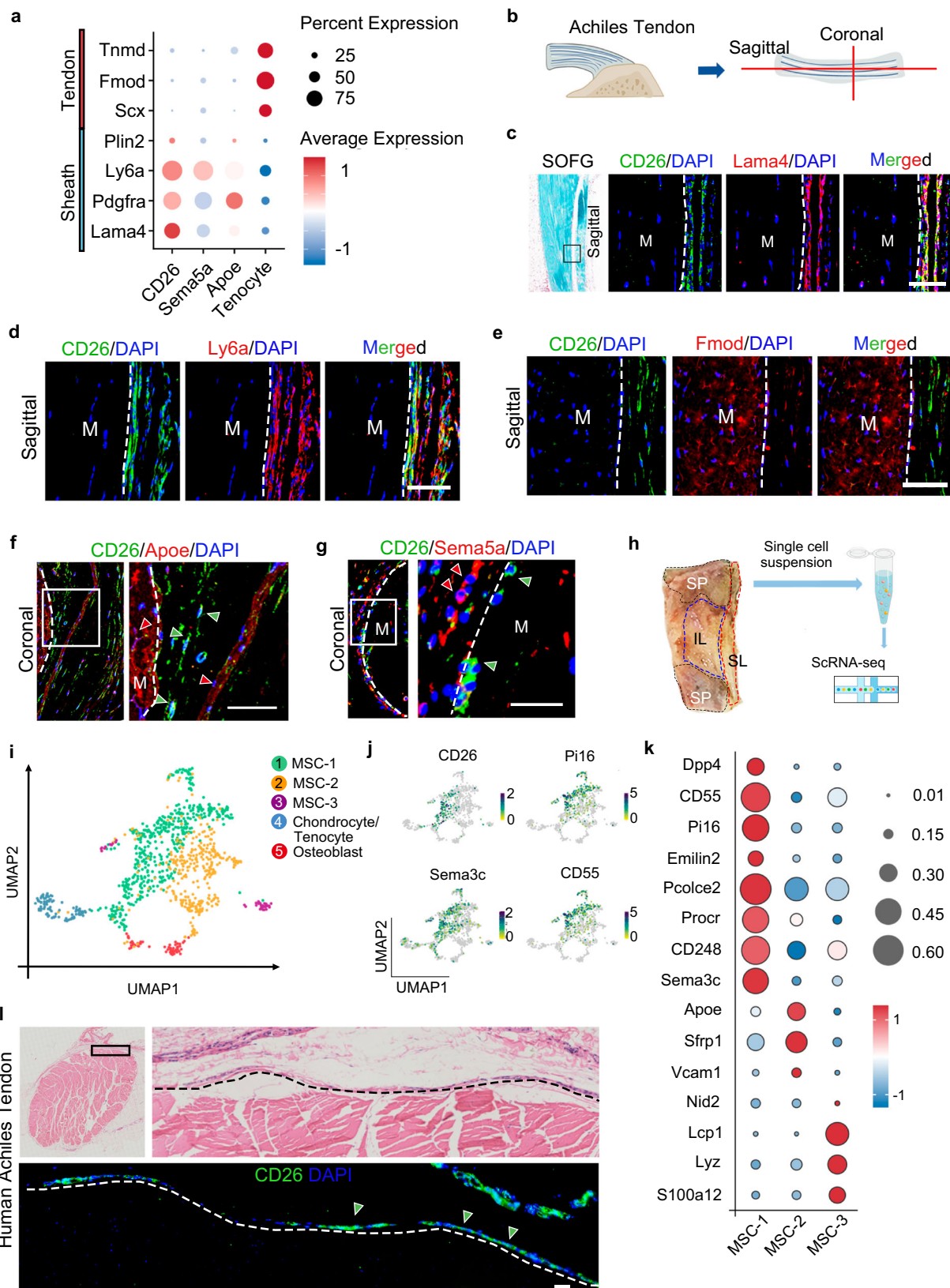

osteogenic, chondrogenic and adipogenic cells, whereas Sema5a⁺ progenitors gave rise to the former two fates (Fig. 3d). Quantifications of the differentiation assays showed that the osteogenic potential of CD26⁺ TSPCs are lower than Sema5a⁺ progenitors (Fig. 3e). Similarly, we found that CD26⁺ TSPCs collected from human Achilles tendons through FACS sorting display proliferative capacity and triple-linage

differentiation potentials (Fig. S3a-c). Previous pseudotime analysis suggested that CD26⁺ TSPCs were precursors of Sema5a⁺ progenitors (Fig. 1e). To validate this hypothesis, we cultured these cells in tentative tendon stem/progenitor cell (TSPCs) medium for 14 days. Flow cytometry analyses showed that CD26⁺ TSPCs could acquire Sema5a expression and differentiate into Sema5a⁺ progenitors at day 14

**Fig. 2 | CD26⁺ TSPCs reside within peritendon. a** Bubble diagram of the expression of peritendon and tenocyte markers in CD26⁺ TSPCs, Apoe⁺ progenitors, Sema5a⁺ progenitors and tenocytes. **b** Illustration of collection of Achilles tendon. **c** SOFG staining and immunofluorescent staining of CD26 and Lama4 of sagittal section of Achilles tendons. n = 3. Scale bar: 100 μm. **d** Immunofluorescent staining of CD26 and Ly6a of sagittal section of Achilles tendons. n = 3. Scale bar: 100 μm. **e** Immunofluorescent staining of CD26 and Fmod of sagittal section of Achilles tendons. n = 3. Scale bar: 100 μm. **f** Immunofluorescent staining of CD26 and Apoe of coronal section of Achilles tendons. n = 3. Scale bar: 100 μm.

**g** Immunofluorescent staining of CD26 and Sema5a of coronal section of Achilles tendons. n = 3. Scale bar: 50 μm. **h** Illustration of collection and scRNA-seq of human ligaments. **i** UMAP plot of MSC-1, MSC-2, MSC-3, chondrocyte/tenocyte and osteoblast. **j** UMAP plot of CD26, Pi16, Sema3c and CD55. **k** Bubble diagram of expression of marker genes in each cluster. **l** H&E staining and immunofluorescent staining of CD26 in human Achilles tendon. n = 3 biological replicates. Scale bar: 100 μm. MSC mesenchymal stromal cell, scRNA-seq single cell RNA sequencing, M midsubstance, SP spinal process, IL interspinal ligament, SL supraspinal ligament.

(Fig. 3f). Nonetheless, Sema5a⁺ progenitors hardly acquired CD26 expression and gave rise to CD26⁺ TSPCs (Fig. 3f). In addition, RT-PCR results showed increased Sema5a⁺ progenitors marker genes (*Sema5a, Enpp1, Hmcn1, Smoc2*) and reduced CD26⁺ TSPCs marker genes (*Dpp4, Sbsn, Stmn4*) in CD26⁺ TSPCs group (Fig. S3d). Nonetheless, Sema5a⁺ progenitors' group did not show increased level of CD26⁺ TSPCs marker genes (Fig. S3e). To further validate the in vivo fate of CD26⁺ TSPCs, we generated a tamoxifen-inducible CreER^T2^-hDTR line specific for *CD26* expression (*CD26-CreER^T2^-hDTR*). The *CD26- CreER^T2^* mice were crossed to *H11-CAG-Loxp-ZsGreen-Stop-LoxP-tdTomato* (*H11-mZsGmtdT*) line to allow lineage tracing and functional ablation of specific populations (Fig. 3g and S3f). Tamoxifen induction of adult *CD26- CreER^T2^-hDTR; H11-mZsGmtdT* mice (*CD26-CreER^T2^-hDTR; mGmT*) resulted in recombination and expression of TdTomato reporter (red fluorescent protein) in CD26⁺ population at peritendon (Fig. S3g-h). Lineage tracing showed that the percentage of tdT⁺ Sema5a⁺ cells increased over time, which further confirmed that CD26⁺ TSPCs gave rise to Sema5a⁺ progenitors under normal condition (Fig. 3h, i). These results indicate that CD26⁺ TSPCs are primitive and display self-renewal and triple-lineage differentiation capacity.

To determine the marker gene expression profile of CD26⁺ TSPCs, we isolated murine CD26⁺ TSPC s with FACS and examined their expression of different stem cell markers. Stem cell markers include CD90, CD44 and CD105, which have been found expressed in previous identified TSPCs and tendon derived progenitor cells (TDPCs)[7,30]. Leptin receptor has been reported as bone marrow-derived MSC marker gene[31]. The results showed that over 99% of CD26⁺ TSPCs were positive for CD90, 26.0 ± 1.0% of these cells were positive for CD44 and 32.9 ± 0.7% of these cells were positive for CD105 (Fig. 3j, k). CD26⁺ TSPCs hardly expressed Leptin receptor (LepR) or tenocyte marker Scx (Fig. 3j, k). To investigate the expression of the stem cell marker profiles in human CD26⁺ TSPCs, we isolated human CD26⁺ TSPCs with FACS and examined their expression of different stem cell markers. Over 94% of human CD26⁺ TSPCs were positive for CD44 and CD90, and 21.1 ± 0.8 of these cells were positive for CD105 (Fig. S4a-b). Human CD26⁺ TSPCs hardly expressed LepR either (Fig. S4a-b). Previous studies also found Tppp3 as tendon stem cell marker[10]. Both flow cytometry analysis and immunofluorescent imaging showed that part of CD26⁺ MSCs, Apoe⁺ progenitors and Sema5a⁺ progenitors were positive for Tppp3 (Fig. 3l–n).

## CD26⁺ TSPCs contribute to tendon healing

To investigate the role of CD26⁺ TSPCs during tendon healing, tendon punch injury model was established and single cell suspension collected at 14 days post injury (dpi) to perform scRNA-seq (Fig. 4a). We further merged scRNA-seq dataset from sham and punch group and identified 13 clusters, including immune cells (monocyte_macrophage, lymphocyte and granulocyte) and non-hematopoietic cells (MSC, chondrocyte/tenocyte, endothelial cell, epithelial cell, smooth muscle cell, glial cell, myocyte and lymphatic cell) (Fig. S4c-d). To investigate the populations within MSCs and its relationship with chondrocyte/tenocyte and chondrocyte, these clusters were extracted and projected through UMAP. Similar pattern of clustering was observed, including CD26⁺ TSPCs, Apoe⁺ progenitor, Sema5a⁺ progenitor and tenocyte. In addition, cluster chondrocyte/tenocyte with high

expression of chondrogenic (*Acan, Col2a1* and *Mmp13*) and tenogenic (*Fmod, Scx, Tnmd*) genes was found (Fig. S4e), along with chondrocyte cluster. To investigate the role of CD26⁺ TSPCs, Apoe⁺ progenitor, Sema5a⁺ progenitor during tendon healing, GO term analysis of differentiated genes in each cluster was performed and the results showed enrichment of cell migration and cell proliferation in all these progenitors after tendon injury, and enrichment of cartilage development and chondrocyte differentiation in CD26⁺ TSPCs and Sema5a⁺ progenitors (Fig. S5a–c). To monitor the proliferation kinetics, 5-ethynyl-2′-deoxyuridine (EdU) was administrated in timed windows during tendon healing (Fig. 4d). We found CD26⁺ and CD26⁺EdU⁺ cells increased in tendon midsubstance at 7 dpi when proliferation of progenitor cells was prominent as previously reported[10], indicating that CD26⁺ cells were undergoing cell proliferation after tendon injury (Fig. 4d–f). Collectively, these results indicate that CD26⁺ MSCs could proliferate and migrate into tendon midsubstance after tendon injury.

To understand the hypothetically developmental relationships that might exist with the TSPCs and tenocytes clusters, we performed trajectory analysis on cluster CD26⁺ TSPCs, Apoe⁺ progenitors, Sema5a⁺ progenitors, tenocytes and chondrocytes/osteoblasts using CytoTRACE and Monocle algorithm. CytoTRACE analysis on CD26⁺ TSPCs, Apoe⁺ progenitors, Sema5a⁺ progenitors, tenocytes, chondrocytes/tenocytes and chondrocytes from merged uninjured and injured group showed that CD26⁺ TSPCs presented the highest cytotrace score, indicating that CD26⁺ TSPCs were least differentiated among these populations (Fig. 4g, S5e). Furthermore, we found that CD26⁺ TSPCs might differentiated into tenocytes and chondrocytes after tendon injury through peudotime analysis (Fig. S5f-i). Tracking differentiated gene expression across CD26⁺ TSPCs to tenocytes also indicated that CD26⁺ TSPCs might develop into tenocytes, as marker genes of tenocytes (*Scx, Mkx, Fmod* and *Tnmd*) increased over pesudotime (Fig. 4h). To confirm whether CD26⁺ TSPCs could differentiated into tenocytes during tendon healing, tendon punch model was generated in *CD26-CreER^T2^-hDTR; H11-mZsGmtdT* mice after tamoxifen induction (Fig. 4i). The results showed that tdT+ cells migrated into tendon midsubstance after injury (Fig. 4j–k). Furthermore, we observed increased Scx⁺tdT⁺ and Fmod⁺tdT⁺ tenocytes in midsubstance (Fig. 4j–m). These results reveal that CD26⁺ MSCs are tendon stem/progenitor cells (TSPCs) and develop into tenocytes during tendon healing.

## Depletion of CD26⁺ TSPCs retards tendon healing

To investigate the role of CD26⁺ TSPCs during tendon healing, we established tendon punch injury model in *CD26-CreER^T2^-hDTR* mice, in which administration of diphtheria toxin (DTx) could deplete CD26⁺ cells. Since some of hematopoietic lineage cells express CD26 and inflammatory response participate in tendon healing process[1], we first investigate the existence of CD45⁺CD26⁺ immune cells under normal condition. The result showed that few CD45⁺CD26⁺ immune cells were found in uninjured tendon (Fig. S6a-b). To investigate whether depletion of CD26⁺ cells near Achillies tendon prior to tendon injury would affect the infiltration of hematopoietic lineage cells during tendon healing, we collected the injured tendon tissues from vehicle and DTx group. Immunofluorescent staining and flow cytometry analysis found the infiltration of CD26⁺CD45⁺ hematopoietic lineage cells

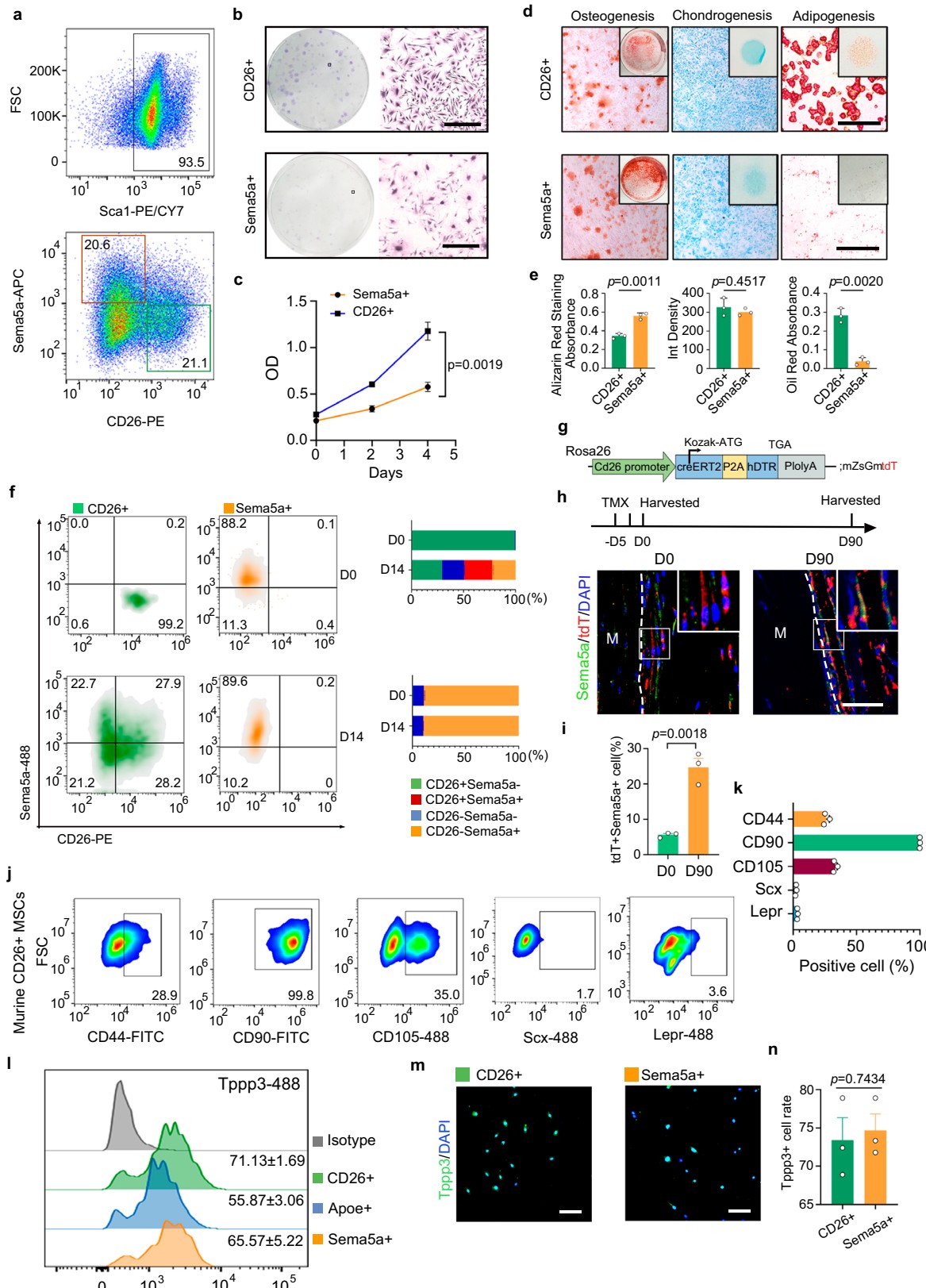

were not affected with DTx administration (Fig. S6c–g). In addition, local administration of DTx near Achillies tendon did not affect the circulation white blood cell profile in *CD26-CreER^{T2}-hDTR* mice (Fig. S6h). These results indicate that the way deplete CD26⁺ cells would not significantly affect the infiltration of hematopoietic lineage cells after tendon injury. To understand the contribution of CD26⁺

TSPCs in tendon healing, we treated *CD26-CreER^{T2}-hDTR* mice with DTx for 5 days to deplete these cells (Fig. 5a) and observed mis-aligned collagen fibrils and gaps in midsubstance in *CD26-CreER^{T2}-hDTR* mice at 30 days post injury (d.p.i) compared to vehicle control (Fig. 5b). Transmission electron microscopy (TEM) analysis showed that injured controls (Veh.) contained collagen fibrils with a shifted distribution

**Fig. 3 | CD26⁺ TSPCs are primitive stem cells. a** Fluorescence activated cell sorting (FACS) of Sca1⁺ CD26⁺ TSPCs and Sca1⁺Sema5a⁺ progenitors. **b** Proliferation assay of CD26⁺ TSPCs and Sema5a⁺ progenitors. n = 3 biological replicates. Scale bar: 100 μm. **c** CCK8 assay of CD26+ TSPCs and Sema5a+ progenitors. n = 3 biological replicates. Two-tailed unpaired Student's t test, t = 7.321, df=4. **d** Alizarin red, alcian blue and oil red O staining of CD26⁺ TSPCs and Sema5a⁺ progenitors under osteogenesis, chondrogenesis or adipogenesis. Scale bar: 100 μm. **e** Quantification of CD26⁺ TSPCs and Sema5a⁺ progenitors under osteogenesis, chondrogenesis or adipogenesis. n = 3 biological replicates. Two-tailed unpaired t test. Osteogenesis: t = 8.675, df=3.868; chondrogenesis: 0.881, df=2.625; adipogenesis: t = 9.796, df= 3.080. **f** Sort-purified CD26⁺ TSPCs and Sema5a⁺ progenitors from adult mice were analyzed before cultured (day 0). 14 days after cultivation, sorted cells were analyzed for the expression of CD26

and Sema5a. n = 3 biological replicates. **g** Scheme for the generation of *CD26-CreER^T2^-hDTR; H11-mZsGmtdT* mice. **h** Top: Scheme for the TMX induction, with the tdTomato (tdT) marking CD26⁺ TSPCs and chased up to 3 months. Bottom: Immunofluorescent images of tdTomato and Sema5a. Scale bar: 100 μm. **i** quantification of tdT⁺ Sema5a⁺ cells. n = 3 biological replicates. Two-tailed unpaired Student's t test, t = 7.352, df=4. **j, k** Flow cytometry analysis of stem cell and tenocyte markers in CD26⁺ TSPCs. n = 3 biological replicates. **l** Flow cytometry analysis of Tppp3 in CD26⁺ TSPCs, Apoe⁺ and Sema5a⁺ progenitors. n = 3 biological replicates. **m** Immunofluorescent images of Tppp3 in CD26⁺ TSPCs and Sema5a⁺ progenitors. Scale bar: 100 μm. **n** Quantification of Tppp3⁺ cell rate in each cluster. n = 3 biological replicates. Two-tailed unpaired Student's t test, t = 0.351, df=4. Data shown are mean ± SEM. Source data are provided as a Source Data file.

towards sizes <80 nM in diameter compared to uninjured controls. Injured DTx group showed structural gaps devoid of fibrils and increased proportion of fibrils with even smaller diameter (Fig. 5c, d). We found decreased expression of Col1a1, Scx and Fmod in DTx group (Fig. 5e, f). The mechanical properties of tendon were evaluated. The results showed that mechanical properties including Effective Modulus, Maximum Force and Stress at failure were decreased with depletion of CD26⁺ cells (Fig. 5g–j). These results indicate that CD26⁺ TSPCs contribute to proper tendon healing.

## CD26⁺ TSPCs contribute to trauma-induced HO

Most of the injury induced heterotopic ossification (HO) developed through endochondral ossification, in which chondrogenesis is essential and prominent at 14 dpi in tendon punch injury model[32,33] To investigate the contribution of CD26⁺ TSPCs in HO formation, pseudotime analysis with differentiated gene expression patterns across the lineage of CD26⁺ TSPCs to chondrocytes were performed and GO pathway analysis of upregulated genes towards chondrocytes revealed enrichment of pathways of cartilage development, chondrocyte differentiation, endochondral ossification, osteoblast differentiation and cartilage condensation (Fig. 6a). Notably, we observed the increased expression of marker genes of chondrogenesis (*Col2a1, Acan, Comp and Mmp13*) over pseudotime (Fig. 6b).

To confirm whether CD26⁺ TSPCs differentiated chondrocytes and osteoblasts during HO formation, we generated the tendon punch model in *CD26-CreER^T2^-hDTR; mGmT* mice (Fig. 6c). The migration of tdT⁺ cells into tendon midsubstance and the increased Sox9⁺tdT⁺ cells in midsubstance indicated that CD26⁺ TSPCs differentiated into chondrocytes during tendon injury (Fig. 6d, e). Meanwhile, at later stage after tendon injury, increased Sp7⁺tdT⁺ osteoblasts in the midsubstance were found (Fig. 6f, g), indicating that CD26⁺ TSPCs could differentiated into osteoblast. At 60 dpi when heterotopic bone was observed, tdT+Sox9+ chondrocytes were found near the heterotopic bone, and tdT+Sp7+ osteoblasts were resident within the heterotopic bone (Fig. 6d–g). Together, these results demonstrate that CD26⁺ TSPCs contribute to HO formation after tendon injury.

## TNC-Hippo signaling is involved in trauma-induced HO formation

To investigate the important role of CD26⁺ TSPCs in HO formation, *CD26-CreER^T2^-hDTR* mice were treated with DTx for 5 days to deplete CD26⁺ TSPCs (Fig. 7a). Micro-CT analysis showed that depletion of CD26⁺ TSPCs reduced the ectopic bone volume (Fig. 7b, c). Decreased expression of chondrogenic markers (Sox9 and Col2a1) and osteogenic markers (Sp7 and Ocn) at punch sites further confirmed the reduced ectopic bone formation after depletion of CD26⁺ TSPCs (Fig. 7d–k).

To investigate the molecular mechanism underlying the development CD26⁺ TSPCs into ectopic bone, we collected Achilles tendon tissues from punched and sham group 14 days after injury and performed bulk RNA sequencing. GO pathway analysis showed enrichment of extracellular matrix organization, ossification, bone

development, cartilage development and cartilage condensation, which further indicated the development of ectopic bone after tendon injury (Fig. 7l). Expression of chondrogenic markers (*Sox9, Acan, Col2a1*) were increased in punched group. In addition, genes positively regulate proliferation (*Cdk1* and *Mki67*) were also upregulated after tendon injury (Fig. 7m), which was consist to previous findings in our scRNA-seq data (Fig. S5a–d).

To uncover the secreted protein that driven CD26⁺ TSPCs developing into ectopic bone, we look through the upregulated genes (log2FC > 1.1, FDR < 0.05) and identified several genes related to chondrogenic differentiation and cartilage protection including *TNC* and *Wnt16* et al. (Fig. 7m). TNC is a large molecular extracellular matrix glycoprotein hexameric multidomain protein and is upregulated in inflammatory conditions but relatively lower in normal conditions[34,35]. We previously found TNC reduced adhesion force, increased phosphorylation of Yap and activate Hippo pathway during chondrogenesis[36]. To investigate whether Hippo pathway is involved in the process of HO formation, we performed GSEA pathway analysis in RNA-seq dataset and KEGG pathway analysis of upregulated genes in CD26⁺ TSPCs in scRNA-seq data. The results showed that enrichment of Hippo signaling pathway was observed in both datasets (Fig. 7n, o). To investigate the role of TNC-Hippo signaling in HO formation, tendon tissues from sham and punched group were collected and IHC staining showed that TNC and p-Yap were upregulated at injured tendon (Fig. 7p–q). Furthermore, we observed that during the process of tendon healing, about 90.00% at 30 dpi and 86.54% at 60 dpi of p-YAP⁺ cells were CD26⁺ cells (Fig. S7a-b). In vitro, TNC promoted chondrogenic differentiation of CD26⁺ TSPCs (Fig. S7c). TNC also increased phosphorylation of Yap and prevented its nuclear translocation (Fig. S7d, e). XMU-MP-1 treatment suppressed the expression of Sox9 and the chondrogenic differentiation of CD26⁺ TSPCs (Fig. S7f, g)[37].

To investigate the in vivo effect of TNC on ectopic bone formation, we generated tendon injury model in TNC knockout (TNC-/-) mice. CT scanning showed that ectopic bone volume was significantly reduced in TNC-/- mice compared to littermate controls (Fig. 7r, s). In addition, Targeting TNC with anti-TNC antibody also suppressed HO formation (Fig. S8a-b). Furthermore, administration of XMU-MP-1, a Hippo pathway antagonist that decreases the phosphorylation and degradation of Yap, suppressed ectopic bone formation (Fig. 7t–u). To further investigate whether targeting TNC would affect tendon healing process, the histological and mechanical properties of tendon were detected. The results showed that collagen fibril diameter and mechanical properties were not significantly affected in anti-TNC group compared to IgG control, indicating that targeting TNC did not significantly affect the process of tendon healing (Fig. S8c–g). To further investigate whether targeting Hippo signlaing pathway would affect tendon healing process, XMU-MP-1 was administrated and the results showed that collagen fibril diameter and mechanical properties were improved in XMU-MP-1 group compared to vehicle control, indicating that targeting

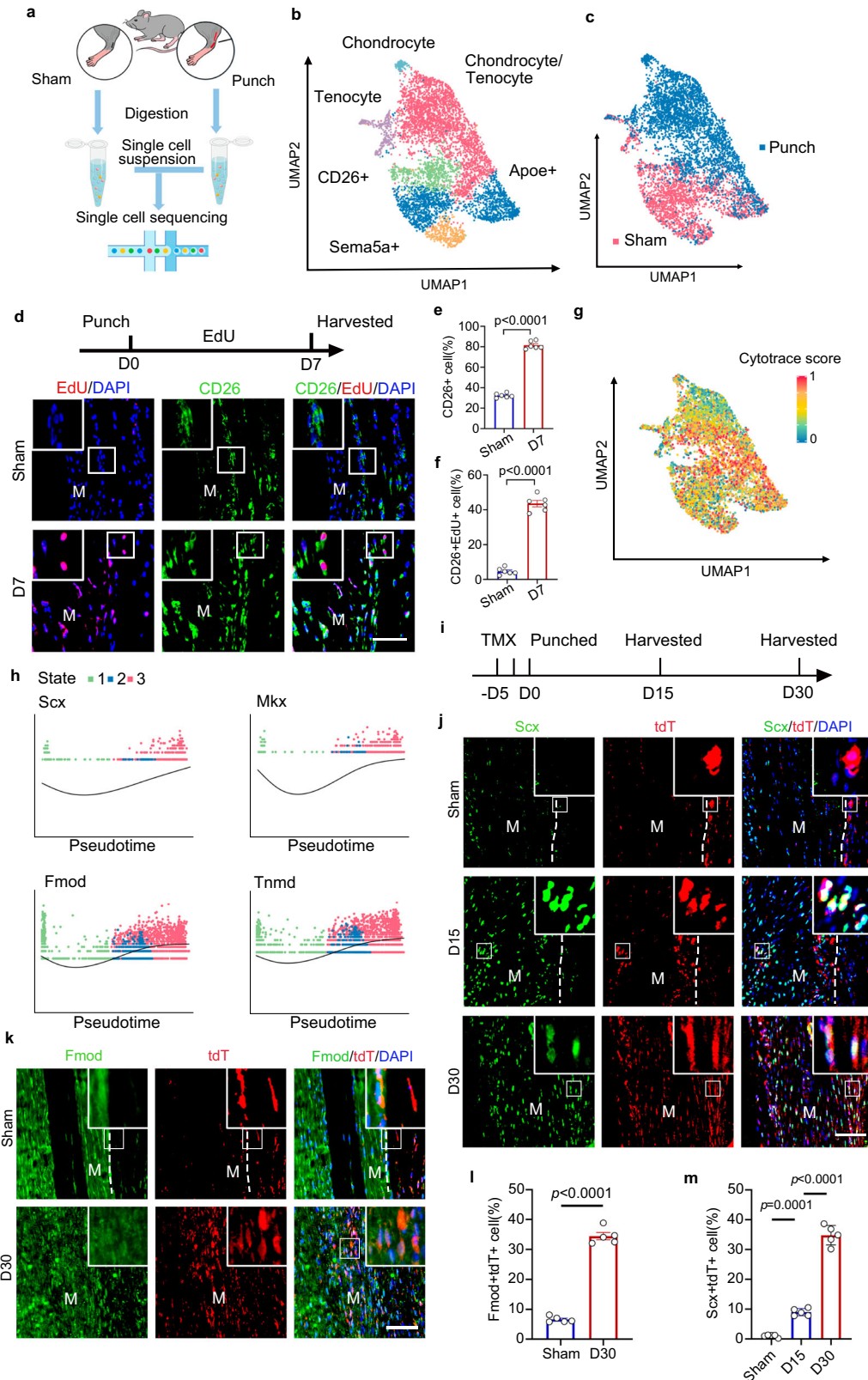

Hippo signaling pathway sightly improve the process of tendon healing (Fig. S8h–l). Collectively, these results indicate that TNC-Hippo signaling is involved in HO formation.

## Discussion

Tendon stem/progenitor cells (TSPCs) were regarded as potential cells for tendon healing[1,3,4]. Despite tentative studies have been carried out

for TSPCs, the identification of TSPCs remain elusive due to the following problems. Firstly, most previous studies rely solely on enzymatic isolation and in vitro characterization of TSPCs. Although these studies have confirmed the existence of TSPCs, the risk of cross-contamination of different tissues during isolation of TSPCs is relatively high due to methodological limitations, which may result in heterogeneous cultures[3–5]. Secondly, absence of well-defined and

**Fig. 4 | CD26⁺ TSPCs contribute to tendon healing. a** Scheme for scRNA-seq of sham and punch group. **b** UMAP plot of CD26⁺ TSPC, Apoe⁺ progenitor, Sema5a⁺ progenitor, chondrocyte/tenocyte, chondrocyte and tenocyte. **c** UMAP plot of sham and punch group. **d** Top: Scheme for EdU administration and collection of tendon tissues. Bottom: Immunofluorescent staining of CD26 and EdU in tendon from sham and punched group. Scale bar: 100 μm. **e** Quantification of CD26 positive cell rate in tendon from sham and punched group. n = 6 biological replicates. Two-tailed unpaired Student's t test, t = 27.72, df=10. **f** Quantification of CD26⁺EdU⁺ cell rate in tendon from sham and punched group. n = 6 biological replicates. Two-tailed unpaired Student's t test, t = 19.77, df=10. **g** UMAP plot of CytoTRACE analysis of cluster CD26⁺ TSPC, Apoe⁺ progenitor, Sema5a⁺ progenitor, chondrocyte/tenocyte, chondrocyte and tenocyte from uninjured and punch

group. **h** Dot plot of differentially expressed genes ordered based on their common kinetics through pseudotime displayed at different state. **i** Scheme for the generation of *CD26*-CreER^T2^-hDTR; H11-mZsGmtdT mice and TMX induction. **j** Immunofluorescent staining of Scx and tdTomato (tdT) in tendon from Sham, Day 15 and Day 30 group. n = 5. Scale bar: 100 μm. **k** Immunofluorescent staining of Fmod and tdTomato (tdT) in tendon from Day 0 and Day 30 group. n = 5. Scale bar: 100 μm. **l** Quantification of Fmod⁺tdT⁺cell rate in tendon from Sham and Day 30 group. n = 5. Two-tailed unpaired t test, t = 21.61, df= 5.055. **m** Quantification of Scx⁺tdT⁺ cell rate in tendon from Sham, Day15 and Day30 group. n = 5 mice. One way ANOVA with Tukey's post hoc test, F = 379.9. Data shown are mean ± SEM. Source data are provided as a Source Data file.

specific markers for TSPCs hinders the identification of TSPCs for endogenous tendon repair. Bi et al. found that TSPCs derived from tendon midsubstance were positive for surface antigen including stem cell antigen-1 (Sca-1), CD90, CD44 and negative for CD34 and CD45[7]. Michael et al. reported that murine TSPCs derived from tendon fascicle express higher level of Tnmd and Scx, while his and other groups found Tppp3 (Staverosky et al.), αSMA (Yin et al.), Nestin (Yin et al.), Musashi1 and Cd133 (Mienaltowski et al.) were abundant in TSPCs from epitendon and vasculature[5,9,13,38]. However, these studies were based on a 'cell isolation-marker determination' strategy and the markers used were less representative and specific. Novel and distinct markers for subpopulation could not be found due to the methodology limitations. Thirdly, the validation of tenogenic capacity of previously identified TSPCs were mainly carried out in vitro, whether these TSPCs participate in endogenous tendon healing remains uncertain. Only a few studies investigated the role of TSPCs in endogenous tendon repair. Howell et al. found Scx⁺ TSPCs, whose tenogenic capacity has been confirmed in vitro, failed to regenerate transected Achilles tendon in adult mice and contributed to fibrosis through in vivo Scx lineage tracing system[11]. These results indicate that in vitro validation of tenogenic capacity is less representative and robust. Collectively, these findings suggest that TSPCs may harbor heterogenic populations and their identification needs to be further clarified in different biological and pathological process during tendon healing.

In this study, we took the advantage of scRNA-seq and identified a population of CD26⁺ TSPCs. These TSPCs highly express mesenchymal markers (*Ly6a*, *Cd90* and *Pdgfrα*) but barely expressed tenocyte markers (*Scx*), bone marrow derived mesenchymal stromal cell (BMMSC) markers (LepR) and pericyte markers (*Cd146* and *Acta2*). CD26⁺ TSPCs were primarily located within the peritendon and exhibited the capacity for self-renewal and triple-lineage differentiation. Additionally, CD26+ TSPCs were also identified in peritendon samples from humans. CD26+ TSPCs derived from human Achilles tendons demonstrated self-renewal and triple-lineage differentiation properties.

In a tendon injury model, we found that CD26⁺ TSPCs could proliferated and migrated into midsubstance of tendon and differentiated into Scx⁺ TSPCs/tenocytes, indicating that these cells were more primitive TSPCs than Scx⁺ TSPCs/tenocytes and participate in tendon healing. Through pseudotime analysis during CD26⁺ TSPCs tenocyte differentiation, we observed upregulation of tenogenic genes (*Mkx*, *Tnmd*, *Scx* and *Kera*), enrichment of GO pathway related to collagen biosynthesis is essential for tendon healing. Upon depletion of CD26⁺ cells, we observed mis-aligned and thinner collagen fibrils and gaps in midsubstance. These results suggest that CD26⁺ TSPCs are responsible for tendon healing.

Through scRNA-seq, we also identified two other progenitors, namely Apoe⁺ progenitors and Sema5a⁺ progenitors. Sema5a+ progenitors displayed self-renewal property and only differentiate into chondrocytes and osteoblasts. Through pseudotime analysis, we identified a hierarchy from CD26⁺TSPCs to Apoe⁺ progenitors/Sema5a⁺ progenitors. In vitro, CD26⁺ TSPCs cultured in TSPC medium could loss CD26 expression and acquired expression of Sema5a⁺

progenitors' markers, indicating that CD26⁺ TSPCs could develop into Sema5a⁺ progenitors. On the contrary, Sema5a⁺ progenitors hardly acquired expression of CD26⁺ TSPCs markers. In vivo lineage tracing of CD26 also confirmed the hierarchy from CD26⁺ TSPCs to Sema5a⁺ progenitors. These findings suggest that CD26⁺ TSPCs are primitive cells. Since both Sema5a⁺ and Apoe⁺ progenitors could be detected under normal and tendon healing condition, we considered these two populations as permanent cell states. Nonetheless, we could not exclude the possibility that these two populations might be transient cell states and further lineage tracing experiments are needed to confirm this hypothesis. GO term analysis showed enrichment of immune system process, neuron differentiation, blood vessel development, skeletal system development, angiogenesis and chemotaxis in Apoe⁺ progenitors, indicating that these cells participate in the above biological processes after tendon injury. In addition, GO term analysis showed enrichment of skeletal system development, extracellular matrix organization, ossification, cartilage development and bone development in Sema5a+ progenitors, indicating that these cells participate in the above biological processes after tendon injury. We were unable to identify analogs populations of Apoe⁺ progenitors and Sema5a+ progenitors in human scRNA-seq dataset. In humans, MSC-2 was enriched in expression of Apoe, Sfrp1, Vcam1 and Pla2g2a, indicating that these cells were associated with lipid metabolism, which were partially resembled Apoe⁺ progeniros found in murine scRNA-seq dataset. As for MSC-3, the enrichment og Lcp1, Lyz, s100a12 and LST1 suggests that these cells are associated with immune response. Due to the differences between human and murine species, the signature genes of these two cell populations may exhibit significant disparities. Nonetheless, we found that the signature genes of CD26⁺ TSPCs were analogous in both murine and human scRNA-seq datasets. The identification of analogs populations of Apoe⁺ progenitors and Sema5a+ progenitors in human required larger samples to confirm.

Recently, Harvey et al. provided highly reliable evidences with an in vivo lineage tracing system that a population of Pdgfra⁺Tppp3⁺ TSPCs could generated new tenocytes during endogenous tendon repair[10]. Meanwhile, they found a population of Pdgfra⁺Tppp3⁻ tendon-fibro-adipogenic progenitors (T-FAPs) were primarily responsible for fibrosis[10]. Pdgfra⁺Tppp3⁺ TSPCs mainly resided within tendon sheath, which was similar to our identified CD26⁺ TSPCs. Through scRNA-seq and flow cytometry analysis, we found CD26⁺ TSPCs, Apoe⁺ progenitors and Sema5a⁺ progenitors contained both Tppp3⁺ and Tppp3⁻ subpopulations. Additionally, scRNA-seq data indicated that Pdgfra⁺Tppp3⁺ cells and Pdgfra⁺Tppp3⁻ cells contained CD26⁺ TSPCs, Apoe⁺ progenitors and Sema5a⁺ progenitors. Therefore, the classification of CD26⁺ TSPCs, Apoe⁺ and Sema5a⁺ progenitors depend on their lineage trajectory while the classification of Pdgfra⁺Tppp3⁺ TSPCs and Pdgfra⁺Tppp3⁻ T-FAPs is mainly based on their differential potentials. Furthermore, Pdgfra⁺Tppp3⁺ TSPCs exhibit chondrogenic and osteogenic potentials, while CD26⁺ TSPCs display triple-lineage potentials, indicating that these two TSPCs populations are different and CD26+ TSPCs are more primitive. Since

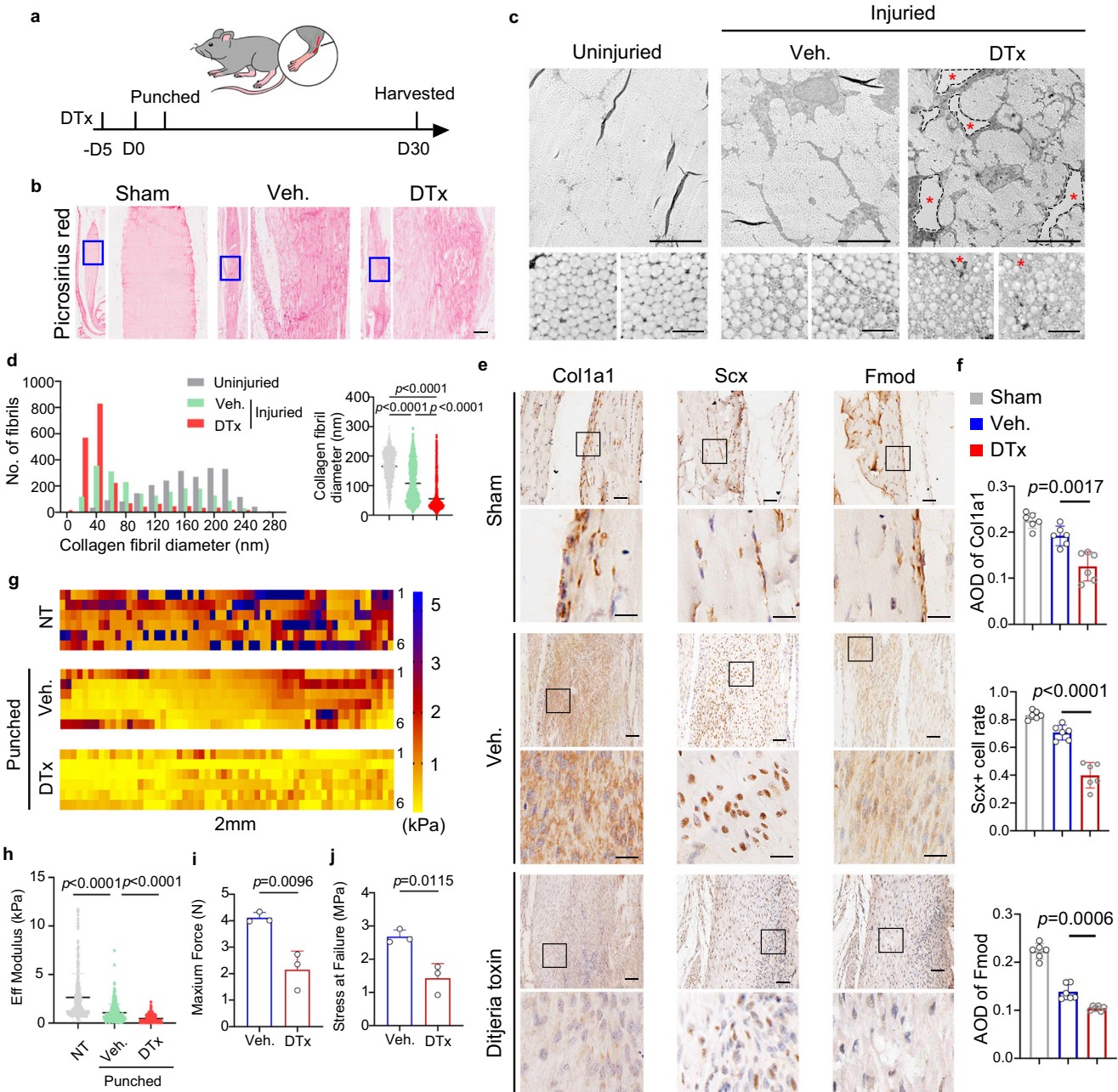

**Fig. 5 | Depletion of CD26⁺ TSPCs leads to poor tendon healing. a** Scheme for establishment of tendon injury model and administration of diphtheria toxin (DTx). **b** Picrosirius red staining of Achilles tendons from sham, Veh. and DTx group. n = 3 biological replicates. Scale bar: 100 μm. **c** TEM images of cross section of Achilles tendons from uninjured, Veh. and DTx group. The red asterisk indicates a hole in the matrix. n = 3 biological replicates. Scale bar: Top, 5 μm; Bottom, 500 nm. **d** Bar graph and dot plot presents the aggregated distribution of collagen fibril diameters using high-magnification TEM images from n = 3 mice per group. One way ANOVA with Turkey's test, F = 2128. **e** Histological analysis of tenogenic genes including Col1a1, Scx, and Fmod in injured tendon treated with diphtheria toxin or vehicle control. n = 6. Scale car: 100 μm. **f** Quantification of of Col1a1, Scx and Fmod. n = 6 biological replicates. Two-tailed unpaired Student's t test, Top: t = 4.258, df=10; Middle: t = 7.105, df=10; Buttom: t = 4.946, df=10. **g, h** Heatmap and quantification of effective modulus of tendons from uninjured, injured with DTx treatment and injured with Veh. treatment. Sixty points were analysed for each tendon (n = 6 per group). One way ANOVA with Turkey's test, F = 194.4. **i–j** Quantification of Maxium Force and Stress at failure of tendons from injured with DTx treatment and injured with Veh. treatment. n = 3 biological replicates. Two-tailed unpaired Student's t test, i: t = 4.653, df=4; j: t = 4.418, df=4. AOD, average of density. Data shown are mean ± SEM. Source data are provided as a Source Data file.

Tppp3 is not a surface antigen and therefore further investigation of the relationship between Pdgfra⁺Tppp3⁺ TSPCs and CD26⁺ TSPCs is limited. The role of Pdgfra⁺Tppp3⁺ TSPCs in endogenous tendon repair was investigated in a biopsy punch injury model with robust tendon healing. Whether these Pdgfra⁺Tppp3⁺ cells represent the entire TSPCs population and contribute to tendon healing in other severer injury model remains uncertain. Additionally, whether

Pdgfra⁺Tppp3⁺ TSPCs can be found in human tendon is unknown. Despite the fact that the different strategies of TSPCs classification might lead to overlapping and difference of these two TSPCs, we have confirmed that CD26⁺ TSPCs resident in peritendon is a highly primitive population that can regenerate tenocytes and participate in tendon healing. Furthermore, we provide a FACS strategy for TSPCs isolation and further investigation.

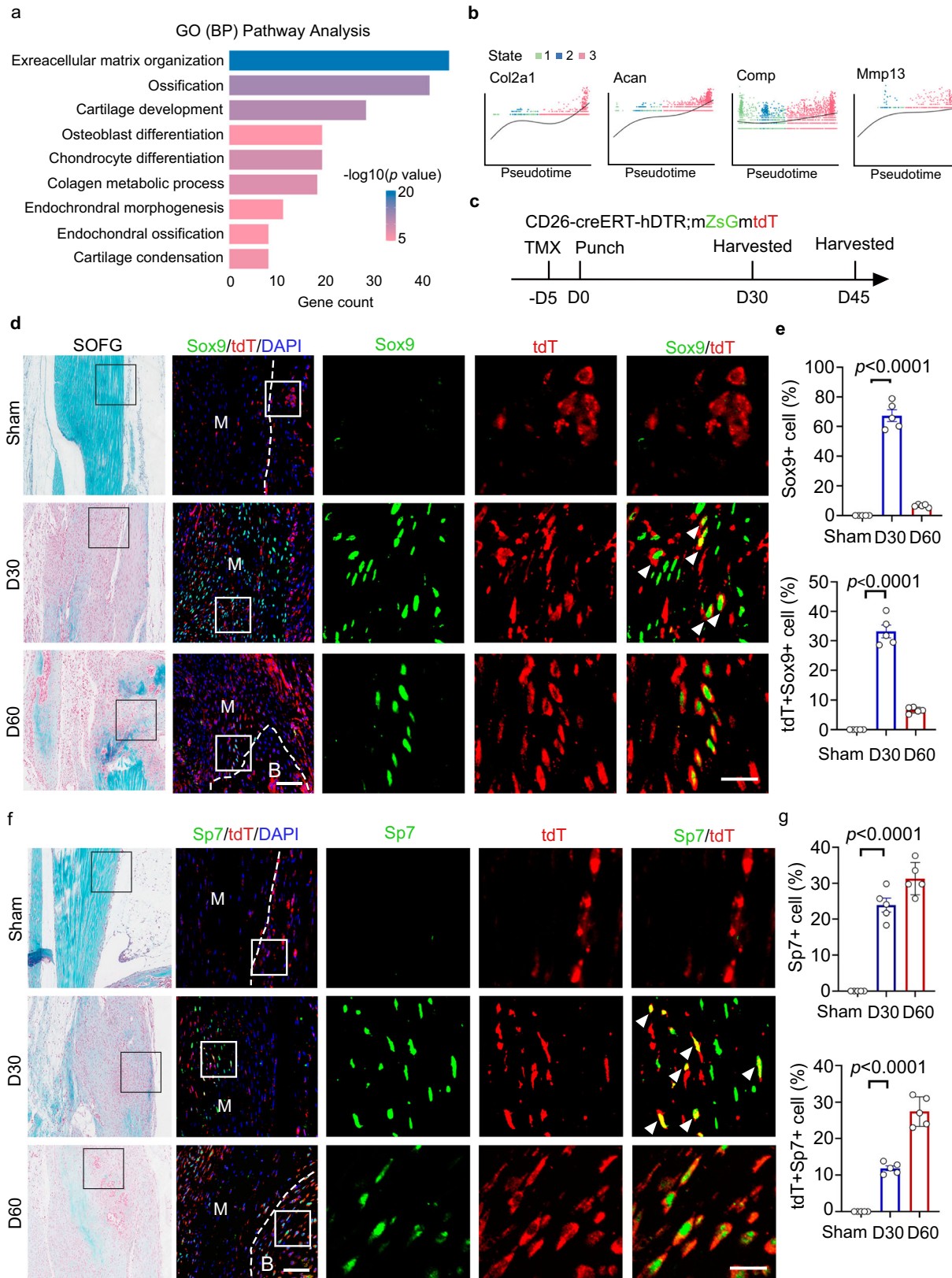

**Fig. 6 | CD26[+] TSPCs contribute to ectopic bone formation. a** GO pathway analysis of upregulated genes toward chondrocyte lineage based on their common kinetics through pseudotime. Hypergeometric test, FDR correction. **b** Pseudotime analysis of expression of chondrocyte markers in each state. **c** Scheme for punched model establishment and tamoxifen (TMX) administration. **d** Immunofluorescent staining of Sox9 and tdT. n = 5. Scale bar: 100 μm. **e** Quantification analysis of Sox9[+] cell and tdT[+]Sox9[+] cell rate. n = 5 biological replicates. One way ANOVA with Turkey's test, Top: F = 257.7; Buttom: F = 190.0. **f** Immunofluorescent staining of Sp7 and tdT. n = 5. Scale bar: 100 μm. **g** Quantification analysis of Sp7[+] cell and tdT[+]Sp7[+] cell rate. n = 5 biological replicates. One way ANOVA with Turkey's test, Top: F = 103.5; Buttom: F = 152.5. Data shown are mean ± SEM. Source data are provided as a Source Data file.

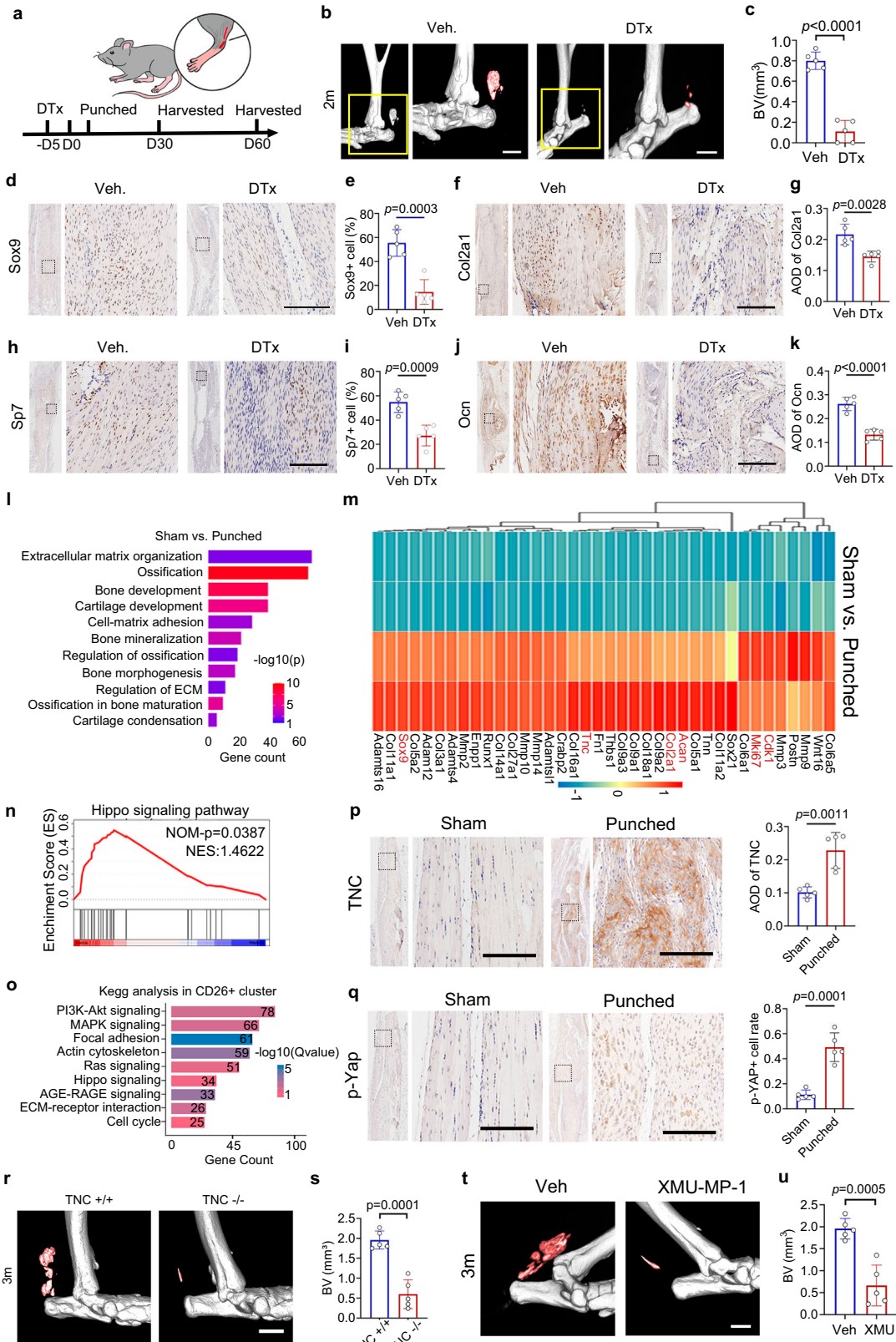

Heterotopic ossification (HO) is a severe complication of tendon injury which is largely thought to be related to the inflammatory response to the inciting injuries[14]. The aggregate data suggest that the predominant cell types underlying the initiation and progression of traumatic HO comprise of connective tissue-resident progenitor cells, marked by Prrx1, Scx, or AlphaV/CD105[14,16,39]. Agarwal et al. found

Prrx1-cre cells resided at hind paws and contributed to HO through in vivo lineage tracing system, but whether Prrx1[+] cells contributed to HO in adult remains uncertain[39]. Other studies have shown that Scx[+] TSPCs contributed to HO[11,22]. However, more Scx[-] cells were observed within the HO lesion, indicating that Scx[-] progenitors were ineligible source for HO. So far, the identification of these Scx[-] cells was unclear.

**Fig. 7 | TNC-Hippo signaling is involved in trauma-induced HO formation.**
**a** Scheme for establishment and tendon collection of punch models in CD26-ERT2-hDTR mice. **b, c** Micro-CT images and quantification of ectopic bone in Achilles tendons from diphtheria toxin (DTx) and vehicle control (Veh) group. n = 5 per group. Scale bar: 1 mm. Two-tailed unpaired Student's t test, t = 11.24, df=8. **d–k** Histological analysis and quantification of chondrogenic (Sox9 and Col2a1) and osteogenic (Sp7 and Ocn) markers. Scale bar: 100µm. n = 5 biological replicates. Two-tailed unpaired t test, e: t = 6.028, df=7.598; g: t = 4.253, df=8; i: t = 5.114, df=7.998; k: t = 8.008, df=8. **l** GO pathway analysis of upregulated genes in punched group compared to sham control. Hypergeometric test, FDR correction. **m** Heatmap of differentially expressed genes from sham and punch group. **n** GSEA analysis of upregulated genes in punched group compared to sham group. **o** KEGG

pathway analysis of upregulated genes in CD26[+] cluster from scRNA-seq data. **p** Histological analysis and quantification of TNC for Achilles tendon from sham and punched group. n = 5. Scale bar: 100 µm. Two-tailed unpaired Student's t test, t = 4.987, df=8. **q** Histological analysis and quantification of p-Yap for Achilles tendon from sham and punched group. n = 5. Scale bar: 100 µm. Two-tailed unpaired Student's t test, t = 7.044, df=8. **r–s** Micro-CT analysis and quantification of ectopic bone formation in TNC[+/+] and TNC[−/−] mice 3 months after punched injury. n = 5. Scale bar: 1 mm. Two-tailed unpaired Student's t test, t = 7.077, df=8. **t–u** Micro-CT analysis and quantification of ectopic bone formation in Veh. and XMU-MP-1 group 3 months after punched injury. n = 5. Scale bar: 1 mm. Two-tailed unpaired Student's t test, t = 5.553, df=8. AOD average of density. Data shown are mean ± SEM. Source data are provided as a Source Data file.

In addition, hind limb soft tissue derived bone-cartilage-stromal progenitors (BCSPs: AlphaV[+]/CD105[+]/Tie2[−]/CD45[−]/CD90[−]/6C3[−]) were found participated in HO formation in an in vivo cell transplantation model[40]. Although these investigations suggest that multiple types of progenitor cells may participate in the process of HO formation, whether depletion of progenitors marked by Prrx1, Scx, or AlphaV/CD105 could suppress HO formation was unknown. In the current study, we found that CD26[+] TSPCs could generate ectopic cartilage and bone during HO formation, indicating that these cells contributed to HO formation. The HO formation was significantly suppressed with depletion of CD26[+] TSPCs, indicating a critical role of CD26[+] TSPCs in the onset and development of HO. CD26[+] TSPCs are positive for CD90, Prrx1 and negative for Scx, which distinguishes them from Scx[+] TSPCs and CD90[−] BCSPs. These findings may explain the observation that numerous Scx[−] cells were also observed in the ectopic bone. Collectively, in the current study, we identified a population of TSPCs which is different from previously identified progenitors for HO and participate in the formation of HO.

The microenvironment plays a crucial role in determining cell fate. Accumulating evidence suggests that a pro-chondrogenic/osteogenic microenvironment is established after tendon injury, within which stem cells aberrantly activate osteogenic or osteochondrogenic programs, driven by mechanical transduction mechanisms, environmental signals, and intricate molecular crosstalk[16]. Our findings indicate that CD26[+] TSPCs differentiate into osteoblasts and chondrocytes after tendon injury. The pro-chondrogenic/osteogenic microenvironment after tendon injury partially explained the aberrant differentiation toward chondrocytes and osteoblasts instead of normal tendon healing of CD26[+] TSPC. In this microenvironment, secreted proteins in cell niche are pivotal for cell fate determination and are hold promises as targets for the translation from basic science to clinical setting. TNC is a large molecular extracellular matrix glycoprotein hexametric multidomain protein and is upregulated in inflammatory conditions but relatively lower in normal conditions[34,35,41,42]. Our previous study identified that overexpression of TNC promoted chondrogenesis and subsequent pathological bone formation in ankylosing spondylitis[36]. In this study, we further observed increased TNC expression in injured sites through RNA-seq analysis and IHC staining. Knockdown of TNC led to reduced HO formation. In vitro studies further confirmed TNC-induced chondrogenesis was dependent on Hippo signaling and inhibition of this signaling suppressed the chondrogenesis of CD26[+] TSPCs. Most of the injury-induced HO developed through endochondral ossification[15]. Endochondral ossification involves several stages: chondrogenic differentiation of mesenchymal stromal cells, cartilage template formation, hypertrophy of chondrocytes, vascular invasion, and subsequent bone formation. Initially, cells in the center of mesenchymal condensations differentiate into chondrocytes, which begin to produce a matrix composed of collagen and proteoglycan, forming the cartilage template. Subsequently, the cells defining the border withdraw from the cell cycle and undergo hypertrophy, prompting the differentiation of osteoblasts from the

perichondrium. Lastly, blood vessels infiltrate the hypertrophic cartilage, triggering osteoblast differentiation and eventual formation of the bone marrow cavity[43]. Indeed, previous studies have shown that the role of Yap signaling in chondrogenesis and osteogenesis is opposite. In chondrogenesis, phosphorylation of Yap leads to its degradation and promotes chondrogenic differentiation, while Yap nuclear translocation promotes osteogenesis[44,45]. We found aberrantly expression of TNC promotes phosphorylation of Yap, resulting in increased chondrogenesis of TSPCs and leads to heterotopic ossification. Huber et al. demonstrate that knockdown of Yap could suppress osteogenic differentiation of mesenchymal progenitor cells and subsequent HO formation, which is consistent with previous findings that Yap is essential for osteogenesis[46]. In the current study, we found that TNC aberrantly expressed at the stage of cartilage template formation and promotes chondrogenesis through promoting the phosphorylation of Yap. Since chondrogenesis and cartilage template formation occur at the initial stage of HO formation, inhibiting Yap degradation at this stage mainly suppresses the process of chondrogenesis, thereby alleviating HO derived from endochondral ossification. Nonetheless, at the stage of bone formation, TNC was not aberrantly expressed and therefore did not regulate the Hippo pathway. Huber and our study demonstrated the different role of Yap in chondrogenesis and osteogenesis, which is consistent with previous studies[44–46]. Since targeting TNC-Hippo signaling suppressed HO formation while did not retard the tendon healing process, indicating that targeting TNC-Hippo signaling provides a potential therapeutic strategy for HO formation.

HO of the tendon is common in populations with high risk of tendon injury[15]. Experimental tendon injury models offer a framework for investigating the mechanisms underlying tendon injury and healing. However, due to the complex nature of tendon injury and repair mechanisms, animal models aiming to reproduce the pathogenesis process do not perfectly mimic the scenario. Tendon injury model includes chemically-induced model, mechanical load-induced model and mechanical injury model (partial injury or total transection) model[15]. Chemically-induced models are attractive because they require less time and resources[1]. Mechanical load-induced models were used to elucidate the molecular and cellular mechanisms underlying tendinopathy and tendon healing in different scenario such as tendon immobilization or overuse[1]. Tendon partial injury models, established through punch injury or partial transection, provide insight into observation of tendon healing process at specific region and investigate the effects of various factors and biomaterials on tendon repair, as well as complications such as fibrosis and heterotopic ossification[10,32,47,48]. Tendon total transection model was used to study molecular process of tendon repair, identify growth factors with potential use in treating tendon disorders and investigate different strategy of surgical repair[15,33]. Nonetheless, in this model, tendon torn ends healing would be disturbed without surgical intervention in this model[15]. In this study, we aimed to investigate the cell populations and underlying mechanisms during tendon healing and subsequent heterotopic ossification. Therefore, we chose the tendon punch injury

model, in which both tendon healing and heterotopic ossification occur[10,32,33,47].

The current study has some limitations. Firstly, it focused on scRNA-seq of single-cell suspensions derived from tendon tissues, without incorporating FACS-sorted CD26+ lineage-traced cells from multiple timepoints, which is challenging but would have enhanced the understanding of the characteristics and differentiation pathways of CD26+ TSPCs. Secondly, the role of CD26 TSPCs and Hippo signaling in other tendon injury models remains uncertain and worth further investigations to established Hippo signaling condition knockout model to validate its contributions to tendon healing in different scenarios. Thirdly, the involvement of Apoe+ or Sema5a+ progenitors in tendon healing and HO formation remains uncertain. Utilizing Apoe or Sema5a lineage-tracing mice would provide a valuable tool for future investigations into the differentiation fate of these cells during tendon healing and HO formation.

In summary, we identified a population of TSPCs with conservative property across species. This CD26+ TSPCs represent primitive TSPC with triple-lineage differentiation capacity and capable of regenerating tenocytes, thereby participating in tendon healing and HO formation. Furthermore, TNC plays an crucial role in HO formation. These findings contribute to a comprehensive understanding of the cellular and signaling mechanisms underlying tendon healing and HO formation, and offer a potential therapeutic strategy for preventing the onset and progression of traumatic HO.

## Methods

All procedures were approved by Institutional Animal Care and Use Committee, Sun yat-sen University and IEC for Clinical Research and Animal Trials of the First Affiliated Hospital of Sun Yat-sen University.

### Animals

Six- to eight-week-old C57BL/6 J, *CD26-CreER^T2-P2A-hDTR, H11-CAG-Loxp-ZsGreen-Stop-LoxP-tdTomato* (*H11-mZsGmtdT*) and TNC KO C57BL/6 J mice were established and purchased from the Gem-Pharmatech and Cyagen. Inbred, mice weighing between 18-22 g were obtained and always housed on a 12 h light/dark cycle, 22-28°C and 40-60% humidity with unrestricted access to food and water. For generation of *CD26-creERT2-hDTR* mice, the CRISPR/Cas9 technology was utilized to precisely integrate the *Dpp4-creERT2-P2A-hDTR-PolyA* gene cassette into the Rosa26 locus of C57BL/6 J mice. The brief procedure is as follows: sgRNA was transcribed in vitro, and a donor vector was constructed. Cas9, donor vector, and sgRNA were co-injected into fertilized eggs of C57BL/6 JGpt mice. F0 generation mice were obtained and validated for correct integration via PCR and sequencing. Positive F0 mice were then bred with C57BL/6JGpt mice to generate a stable F1 generation of mice carrying the genetic modification. Genotyping of all Cre-knock in mice and *H11-mZsGmtdT* C57BL/6J mice was performed by PCR using primers detecting the Cre sequence and H11 locus. For *CD26-creER^T2* induction, mice were injected intraperitoneally with 100 mg/kg daily tamoxifen for 5 days (Sigma). For inducible ablation of CD26+ cells at the injury sites, each mouse was locally injected with 50 ng diphtheria toxin (List Labs) daily for 5 days. For generation of TNC KO mice, exon 2−4 of Tnc-201 transcript is recommendedas the knockout region. In this project we use CRISPR/Cas9 technology to modify *Tnc* gene. The brief process is as follows: gRNA was transcribed in vitro. Cas9 and gRNA were microinjected into the fertilized eggs of C57BL/6 J mice. Fertilized eggs were transplanted to obtain positive F0 mice which were confirmed by PCR and sequencing. Astable F1generationmouse model was obtained by mating Positive F0 generation mice with C57BL/6JGpt mice. All mice were maintained in pathogen-free conditions, and all procedures were approved by Institutional Animal Care and Use Committee, Sun yat-sen University.

### Human ligaments and tendons

For human spinal ligament and tendon cell isolation, unaffected Achilles tendons from amputated limbs from 3 osteosarcoma patients (2 male and 1 female, average age: 20 years old) and spinal ligament tissues from 3 patients (2 male and 1 female, average age: 20.7 years old) who fulfilled the criteria for correction of scoliosis were collected. All samples were collected from the patients who had provided informed consent at The First Affiliated Hospital, Sun Yat-sen University. All procedures were approved by The Medical Ethics Committee of the First Affiliated Hospital of Sun Yat-sen University.

### Immunofluorescence

Histologic evaluation was performed at indicated time points in wild-type and *CD26-CreER^T/mGmT^fl/fl* and CD26^DTR/- tandem reporter mice following tendon injury. Tendon tissues were carefully dissected and fixed in 4% paraformaldehyde for 12 h in 4°C. 6 µm thick sections were prepared and pretreated with xylene and different concentrations of ethanol. All specimens underwent citrate buffered heat antigen retrieval at 90 °C for 30 min except cell samples. Sections or cell samples were subsequently blocked with 10% serum. Sections were labeled with the following antibodies: anti-Dpp4 antibody (Abcam, 1:100, CAT: ab187048); anti-Dpp4 antibody (Abcam, 1:100, CAT: ab114033); anti-Lama4 antibody (Abcam, 1:100, CAT: ab242198); anti-Fmod antibody (Proteintech, 1:100, CAT: 60108-1-Ig); anti-Sema5a (R&D system, 1:100, CAT: AF5896-SP); anti-Tppp3 (Abcepta, 1:100, CAT: AP5004b); anti-Sox9 antibody (Abcam, 1:100, CAT: ab185230); anti-Sox9 antibody (Abcam, 1:100, CAT: ab238053); anti-Sp7 antibody (Abcam, 1:100, CAT: ab209484); anti-Apoe (Abcam, 1:100, ab183597); anti-Scx antibody (Abcam, 1:100, CAT: ab58655); anti-Tdtomato antibody (Biorbyt, 1:100, CAT: orb182397); anti-colagen II antibody (Abcam, 1:100, CAT: ab34712); anti-Ocn antibody (Abcam, 1:100, CAT: ab93876); anti-Yap1 antibody (Abcam, 1:100, CAT: ab205270); Donkey anti rabbit 488 antibody (Abcam, 1:100, CAT: ab150073); Donkey anti rabbit 568 antibody (Abcam, 1:100, CAT: ab175470); Donkey anti rabbit 647 antibody (Abcam, 1:100, CAT: ab150075); Donkey anti mouse 488 antibody (Abcam, 1:100, CAT: ab150108);. Donkey anti mouse 594 antibody (Abcam, 1:100, CAT: ab175470); Donkey anti goat 647 antibody (Abcam, 1:100, CAT: ab150135). Appropriate primary antibody and biological negative controls were run simultaneously with each tested sample.

Antibody labeled sections were imaged with epifluorescent upright scope (Leica) equipped with DAPI, and high-resolution digital camera. Each site was imaged in all channels and images were overlaid in Leica LAX Viewer or Photoshop before examination and quantifications in Image J. Images were adjusted only in brightness and contrast identically across comparison groups for clarity were indicated.

### Immunohistochemistry

Histologic evaluation was performed at indicated time points in wild-type and CD26-CreER^T/mGmT^fl/fl and CD26^DTR/- tandem reporter mice following tendon injury. 6 µm thick sections were prepared and pretreated with xylene and different concentrations of ethanol. All specimens underwent citrate buffered heat antigen retrieval at 90 °C for 30 min. After washing in PBS, the sections were incubated in 3% hydrogen peroxide for 20 min for blocking with endogenous peroxidase and followed by washing with PBS. Next, the slides were blocked with 10% goat serum for 1 h and then probed with following antibodies: anti-Scx antibody (Abcam, 1:100, CAT: ab58655); anti-Col1a1 antibody (Abcam,1:100, CAT: ab270993); anti-Tnc antibody (Abcam, 1:100, CAT: ab108930); anti-Fmod antibody (Proteintech, 1:100, CAT: 60108-1-lg); anti-Tnmd antibody (Abcam, 1:100, CAT: ab203676); anti-Sox9 antibody (Abcam, 1:100, CAT: ab185230); anti-Sp7 antibody (Abcam, 1:100, CAT: ab209484); anti-colagen II antibody (Abcam, 1:100, CAT: ab34712); anti-Ocn antibody (Abcam, 1:100, CAT: ab93876); anti-Dpp4 antibody (Abcam, 1:100, CAT: ab187048); anti-p-Yap1 antibody

(Abcam, 1:100, CAT: ab76252). The next day, the sections were washed off with PBS and the slides were probed with HRP secondary antibody (1:200) for 1 h at room temperature. Color development was achieved by treatment with the chromogen DAB (Servicoeb) and was carried out for 5-10 min under a microscope. Hematoxyline staining was also done at the end for nuclear staining. Appropriate primary antibody and biological negative controls were run simultaneously with each tested sample.

## Achilles tendon punch

Mice aged 9 weeks received presurgical analgesia consisting of 0.1 mg/kg buprenorphine, followed by anesthesia with inhaled isoflurane, and close postoperative monitoring with analgesic administration. Achilles' tendon received 10 times of punch injury with 0.55 mm needle. The tenotomy site was closed with a single 5–0 vicryl stitch placed through the skin only.

## Histology

For haematoxylin and eosin (H&E), safranin O and fast green (SOFG) and Picrosirius Red staining (Sigma), sections were pre-treated with xylene and different concentrations of ethanol and then processed according to the manufacturer's instructions.

## FACS

**Mouse studies.** Single cell suspension of Achiles tendon or tail tendon from -50 to 70 mice (8 weeks old) were pooled and resuspended in FACS buffer for incubation with the following antibodies for 30 min at 4°C: CD26 (Dpp4)-PE (Bioledgend, 1:100, CAT: 137803); CD90-PE/CY7 (Bioledgend, 1:100, CAT: 140310); CD90-FITC (Biolegend, 1:100, CAT: 140303); CD31-PB450 (Bioledgend, 1:100, CAT: 102421); CD45-PB450 (Bioledgend, 1:100, CAT: 103125); CD44-FITC (Bioledgend, 1:100, CAT: 103021); Ly6a-PE/CY7 (Bioledgend, 1:100, CAT: 108113); CD105-488 (eBioscience, 1:100, CAT: 53-1051-82); CD200-PE/CY7 (Bioledgend, 1:100, CAT: 123818); anti-leptin receptor (Abcam, 1:100, CAT: ab216690); anti-Tppp3 (Abcepta, 1:100, CAT: AP5004b); anti-Apoe (Abcam, 1:100, ab183597); anti-Sema5a (R&D system, 1:100, CAT: AF5896-SP), then incubated with zombie violet fixable viability kit for 10 min at 4°C. The cells were washed three times with FACS buffer to remove unbound antibodies. The cells were sorted with a BD FACS Aria cell sorter (BD Biosciences) equipped with a 100-mm nozzle and the following lasers and filters: Zombie violet, 405 and 450/50 nm; FITC, 488 and 515/20 nm; PE, nm; PE/Cy7, 532 and 780/60 nm; and APC/Cy7 and APC nm. All compensation was performed at the time of acquisition by using single-color staining and negative staining.

**Human studies.** Single cell suspension of Achilles tendon or spinal ligaments from patients were pooled and resuspended in FACS buffer for incubation with the following antibodies for 30 min at 4°C: CD26-PE (Bioledgend, 1:100, CAT: 302705); CD31-APC/CY7 (Bioledgend, 1:100, CAT: 303119); CD45-APC/CY7 (Bioledgend, 1:100, CAT: 304014); CD44-FITC (Bioledgend, 1:100, CAT: 103021); CD105-FITC (Bioledgend, 1:100, CAT: 323203); CD90-FITC (Bioledgend, 1:100, CAT: 328107); anti-leptin receptor (Abcam, 1:100, CAT: ab216690), then incubated with zombie violet fixable viability kit for 10 min at 4°C. For leptin receptor staining, secondary goat anti-rabbit 488 antibody (Abcam, 1:200, CAT: ab150077) was added. The cells were washed three times with FACS buffer to remove unbound antibodies. The cells were sorted with a BD FACS Aria cell sorter (BD Biosciences) equipped with a 100-mm nozzle and the following lasers and filters: Zombie violet, 405 and 450/50 nm; FITC, 488 and 515/20 nm; PE, nm; PE/Cy7, 532 and 780/60 nm; and APC/Cy7 and APC nm. All compensation was performed at the time of acquisition by using single-color staining and negative staining.

**Differentiation assay.** To induce osteogenesis, cells were plated in a 12-well plate at a density of $0.9*10^5$ cells/well and cultured in DMEM medium supplemented with 10% fetal bovine serum (FBS) for 24 h. The cells were then switched to osteogenic medium consisting of aminimum essential medium supplemented with 10% FBS, 50 μg/ml L-ascorbic acid, 0.1 μM dexamethasone and 10 mM b-glycerophosphate in DMEM medium to induce osteogenesis. The medium was changed every 3–4 days and after 14 days, cells were stained with alizarin red.

To induce chondrogenic differentiation, $1*10^5$ cells were seeded as micro-mass in the middle of a 24-well plate and cultured in DMEM medium supplemented with 10% FBS. Cells could attach for 1 h at 37 °C, after which chondrogenic differentiation medium (Pythonbio, CAT: AAPR219-500) were added to the wells. Medium was refreshed every other day, and after 9 days, micromasses were stained with alcian blue.

To indue adipogenic differentiation, $0.9*10^5$ cells cells were plated in a 12-well plate and cultured in DMEM medium supplemented with 10% FBS for 24 h. The cells were then switched to adipogenic medium (Stemcell, CAT: 05412). The medium was changed every 3–4 days and after 14 days, cells were stained with Oil red O.

**Proliferation assay.** For CFU assay, different sorted cells were seeded at the concentration of 1000 cells per dish with 10 cm diameter and cultured in DMEM medium supplemented with 10% FBS for 9 days. Then cells were stained with crystal violet dye.

For CCK8 assay, different sorted cells were seeded at the concentration of 1200 cells per well in a 96-well plate and cultured in DMEM medium supplemented with 10% FBS. The CCK8 assay was performed according to instruction manual.

**EdU administration and staining.** EdU (0.5 mg/ml in PBS) was administered by i.p. injection at 10 μl per g body weight daily and tendon tissues were collected at 7 dpi since previous study found TSPCs proliferated at this stage[10]. EdU staining was performed according to the protocol provided by manufacturers (Cell-Light Apollo 567 Stain Kit, CAT: C1037-1).

**TEM sample preparation and analyses.** Achilles tendons were isolated at 30 d.p.i, and fixed in stationary kit (Servicebio, G1123). TEM images of transverse sections were collected at several magnifications for evaluation. Collagen fibril diameters were measured using Fiji; 3 representative images were analysed for each tendon (n = 3 per group) and about 2000 fibrils analysed per sample.

**Effective modulus detection.** Achilles tendons were isolated at 30 d.p.i, and prepared in PBS. Effective modulus was detected by OPTICS11 according to the standard protocol; 60 points were analysed for each tendon. Stiffness refers to the measure of a material's resistance to deformation. The effective Young's modulus is a more specific term that quantifies the material's stiffness, which is the slope of the stress-strain curve over a particular range of deformation.

**Mechanical properties testing.** Mechanical testing was performed using a mechanical testing system (DR-507AS). The hind limbs were collected and tissues spanning the ankle, except for Achilles tendon, were transected. The muscle-tendon-calcaneus complex was rigidly fixed to clamps. Mechanical properties were evaluated by a load-to failure test at an elongation rate of 10 mm/min. Maximum force is the greatest amount of force a material can withstand before it begins to experience permanent deformation or failure. Stress at failure is the stress a material is under when it fails or breaks. It is calculated by dividing the applied force at the point of failure by the cross-sectional area of the material. This value indicates the strength of the material.

**MicroCT.** All specimens were obtained from mice at indicated time points post-mortem and fixed with 4% paraformaldehyde. For µCT scanning, specimens were fitted in a cylindrical sample holder and scanned using a Scanco lCT40 scanner set to 55 kVp and 70 lA. For visualization, the segmented data were imported and reconstructed as three-dimensional images using MicroCT Ray V3.0 software (Scanco Medical). Quantitative analysis was done by drawing a volume of interest. Bone volume was calculated inside this region after measuring the threshold for non-bone versus bone tissue, with the threshold value automatically determined.

### White blood cell profile detection
For inducible ablation of CD26+ cells at the injury sites, each mouse was locally injected with 50 ng diphtheria toxin (DTx) daily for 5 days. At day 3 after last time of DTx injection, peripheral blood from PBS and DTx group were collected. White blood cell profiles were detected with auto hematology analyzer (BC-5000 Vet) according to operation manual.

### Single-cell RNA-seq with 10× Genomics chromium platform
Tendon tissues were collected and dissected into pieces. Then they were digested in a mix of collagenase II (Roche, 3 mg/ml) and dispase II (Sigma-Aldrich, 4 mg/ml), prepared in DMEM for 1 h at 37 °C. Digestions were subsequently quenched with 10% FBS DMEM and filtered through 40µm sterile strainers. Cells were then washed in PBS with 0.04% BSA, counted and resuspended at a concentration of ~1000 cells/µl. Cell viability was assessed with Trypan blue exclusion on a Countess II (Thermo Fisher Scientific) automated counter and only samples with >85% viability were processed for further sequencing. Cellular suspensions were loaded on a 10X Genomics GemCode Single-cell instrument that generates single-cell Gel Bead-In-EMlusion (GEMs). Libraries were generated and sequenced from the cDNAs with Chromium Next GEM Single Cell 3' Reagent Kits v3.1. Upon dissolution of the Gel Bead in a GEM, primers containing (i) an Illumina® R1 sequence (read 1 sequencing primer), (ii) a 16 nt 10x Barcode, (iii) a 10 nt Unique Molecular Identifier (UMI), and (iv) a poly-dT primer sequence were released and mixed with cell lysate and Master Mix. Barcoded, full-length cDNAs were then reverse-transcribed from poly-adenylated mRNA. Silane magnetic beads were used to remove leftover biochemical reagents and primers from the post GEM reaction mixture. Full-length, barcoded cDNAs were then amplified by PCR to generate sufficient mass for library construction. R1 (read 1 primer sequence) were added to the molecules during GEM incubation. P5, P7, a sample index, and R2 (read 2 primer sequence) were added during library construction via End Repair, A-tailing, Adaptor Ligation, and PCR. The final libraries contained the P5 and P7 primers used in Illumina bridge amplification. The Single Cell 3' Protocol produced Illumina-ready sequencing libraries. A Single Cell 3' Library comprised standard Illumina paired-end constructs which begin and end with P5 and P7. The Single Cell 3' 16 bp 10x Barcode and 10 bp UMI were encoded in Read 1, while Read 2 was used to sequence the cDNA fragment. Sample index sequences were incorporated as the i7 index read. Read 1 and Read 2 were standard Illumina® sequencing primer sites used in paired-end sequencing.

### Data quality control and gene expression quantification
10X Genomics Cell Ranger software (version 3.1.0) was used to convert raw BCL files to FASTQ files, alignment and counts quantification. The analysis of scRNA-seq data was performed on website (https://www.omicsmart.com/10X/home.html#/). Briefly, reads with low-quality barcodes and UMIs were filtered out and then mapped to the reference genome. Reads uniquely mapped to the transcriptome and intersecting an exon at least 50% were considered for UMI counting. Before quantification, the UMI sequences would be corrected for sequencing errors, and valid barcodes were identified based on the EmptyDrops method. The cell by gene matrices were produced via UMI counting and cell barcodes calling. We filtered out cells with less than 300 genes per cell, with more than 10% mitochondrial read content and nCount over 15000.

### Dataset integration and cell clustering
We used the R (version 4.2.2) and Seurat package to perform the integration of datasets. We first merged datasets using Merge() function and then normalize data using NormalizeData() and then identified the top 2000 highly variable genes by FindVariableFeatures(). Next, we scaled the seuratobject using ScaleData() and RunPCA() with informative features. We used package Harmony to correct the batch effects. Seurat implements a graph-based clustering approach. Distances between the cells were calculated based on previously identified PCs. Briefly, Seurat embed cells in a shared-nearest neighbor (SNN) graph, with edges drawn between cells via similar gene expression patterns. To partition this graph into highly interconnected quasi-cliques or communities, we first constructed the SNN graph based on the Euclidean distance in PCA space and refined the edge weights between any two cells based on the shared overlap in their local neighborhoods (Jaccard distance). We then cluster cells using the Louvain method to maximize modularity. For visualization of clusters, UMAP were generated using the same PCs.

### Differentially expressed genes analysis
The analysis of data was performed on website (https://www.omicsmart.com/10X/home.html#/). Expression value of each gene in given cluster were compared against the rest of cells using Wilcoxon rank sum test. Significant upregulated genes were identified using a number of criteria. First, genes had to be at least 1.28-fold overexpressed in the target cluster. Second, genes had to be expressed in more than 25% of the cells belonging to the target cluster. Third, $p$ value is less than 0.05.

### Pseudotime analysis
The analysis of scRNA-seq data was performed on website (https://www.omicsmart.com/10X/home.html#/). Pseudotemporal analysis was performed on a filtered subset of clusters (cluster 1 to 4) from the uninjuried Achilles tendon. Ordering genes were selected by using a cutoff of expression in at least 10 cells and a combination of inter-cluster differential expression and dispersion with a q value cutoff of $< 1 \times 10\text{-}10$. A split heatmap was generated from selected genes showing significant change through pseudotime, high differential expression, or known biologic identity by using the Monocle function plot_genes_branched_heatmap at branch_point 1.

### GO enrichment analysis
Gene Ontology (GO) is an international standardized gene functional classification system which offers a dynamic-updated controlled vocabulary and a strictly defined concept to comprehensively describe properties of genes and their products in any organism. GO has three ontologies: molecular function, cellular component, and biological process. The basic unit of GO is GO-term. Each GO-term belongs to a type of ontology. GO enrichment analysis provides all GO terms that significantly enriched in differentially expressed genes comparing to the genome background and filter the differentially expressed genes that correspond to biological functions. Firstly, all peak related genes were mapped to GO terms in the Gene Ontology database (http://www.geneontology.org/), gene numbers were calculated for every term, significantly enriched GO terms in differentially expressed genes comparing to the genome background were defined by hypergeometric test. The calculated $p$-value were gone through FDR Correction, taking FDR ≤ 0.05 as a threshold. GO terms meeting this condition were defined as significantly enriched GO

terms in differentially expressed genes. This analysis was able to recognize the main biological functions that differentially expressed genes exercise.

## KEGG enrichment analysis

A differential gene expression analysis was initially conducted on the CD26[+] TSPCs of the uninjured and injured groups to identify a set of genes that are highly expressed in the injured group ($p < 0.05$). Subsequently, all peak related genes were mapped to KEGG (Kyoto Encyclopedia of Genes and Genomes) database and we performed a KEGG enrichment analysis with on this gene set. The calculated $p$-value were gone through FDR Correction, taking FDR $\leq 0.05$ as a threshold.

## Mice model intervention

Mice with tendon injuries received treatment intraperitoneally three times a week with XMU-MP-1 (2 mg/kg) or a vehicle control for Hippo signaling inhibition. For TNC neutralization, anti-TNC antibody (R&D, MAB2138) or IgG control (1 mg/kg) was administrated locally three times a week.

## Statistics

Data obtained from experiments repeated at least three times was represented as mean ± SEM unless specifically mentioned. Two tailed Student's t test was used to compare two groups. One way ANOVA with Tukey's post hoc test was used to compare differences between multiple groups. Statistical significance was accepted at p < 0.05. All graphs and all statistical tests were generated using Prism V.7 (GraphPad) and SPSS V.21 (IBM).

## Reporting summary

Further information on research design is available in the Nature Portfolio Reporting Summary linked to this article.

# Data availability

The murine ScRNA-seq and RNA-seq generated in this study have been deposited in GSA database under accession code [CRA020509]. The human ScRNA-seq generated in this study have been deposited in GSA for Human database under accession code [HRA009395]. Source data are provided with this paper.

# Code availability

The programming codes in this study are provided in Zenodo (https://doi.org/10.5281/zenodo.14175269).

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

## Acknowledgements
The work was supported by Department of Science and Technology of Guangdong Province (Grant no 2021B1515020080, H.L.), National Natural Science Foundation of China (Grant no 82372370, H.L.; 82172384, H.L.; 81972039, H.L.), Fellowship of China National Post-doctoral Program for Innovative Talents (Grant no BX20240433, S.C.), Innovation Talent Cultivation Program of Sun Yat-sen University (Grant no 22yklj02, H.L.), National Key Research and Development Program of China (Grant no 2022TFC2502901, H.L.; 2022YFC2502902, H.L.; 2022YFC2502903, H.L.; 2022YFC2502904, H.L.) and KELIN New Talent Project of The First Affiliated Hospital, Sun Yat-sen University (Grant no Y12001, H.L.).

## Author contributions
S.C., Y.L., H.Y. contributed equally to this work. H.L. conceived the ideas for experimental designs. S.C., Y.L., H.Y. conducted the majority of the experiments, analysed data, and prepared the manuscript. S.C., ZihaoL, J.Z., S.L., J.W. and X.L. conducted sample collection and performed statistical analysis. X.L., ZeminL., D.C., J.W., and Z.Z. provided critical suggestions and instructions for the project and helped compose the manuscript. S.C., S.L., S.Z. and W.H. provided μCT analysis. S.C., H.C., W.H. and ZihaoL conducted the most animal experiments and performed analysis. H.L. developed the concept, supervised the project, and conducted data analysis.

## Competing interests
The authors declare no competing interests.
