## [Transparent Peer Review file · Nature Communications]

A CD26+ tendon stem progenitor cell population contributes to tendon repair and heterotopic ossification

Corresponding Author: Professor Hui Liu

Version 0:

Reviewer comments:

Reviewer #1

(Remarks to the Author)
NCOMMS-24-17179

Thank you for the opportunity to review the manuscript entitled “A CD26+ tendon stem cell population contributes to tendon repair and heterotopic ossification.” This manuscript describes a newly identified population of progenitor cells within the tendon that give rise to tendon cells in tendon healing, and chondrocytes and osteoblasts in injury-induced heterotopic ossification. The authors support their claims through bioinformatics of large uninjured tendon and injured tendon single-cell RNA-sequencing datasets, histology, in vitro assays, and in vivo models, suggesting that the Cd26+ population of tendon progenitor cells function through TNC and inhibition of YAP signaling to give rise to chondrocytes in HO. The publication provides many novel and impactful findings; however, additional work should be completed prior to publication in Nature Communications. Please find the following major and minor comments below:

Major Comments:

1. Mouse lineage-tracing experiments are used frequently to show co-expression of cell-specific markers and endogenous cre recombinase-induced Tdtomato expression. These findings are critical for the paper. To ensure there is adequate co-expression, higher magnification images should be included to verify clearly that the cells are co-expressing both fluorescent markers.
2. Uninjured tendon images should accompany all lineage-tracing post-injury tendon images. Without side-by-side comparisons it is challenging to determine whether the cells are there at simply there at baseline or repair the wound.
3. Flow cytometry, rather than immunofluorescent staining, should be performed to confirm that neutrophils and monocyte-macrophage populations are not altered at the injection site by diphtheria-toxin induced ablation of Cd26 cells because inflammation is a key process in tendon regeneration after injury.
4. Please clarify lines 320-325 as it is unclear the role of TNC in degradation, phosphorylation, and nuclear translocation of YAP. It is currently written that “TNC also increased phosphorylation of Yap and prevented its nuclear translocation” and “XMU-MP-1, a Hippo/YAP pathway antagonist that decreases the phosphorylation and degradation of YAP, also suppressed the expression of Sox9 and chondrogenic differentiation of CD26+ TSCs.” It is well established in the field that YAP signaling has a pro-osteogenic effect. Is the effect of XMU-MP-1 only anti-chondrogenic? Inhibiting YAP degradation would also have a pro-osteogenic effect, however, there is reduced HO formation. Please provide more discussion on these results.
5. Please show wide-field H&E section of early (~three weeks) and late stage (~nine weeks) heterotopic ossification in your model with side-by-side immunofluorescent histology showing TdTomato+ cells co-expressing chondrogenic and osteogenic differentiation markers to convince readers that these cells are within heterotopic bone.
6. YAP is expressed in many cells at the injury site. To claim that YAP is required for the chondrogenic differentiation of Cd26 cells in vivo requires deletion of Cd26 lineage deletion of at least YAP signaling. To ensure co-activator TAZ is also not at play, dual conditional deletion of YAP and TAZ should be done within Cd26 lineage cells to confirm that Cd26.
7. Monocle trajectory analyses can be misleading if the authors selected the Cd26 high region of the UMAP as the start point, rather than the region of the UMAP that has cells that are truly in the least differentiated state. Please perform CytoTrace analysis on your mesenchymal stromal cell population to validate that the Cd26+ cells are least differentiated and may give rise to the tenogenic and chondrogenic populations.
8. Please merge uninjured and injured tendon single-cell RNA sequencing samples and perform a trajectory analysis to determine the relationship of Cd26 cells before and after injury.

Minor Comments:

1. Please carefully edit the manuscript for spelling and grammatical errors.
2. Please provide additional information about what is known about Dpp4/Cd26. What happens when Dpp4 is deleted from cells?
3. The authors should consider modifying the nomenclature for tendon stem cell to tendon progenitor cell as these cells are lineage-restricted to the mesenchyme in vivo.
4. Larger magnification or tiled images should be included to guide the audience to where in the tendon the image is taken.
5. The authors should quantify Figure 4K to determine the amount of mature tenocytes the Cd26 lineage cells become after injury.
6. Please provide violin plots and gene expression feature plots on UMAP of Cd26 expression across all cell types in both uninjured tendon and injured tendon datasets.
7. The authors may want to adjust the scale for Figure 5L to $-\log(p)=(1,8)$. This would be less confusing to the audience as it appears the top term "Extracellular matrix organization" is not significant since it is blue, not red.

Reviewer #2

(Remarks to the Author)

The authors present a strong paper revealing a new tendon-specific stem cell marked by CD26. A wide range of experiments were performed to characterize these cells in the healthy and healing tendon. The authors should consider expanding parts of their discussion and performing additional experiments, as described below.

1. The authors should add a deeper analysis of the uniqueness of CD26+ TSCs relative to previously identified tendon stem cells. They set this up nicely in the introduction, but they don't explore the comparison in the results or discussion. There are, for example, publicly available scRNAseq data sets that the authors can compare to.
2. Figure 1 nicely presents the existence of 3 clusters of MSCs and 1 cluster of tenocytes. The authors also show convincing data that Cd26 was unique to one of the MSC clusters.
3. Figure 2 includes human scRNAseq data that the authors compare to mouse data. Unlike the mouse data, this data is not very convincing. Are there enough cells to analyze in a meaningful way? The clusters look sparse and overlapping. Panel J does not show evidence of Cd26's uniqueness.
4. a. Figure 3 demonstrates increased proliferative capacity and trilineage differentiation capacity for CD26+ cells. This data strongly supports the stem cell characteristics of the CD26+ MSCs. b. The mouse reporter model is useful for lineage tracing, but the authors must first show the efficiency and specificity of the labeling? c. Panel S3A should be added to this main document figure.
5. a. Figure 4 presents data from the injury model. The authors should justify the relevance of the punch injury, which is not a scenario seen clinically. Would the response be different in a relevant tendon injury model. For example, an Achilles tendon rupture with a repair presents a very different scenario. In the punch defect, a void is filled with infiltrating cells and remodeled. In a typical tendon injury, the torn ends heal back together. b. Panels S5F and S5G should be in main document figure; these panels are important, as they show early expression and expansion of CD26+ cells. c. The paper would be strengthened if the authors added scRNAseq of FACS-sorted CD26+ lineage traced cells from multiple timepoints. This would provide a cleaner isolation and analysis of the cells of interest.
6. a. Figure 5 lacks functional assessment. The authors should add gait and/or tendon biomechanics to assess healing. b. What is the efficiency of DTR for cell ablation? c. The authors cannot claim these cells are "essential" for tendon healing based solely on histologic evidence.
7. Figures 6-7: How clinically relevant is the HO model? This level of ossification not typical in human tendon injury responses.
8. Figure 7: The authors cannot claim "TNC-Hippo pathways dependent", as only descriptive in vivo experiments were performed. Even in vitro, only a limited set of mechanistic experiments were done.
9. Please check for grammar and word choice - there are a number of errors in the paper.

Reviewer #3

(Remarks to the Author)

The manuscript by Chen and colleagues have set out to test the mechanisms underlying tendon regeneration and healing. Setting out from a single cell analysis, they identified a cell population, CD26 positive, which serve as tendon progenitor cells. Using bioinformatics analyses, in vitro differentiation assays, in vivo lineage tracing and ablation assays they show this cell population plays a critical role in tendon healing as well as in heterotopic bone formation, a debilitating process that occurs in multiple cases of tendon healing. The authors finally claim that this heterotopic bone formation and the regulation of the CD26 population is TNC and hippo pathway dependent.

Major Comments

1. Overall, hardly any rationale for carrying out the experiments and methodology is given rendering reading the manuscript

and following the work very hard. For example, they generate the Cd26-CreERT2; hDTR mouse line yet do not show how it was done, what is the definition of DTR. It is also not clear when the toxin was given and when just tamoxifen. Further, in the injury experiment - what age were the mice used? It is only mentioned adult. Also, why 14 dpi were chosen? Likewise, the EdU regime – when and why were the specific times chosen.

2. Throughout the manuscript, no or hardly any quantifications are shown and it is therefore extremely hard to validate the data.
3. The authors fail to describe in detail how they carried out pseudotime trajectory analysis of MSC clusters in their scRNAseq data. As CD26+ are shown to be at the origin of the pseudotime trajectory, was this cluster chosen manually as such or was it chosen by the package's algorithm? If it was chosen manually, what was the rationale behind this choice? Additionally, the authors use Monocle 3 package for their first trajectory analysis, then switch over to slingshot algorithm in Figure 4E - what is the reason for this change of practice?
4. The authors show that ~40% of the CD26 cells are EdU positive. What about the other 60%? What are they and where are they found?
5. In Figure 2C, Cd26 and Lama4 show almost complete overlap in the tendon sheath, however in Figure S2A, significantly less overlap between these stainings is observed. Moreover, signal for both Cd26 and Lama4 is detected in these images in the middle of the tail tendon, which the authors do not address. Please clarify.
6. The authors claim the mouse and human MSC clusters are analogous. They back this claim by presenting expression of very few genes in the human data. Claiming such a claim, requires they show expression of a larger number of genes, or compute the correlation between cluster identities between species. Following such additions, their claim would be much stronger.
7. Quantifications of the differentiation assays shown in Figure 3C must be added. Moreover, from the image it seems as if the Sema5a+ MSC show stronger Alizarin red staining than Cd26+ MSC, which could suggest better differentiation. Results of the quantifications should be addressed.
8. Leinroth et al.,2022 (PMID 35545045) carried out similar experiments with FAPs isolated from skeletal muscles, where they demonstrated higher adipogenic potential for Cd26+ FAPs, but low osteogenic potential. As connective tissue lineage plasticity has been a subject of extensive research, the authors should address Leinroth et al.,2022 findings and the differentiation potential of FAP subpopulations compared to their tendon MSC?
9. Along these lines, the authors claim also in the Abstract, that the CD26 population is a novel population. However, this population has been shown in multiple other settings therefore the authors should rephrase their claim.
10. In Figure 3D, the authors describe an in-vitro differentiation experiment of MSC subpopulation identities, using only two markers (one for each subpopulation). This claim is not supported by sufficient data. Additional markers for each subpopulation (e.g., using FACS or RT-PCR for example) is required to claim this change of identity, especially when using ex-vivo assays where the cells are not in their natural environment.
11. The section from line 186 to line 197 is very poorly written. It has insufficient background and requires citation and references to justify the genes used by the authors for their claim. Labeling of the figures should be more clear to enable the reader understand when were human or mouse cells used, and what subpopulation of MSC was examined.
12. Throughout the manuscript, the authors hardly describe what were ages the mice used and why were these ages chosen.
13. In Figure 4, the Apoe+ cluster seems significantly bigger than in the previous scRNAseq data presented. Could the authors address what were the differences between experiments?
14. Figure S6D- stiffness and modulus are poorly explained both in meaning and methods.
15. The authors need to better introduce the process of HO as part of tendon healing. Is HO a naturally occurring process in healthy tendon healing? Is it always pathological? How long does it take for the HO to disappear from the tendon in normal healing?
16. In Figure S7D-H, why did the authors focus on chondrogenic and not osteogenic differentiation, given they describe an ossification process? Further, could the authors repeat these experiments with Sema5a+ MSC, given they did show differentiation potential to both relevant lineages?
17. TNC is a highly expressed gene in tendons, and specifically along the muscle tendon junction. Complete knockout of TNC is not a minor mutation and could have widespread effects. The authors should provide description of the mutant mice, specifically tendon and muscle tendon junction development, and describe the tendon regeneration process regardless of the CD26 population in these mutants compared to wild-type mice?
18. Does YAP inhibition improve tendon healing or just eliminates HO? Comparison of the regeneration and healing process in the inhibited mice and control wild-type should be carried out.
19. Why were TNC and Hippo pathway chosen? Clearly, the KEGG pathway analysis shown other pathways which are more significant.
20. The authors should address what happens to the CD26+ cells in the TNC KO and following Hippo-pathway inhibition? Is it only their osteogenic differentiation that is effected? Are they proliferating at the same rate? Are they invading the tendon midsubstance at the same rate?
21. As shown now, with the lack of explanation and insufficient data, the data in Figure 7 (TNC and hippo pathway) seem irrelevant for this manuscript and could fit in a separate manuscript.
22. Please expand the discussion on the role of the Sema5a+ and Apoe+ MSC clusters. Are they transient or permanent cell states found in the tissue? What are their biological roles? Do they differentiate into other lineages? Based on what exactly did the authors term Apoe+ cells as intermediate progenitors?
23. The authors mention generating several mice lines for this study, however there is no description of the generation process, or references to other papers detailing the generation and relevant phenotypes of these mutants.
24. In Figure S6I, the comparison between PBS and DTX groups is not reliable due to the wide spread of the control group compared to the narrow spread of the DTX group. The authors would need to increase the n in these experiments to make them reliable. Also, this experiment is not detailed in the methods at all, the authors need to detail when and from where was the blood collected, along with other experimental procedures.

25. To get a sense of the healing properties of the tendon, images of a non-injured control next to the images and quantifications in Fig. 5 should be added.

Minor comments

1. Fig.1 A-B show 14 clusters, line 108 in the text describes 13 clusters.
2. From line 113 to line 120, the authors detail highly expressed marker genes for three MSC subpopulations in the scRNAseq dataset. They reference Figure S2A-B, which relates to the tail tendon structure and a tSNE map of its scRNAseq. Could it be possible the authors meant to refer to Figure S1B-D instead? (as these figures show expression of the genes discussed in the text)
3. In Figure 3A, the authors use the Ly6a gene name in the FACS plot. Ly6a encodes for the Sca1 protein, which would be more appropriate to use in this context.
4. In Figure 3F, the framing of the tendon regions do not seem consistent (comparing spread of cell nuclei). Additionally, adding labels for the tissue regions would greatly help understanding the images.
5. The slingshot algorithm is missing entirely from methods section.
6. In line 238, the authors refer to Figure S5A-C when discussing gene expression over pseudotime, however this figure deals with GO analysis. Could it be possible the authors meant to refer to Figure S6A-C, which seems more fitting?
7. Lines 249-250 are incoherent and their meaning unclear.
8. The authors did not specify in the differentiation assay what was the cell concentration used (cells per well), nor what was the medium used before switching to the differentiation mediums.
9. The authors did not specify what medium they were using for the proliferation assay, or in what dish they cultured the cells (what was the cell concentration).
10. In the scRNAseq section in the methods, did the authors control for mitochondrial genes, nCount or nFeatures data to eliminate doublets or low quality cells as commonly used pipelines instructs? If not, what was their QC strategy for these issues?
11. In the tendon punch experiment, the authors did not specify what package or algorithm were used to integrate the two datasets for analysis.
12. The authors do not detail how they carried out the KEGG pathway analysis.
13. The authors refer a couple of times to Figure 8 however this MS has only 7 figures.
14. Lower magnifications of immunostained regions should be shown in order to better understand what regions were these images taken from (in a like manner to that shown in Fig. 2L).
15. It is not clear why immune CD26+ cells are not ablated in the DTR experiment. Please explain.
16. In the figures describing the Cre construct, instead of PolyA it is written as PlolyA

Reviewer #4

(Remarks to the Author)

Version 1:

Reviewer comments:

Reviewer #1

(Remarks to the Author)

Thank you for the opportunity to review the manuscript entitled "A CD26+ tendon stem cell population contributes to tendon repair and heterotopic ossification." The authors have adequately addressed all comments raised in the first round of reviews with additional experiments and explanations. Including additional images, single-cell RNA-sequencing analyses, and quantifications have strengthened their claims. A minor addition should be included in the manuscript to solidify the single-cell RNA-sequencing results. Please address the comment below before publication:

1. The CytoTrace analysis supports your findings by showing high CytoTrace scores in the "Cd26+" group. It would be even more convincing to show a feature plot (expression overlaid on dimensional reduction) of the Cd26 expression on the CytoTrace UMAP, to correlate the regions of the UMAP with high CytoTrace scores to potentially high expression of Cd26.

Reviewer #2

(Remarks to the Author)

The authors have addressed all comments.

Reviewer #3

(Remarks to the Author)

The authors have nicely carried out the majority of the concerns however a few have remained that need corrections.
Previous main comment 20: The authors claim that the CD26+ cells have proliferated and invaded the tendon midsubstance to the same extent in the TNC KO. While proliferation is shown in vitro, the fact that a similar number of cells is observed in the tissue does not conclude that invasion was not affected. In order to claim this, lineage analysis must be done. It should either be carried out or simply to modify the claim.

Previous minor comment 9: The authors claim they plated 1000 cells per 10 cm tissue culture plate. This does not seem realistic. Could it be a typo?

Previous minor comment 11: the methods used by the authors to integrate datasets seem inadequate to the commonly used pipelines. Seurat offers specific integration tools for this purpose, and not just by merging datasets.

Reviewer #4

(Remarks to the Author)

Version 2:

Reviewer comments:

Reviewer #3

(Remarks to the Author)

The authors satisfactorily addressed our concerns.

Reviewer #4

(Remarks to the Author)

REVIEWER COMMENTS

Reviewer #1 (Remarks to the Author):

NCOMMS-24-17179

Thank you for the opportunity to review the manuscript entitled “A CD26+ tendon stem cell population contributes to tendon repair and heterotopic ossification.” This manuscript describes a newly identified population of progenitor cells within the tendon that give rise to tendon cells in tendon healing, and chondrocytes and osteoblasts in injury-induced heterotopic ossification. The authors support their claims through bioinformatics of large uninjured tendon and injured tendon single-cell RNA-

sequencing datasets, histology, in vitro assays, and in vivo models, suggesting that the Cd26+ population of tendon progenitor cells function through TNC and inhibition of YAP signaling to give rise to chondrocytes in HO. The publication provides many novel and impactful findings; however, additional work should be completed prior to publication in Nature Communications. Please find the following major and minor comments below:

Major Comments:

1. Mouse lineage-tracing experiments are used frequently to show co-expression of cell-specific markers and endogenous cre recombinase-induced Tdtomato expression. These findings are critical for the paper. To ensure there is adequate co-expression, higher magnification images should be included to verify clearly that the cells are co-expressing both fluorescent markers.

Response:

Thank you for your thorough review and constructive comments. We have now included higher magnification images of immunofluorescent staining in Figure 4J, 4K, 6D, and 6F to demonstrate the co-expression of cell-specific markers (Scx, Sp7, Sox9) and endogenous cre recombinase-induced Tdtomato expression.

Figure 4J. Immunofluorescent staining of Scx and tdTomato (tdT) in tendon from Sham, 15 and 30 dpi group. n=5. Scale bar: 100 μ m.

Figure 4K. Immunofluorescent staining of Fmod and tdTomato (tdT) in tendon from Sham and 30 dpi group. n=5. Scale bar: 100 μ m.

Figure 6D. SOFG staining and immunofluorescent staining of Sox9 and tdTomato (tdT) of Achilles tendon from Sham, 30 dpi and 60 dpi group. n=5. Scale bar: 100 μ m.

Figure 6F. SOFG staining and immunofluorescent staining of Sp7 and tdTomato (tdT) of Achilles tendon from Sham, 30 dpi and 60 dpi group. n=5. Scale bar: 100 μ m.

2. Uninjured tendon images should accompany all lineage-tracing post-injury tendon images. Without side-by-side comparisons it is challenging to determine whether the cells are there at simply there at baseline or repair the wound.

Response:

Thank you for your careful review and constructive comments. To provide a comprehensive analysis, we have now included uninjured tendon images in Figure 4J, 4K, 6D, and 6F to accompany all lineage-tracing data.

post-injury tendon images.

Figure 4J. Immunofluorescent staining of Scx and tdTomato (tdT) in tendon from Sham, 15 and 30 dpi group. n=5. Scale bar: 100 μ m.

Figure 4K. Immunofluorescent staining of Fmod and tdTomato (tdT) in tendon from Sham and 30 dpi group. n=5. Scale bar: 100 μ m.

Figure 6D. SOFG staining and immunofluorescent staining of Sox9 and tdTomato (tdT) of Achilles

tendon from Sham, 30 dpi and 60 dpi group. n=5. Scale bar: 100µm.

Figure 6F. SOFG staining and immunofluorescent staining of Sp7 and tdTomato (tdT) of Achilles tendon from Sham, 30 dpi and 60 dpi group. n=5. Scale bar: 100µm.

3. Flow cytometry, rather than immunofluorescent staining, should be performed to confirm that neutrophils and monocyte-macrophage populations are not altered at the injection site by diphtheria-toxin induced ablation of Cd26 cells because inflammation is a key process in tendon regeneration after injury.

Response:

Thank you for your careful review and constructive comments. It has been reported that myeloid cells, including neutrophils and monocyte-macrophages, participate in the inflammatory process after tendon injury. To validate whether myeloid cells are not

altered after diphtheria-toxin-induced ablation of CD26+ cells, tendon tissues were collected from both the vehicle and diphtheria-toxin groups at day 1 after tendon injury, and flow cytometry analysis was performed. The results showed that the number of CD45+CD11b+ myeloid cells was not altered after diphtheria-toxin-induced ablation of CD26+ cells (Figure S6E-G). Furthermore, our scRNA-seq data revealed that only 5.19% of monocyte-macrophages and few neutrophils expressed CD26 (Figure R1), which may explain why diphtheria-toxin-induced ablation of CD26+ cells did not alter the populations of neutrophils and monocyte-macrophages.

Figure S6. E, Flow cytometry analysis of CD45+CD11b+ myeloid cells in vehicle and diphtheria-toxin group. n=3. F-G. Quantification of CD45+ immune cells and CD45+CD11b+ myeloid cells in vehicle and diphtheria-toxin group.

Figure R1. A. UMAP plot and keys to the clustering numbers and cell types are shown to the right.

B. UMP plot of expression of CD26. C. Bubble plot of expression of CD26.

4. Please clarify lines 320-325 as it is unclear the role of TNC in degradation, phosphorylation, and nuclear translocation of YAP. It is currently written that “TNC also increased phosphorylation of Yap and prevented its nuclear translocation” and “XMU-MP-1, a Hippo/YAP pathway antagonist that decreases the phosphorylation and degradation of YAP, also suppressed the expression of Sox9 and chondrogenic differentiation of CD26+ TSCs.” It is well established in the field that YAP signaling has a pro-osteogenic effect. Is the effect of XMU-MP-1 only anti-chondrogenic? Inhibiting YAP degradation would also have a pro-osteogenic effect, however, there is reduced HO formation. Please provide more discussion on these results.

Response:

Thank you for your careful review and constructive comments. Most cases of injury-induced heterotopic ossification (HO) develop through endochondral ossification [1]. This process involves several stages: chondrogenic differentiation of mesenchymal stromal cells, cartilage template formation, hypertrophy of chondrocytes, vascular invasion, and subsequent bone formation. Initially, cells in the center of mesenchymal condensations differentiate into chondrocytes, which begin to produce a matrix composed of collagen and proteoglycan, forming the cartilage template. Subsequently, the cells defining the border withdraw from the cell cycle and undergo hypertrophy, prompting the differentiation of osteoblasts from the perichondrium. Lastly, blood vessels infiltrate the hypertrophic cartilage, triggering osteoblast differentiation and eventual formation of the bone marrow cavity [2]. Indeed, previous studies have shown that the role of Yap signaling in chondrogenesis and osteogenesis is opposite. In chondrogenesis, phosphorylation of Yap leads to its degradation and promotes chondrogenic differentiation, while Yap nuclear translocation promotes osteogenesis [3,4]. Huber et al. demonstrated that knockdown of Yap could suppress osteogenic differentiation of mesenchymal progenitor cells and subsequent HO formation, which is consistent with previous findings that Yap is essential for osteogenesis [5]. In the current study, we found that TNC aberrantly expressed at the stage of cartilage template formation and promotes chondrogenesis through promoting the phosphorylation of Yap (Figure R2). Since chondrogenesis and cartilage template formation occur at the initial stage of HO formation, inhibiting Yap degradation at this stage mainly suppresses the process of chondrogenesis, thereby alleviating HO derived from endochondral

ossification. Nonetheless, at the stage of bone formation, TNC was not aberrantly expressed and therefore did not regulate the Hippo pathway (Figure R2). Both Huber's and our studies demonstrate the different roles of Yap in endochondral ossification, which is consistent with previous studies.

The above information has been added in the revised manuscript [Page 27-28, Line 532-561].

Figure R2. Histological analysis and quantification of TNC in Sham, 30 dpi and 90 dpi group. C: cartilage formation site; B: Bone. n=5. Scale bar: 100 μ m.

References:

1. Zhang Q, Zhou D, Wang H, Tan J. Heterotopic ossification of tendon and ligament. *J Cell Mol Med.* 2020 May;24(10):5428-5437. doi: 10.1111/jcmm.15240. Epub 2020 Apr 15. PMID: 32293797; PMCID: PMC7214162.
2. Salhotra A, Shah HN, Levi B, Longaker MT. Mechanisms of bone development and repair. *Nat Rev Mol Cell Biol.* 2020 Nov;21(11):696-711. doi: 10.1038/s41580-020-00279-w. Epub 2020 Sep 8. PMID: 32901139; PMCID: PMC7699981.
3. Pan JX, Xiong L, Zhao K, Zeng P, Wang B, Tang FL, Sun D, Guo HH, Yang X, Cui S, Xia WF, Mei L, Xiong WC. YAP promotes osteogenesis and suppresses adipogenic differentiation by regulating β -catenin signaling. *Bone Res.* 2018 Jun 1;6:18.

4. Deng Y, Wu A, Li P, Li G, Qin L, Song H, Mak KK. Yap1 Regulates Multiple Steps of Chondrocyte Differentiation during Skeletal Development and Bone Repair. *Cell Rep.* 2016 Mar 8;14(9):2224-2237.

5. Huber AK, Patel N, Pagani CA, Marini S, Padmanabhan KR, Matera DL, Said M, Hwang C, Hsu GC, Poli AA, Strong AL, Visser ND, Greenstein JA, Nelson R, Li S, Longaker MT, Tang Y, Weiss SJ, Baker BM, James AW, Levi B. Immobilization after injury alters extracellular matrix and stem cell fate. *J Clin Invest.* 2020 Oct 1;130(10):5444-5460.

5. Please show wide-field H&E section of early (~three weeks) and late stage (~nine weeks) heterotopic ossification in your model with side-by-side immunofluorescent histology showing TdTomato+ cells co-expressing chondrogenic and osteogenic differentiation markers to convince readers that these cells are within heterotopic bone.

Response:

Thank you for your careful review and constructive comments. To validate the presence of TdTomato+ cells within heterotopic bone, we have included wide-field Safranin O-Fast Green staining sections in the revised manuscript. This staining method, commonly used in histology, visualizes the presence and distribution of proteoglycans and other matrix components within tissues, particularly in bone and cartilage. The results demonstrated that at 60 days post-injury (dpi), when heterotopic bone was observed, tdT+Sox9+ chondrocytes were found near the heterotopic bone, and tdT+Sp7+ osteoblasts were resident within the heterotopic bone (Figure 6D and 6F).

Figure 6D. SOFG staining and immunofluorescent staining of Sox9 and tdTomato (tdT) of Achilles tendon from Sham, 30 dpi and 60dpi group. n=5. Scale bar: 100 μ m.

Figure 6F. SOFG staining and immunofluorescent staining of Sp7 and tdTomato (tdT) of Achilles

tendon from Sham, 30 dpi and 60dpi group. n=5. Scale bar: 100µm.

6. YAP is expressed in many cells at the injury site. To claim that YAP is required for the chondrogenic differentiation of Cd26 cells in vivo requires deletion of Cd26 lineage deletion of at least YAP signaling. To ensure co-activator TAZ is also not at play, dual conditional deletion of YAP and TAZ should be done within Cd26 lineage cells to confirm that Cd26.

Response:

Thank you for your careful review and constructive comments. YAP signaling is essential for cell proliferation and differentiation. In the current study, we found that depletion of CD26⁺ cells led to reduced HO formation, indicating that these cells are progenitors of ectopic bone (Figure 7B-C). In addition, we observed that during the process of tendon healing, about 90.00% at 30 dpi and 86.54% at 60 dpi of p-YAP⁺ cells were CD26⁺ cells, indicating that inhibition of Hippo pathway mainly affects the CD26⁺ cell population at these stages (Figure S7A-B).

Indeed, deletion of YAP and TAZ in CD26 lineage cells would provide a more rigorous validation of our hypothesis in vivo. Nonetheless, due to budgetary and time limitations, we are currently unable to performed the experiment. The above information has been added in the limitation section. Therefore, we have rephrased the overstatement in the revised manuscript [Page 3, Line 36-37; Page 17, Line 330; Page 20, Line 383-384].

This limitation has been added in the discussion section of the revised manuscript.

Figure 7. B-C, Micro-CT images and quantification of ectopic bone in Achilles tendons from diphtheria toxin (DTx) and vehicle control (Veh) group. n=5 per group. Scale bar: 1mm.

Figure S7. A-B Immunofluorescent staining and quantification of CD26 and p-YAP of Achilles tendon from Sham, 30 dpi and 60dpi group. n=3. One way ANOVA with Turkey's test. Scale bar: 100 μ m. Data shown are mean \pm SEM.

7. Monocle trajectory analyses can be misleading if the authors selected the Cd26 high region of the UMAP as the start point, rather than the region of the UMAP that has cells that are truly in the least differentiated state. Please perform CytoTrace analysis on your mesenchymal stromal cell population to validate that the Cd26+ cells are least differentiated and may give rise to the tenogenic and chondrogenic populations.

Response:

Thank you for your careful review and constructive comments. To objectively select the start point of the pseudotime analysis, we conducted CytoTRACE analysis on CD26+ TSPCs, Apoe+ progenitors, Sema5a+ progenitors, and tenocytes from the uninjured group. The results revealed that CD26+ tendon progenitor cells exhibited the highest CytoTRACE score, indicating that they were the least differentiated among these populations (Figure 1E). [Page 9, Line 149-153]

Additionally, we performed CytoTRACE analysis on CD26+ TSPCs, Apoe+ progenitors, Sema5a+ progenitors, chondrocytes/tenocytes, tenocytes, chondrocytes from a merged group of uninjured and injured samples. The results consistently showed that CD26+ TSPCs presented the highest CytoTRACE score, indicating that they were the least differentiated among these populations (Figure 4B, G and S5J). [Page 14, Line 262-268]

Figure 1. E, CytoTRACE analysis of CD26⁺ TSPCs, Apoe⁺ progenitors and Sema5a⁺ progenitors from the uninjured group.

Figure 4B. Umap plot of CD26⁺ TSPCs, Apoe⁺ progenitors, Sema5a⁺ progenitors, Chondrocyte/Tenocyte, Tenocytes and Chondrocyte from the merged group. G, Cytotrace analysis on CD26⁺ TSPCs, Apoe⁺ progenitors, Sema5a⁺ progenitors, Chondrocyte/Tenocyte, Tenocytes and Chondrocyte from the merged group.

Figure S5J. Cytotrace score among CD26⁺ TSPCs, Apoe⁺ progenitors, Sema5a⁺ progenitors, Chondrocyte/Tenocyte, Tenocytes and Chondrocyte

8. Please merge uninjured and injured tendon single-cell RNA sequencing samples and

perform a trajectory analysis to determine the relationship of Cd26 cells before and after injury.

Response:

Thank you for your careful review and constructive comments. Firstly, we conducted CytoTRACE analysis on CD26⁺ TSPCs, Apoe⁺ progenitors, Sema5a⁺ progenitors, chondrocytes/tenocytes, tenocytes, chondrocytes from the merged uninjured and injured group. The results indicated that CD26⁺ tendon progenitor cells exhibited the highest CytoTRACE score, suggesting that they were the least differentiated among these populations (Figure 4B and S5I-J). Furthermore, pseudotime trajectory analysis of clusters CD26⁺ TSPC, Apoe⁺ progenitors, Sema5a⁺ progenitors, tenocytes, and chondrocytes/tenocyte and chondrocyte in the merged uninjured and injured group with Monocle revealed that CD26⁺ TSPC was the beginning of the trajectory and might differentiate into tenocytes and chondrocytes after tendon injury (Figure S5F)

[Page 14, Line 262-268].

Figure 4B. Umap plot of CD26⁺ TSPCs, Apoe⁺ proenitors, Sema5a⁺ progenitors, Chondrocyte/Tenocyte, Tenocytes and Chondrocyte from the merged group. G, CytoTRACE analysis on CD26⁺ TSPCs, Apoe⁺ proenitors, Sema5a⁺ progenitors, Chondrocyte/Tenocyte, Tenocytes and Chondrocyte from the merged group.

Figure S5J. Cytotrace score among CD26⁺ TSPCs, Apoe⁺ progenitors, Sema5a⁺ progenitors, Chondrocyte/Tenocyte, Tenocytes and Chondrocyte

Figure S5F. UMAP plot of pseudotime analysis of cluster CD26⁺ TSPC, Apoe⁺ progenitor, Sema5a⁺ progenitor, Chondrocyte/Tenocyte, Tenocytes and Chondrocyte from uninjured and punched group.

Minor Comments:

1. Please carefully edit the manuscript for spelling and grammatical errors.

Response:

Thank you for your careful review and constructive comments. The spelling and grammatical errors have been corrected by native English speakers to ensure clarity and accuracy.

2. Please provide additional information about what is known about Dpp4/Cd26. What happens when Dpp4 is deleted from cells?

Response:

CD26/DPP4 is expressed as a type II transmembrane protein, featuring a short six

amino acid cytoplasmic tail [1]. It functions as a dimer with a monomer molecular weight of 110 kDa. DPP4 exerts its physiological roles through both its enzymatic activity, regulating numerous peptides, and its interactions with a diverse range of binding partners [2]. DPP4 is involved in various processes, including nutrition, nociception, cell-adhesion, psychoneuroendocrine regulation, immune response, and cardiovascular adaptation [3]. [Page 7-8, Line 124-128]

Furthermore, DPP4 deficiency has been associated with improved glucose tolerance, increased susceptibility to infection, and altered immune response in inflammatory diseases [3, 4].

References:

1. Nieto-Fontarigo JJ, González-Barcala FJ, San José E, Arias P, Nogueira M, Salgado FJ. CD26 and Asthma: a Comprehensive Review. *Clin Rev Allergy Immunol.* 2019 Apr;56(2):139-160. doi: 10.1007/s12016-016-8578-z. PMID: 27561663
2. Matteucci E, Giampietro O. Dipeptidyl peptidase-4 (CD26): knowing the function before inhibiting the enzyme. *Curr Med Chem.* 2009;16(23):2943-51. doi: 10.2174/092986709788803114. PMID: 19689275
3. Klemann C, Wagner L, Stephan M, von Hörsten S. Cut to the chase: a review of CD26/dipeptidyl peptidase-4's (DPP4) entanglement in the immune system. *Clin Exp Immunol.* 2016 Jul;185(1):1-21. doi: 10.1111/cei.12781. Epub 2016 May 13. PMID: 26919392
4. Marguet D, Baggio L, Kobayashi T, Bernard AM, Pierres M, Nielsen PF, Ribel U, Watanabe T, Drucker DJ, Wagtmann N. Enhanced insulin secretion and improved glucose tolerance in mice lacking CD26. *Proc Natl Acad Sci U S A.* 2000 Jun 6;97(12):6874-9. doi: 10.1073/pnas.120069197.

PMID: 10823914

3. The authors should consider modifying the nomenclature for tendon stem cell to tendon progenitor cell as these cells are lineage-restricted to the mesenchyme in vivo.

Response:

Thank you for your careful review and constructive comments. Stem cells are currently defined as single cells that are clonal precursors of both more stem cells of the same type, as well as a defined set of differentiated progenies [1]. The definition of tendon stem cell and tendon progenitor cell is under debated. Harvey et al. identified Tppp3+Pdgfra+ tendon stem cells which are capable of osteogenic and chondrogenic differentiation, as well as developing into tenocytes during tendon healing [2]. While other studies demonstrate the cells with tenogenic potential as tendon stem/progenitor cells [2-4]. Nathaniel et al. identified a population of SMA+ tendon progenitor cells that contribute to tendon healing [5]. To ensure consistency with descriptions in the majority of studies, we decided to modified the nomenclature for tendon stem cell to tendon stem/progenitor cell (TSPC).

References:

1. Weissman IL, Anderson DJ, Gage F. Stem and progenitor cells: origins, phenotypes, lineage commitments, and transdifferentiations. *Annu Rev Cell Dev Biol.* 2001;17:387-403. doi: 10.1146/annurev.cellbio.17.1.387. PMID: 11687494.
2. Harvey T, Flamenco S, Fan CM. A Tppp3⁺Pdgfra⁺ tendon stem cell population contributes to regeneration and reveals a shared role for PDGF signalling in regeneration and fibrosis. *Nat Cell Biol.* 2019 Dec;21(12):1490-1503. doi: 10.1038/s41556-019-0417-z. Epub 2019 Nov 25. PMID:

31768046.

2. Bi Y, Ehrchiou D, Kilts TM, Inkson CA, Embree MC, Sonoyama W, Li L, Leet AI, Seo BM, Zhang L, Shi S, Young MF. Identification of tendon stem/progenitor cells and the role of the extracellular matrix in their niche. *Nat Med.* 2007 Oct;13(10):1219-27. doi: 10.1038/nm1630. Epub 2007 Sep 9. PMID: 17828274.3.

3. Yin Z, Hu JJ, Yang L, Zheng ZF, An CR, Wu BB, Zhang C, Shen WL, Liu HH, Chen JL, Heng BC, Guo GJ, Chen X, Ouyang HW. Single-cell analysis reveals a nestin⁺ tendon stem/progenitor cell population with strong tenogenic potentiality. *Sci Adv.* 2016 Nov 18;2(11):e1600874. doi: 10.1126/sciadv.1600874. PMID: 28138519; PMCID: PMC5262457.

4. Lee CH, Lee FY, Tarafder S, Kao K, Jun Y, Yang G, Mao JJ. Harnessing endogenous stem/progenitor cells for tendon regeneration. *J Clin Invest.* 2015 Jul 1;125(7):2690-701. doi: 10.1172/JCI81589. Epub 2015 Jun 8. PMID: 26053662; PMCID: PMC4563693.

5. Dymant NA, Hagiwara Y, Matthews BG, Li Y, Kalajzic I, Rowe DW. Lineage tracing of resident tendon progenitor cells during growth and natural healing. *PLoS One.* 2014 Apr 23;9(4):e96113. doi: 10.1371/journal.pone.0096113. PMID: 24759953; PMCID: PMC3997569.

4. Larger magnification or tiled images should be included to guide the audience to where in the tendon the image is taken.

Response:

Thank you for your careful review and constructive comments. To provide better guidance for the audience, larger images of histological staining have been added to indicate the specific locations within the tendon where the images in Figure 2C, Figure

6D, and Figure 6F were captured.

Figure 2C. SOFG staining and immunofluorescent staining of CD26 and Lama4 of of Achilles tendons. n=3. Scale bar: 100 μ m.

Figure 6D. SOFG staining and immunofluorescent staining of Sox9 and tdTomato (tdT) of Achilles tendon from Sham, 30 dpi and 60dpi group. n=5. Scale bar: 100 μ m.

Figure 6F. SOFG staining and immunofluorescent staining of Sp7 and tdTomato (tdT) of Achilles tendon from Sham, 30 dpi and 60dpi group. n=5. Scale bar: 100 μ m.

5. The authors should quantify Figure 4K to determine the amount of mature tenocytes the Cd26 lineage cells become after injury.

Response:

Thank you for your careful review and constructive comments. We have performed the quantification of Fmod+tdT+ cells, which indicate mature tenocytes derived from CD26 lineage cells, as per your suggestion (Figure 4K-L).

Figure 4K-L. Immunofluorescent staining and quantification of Fmod and tdTomato (tdT) in tendon from Sham and 30 dpi group. n=5. Scale bar: 100 μ m.

6. Please provide violin plots and gene expression feature plots on UMAP of Cd26 expression across all cell types in both uninjured tendon and injured tendon datasets.

Response:

Thank you for your careful review and constructive comments. Bubble plots and gene expression feature plots on UMAP of Cd26 expression across all cell types in the combined uninjured and injured tendon datasets have been generated to better illustrate the expression of CD26 across all cell types. The results indicate that CD26 is primarily expressed in the MSC cluster.

Figure R1 A. UMAP plot and keys to the clustering numbers and cell types are shown to the right.

B. UMAP plot of expression of CD26. C. Bubble plot of expression of CD26.

7. The authors may want to adjust the scale for Figure 5L to $-\log(p) = (1,8)$. This would be less confusing to the audience as it appears the top term “Extracellular matrix organization” is not significant since it is blue, not red.

Response:

Thank you for your careful review and constructive comments. We have adjusted the scale for Figure 7L to $-\log(p) = (1,10)$ to provide a clearer illustration of the significance of Extracellular matrix organization ($p = 0.0094$).

Figure 7L. GO pathway analysis of upregulated genes in punched group compared to sham control.

Reviewer #2 (Remarks to the Author):

The authors present a strong paper revealing a new tendon-specific stem cell marked by CD26. A wide range of experiments were performed to characterize these cells in the healthy and healing tendon. The authors should consider expanding parts of their discussion and performing additional experiments, as described below.

1. The authors should add a deeper analysis of the uniqueness of CD26+ TSCs relative to previously identified tendon stem cells. They set this up nicely in the introduction, but they don't explore the comparison in the results or discussion. There are, for example, publicly available scRNAseq data sets that the authors can compare to.

Response:

Thank you for your careful review and constructive comments. To investigate the

uniqueness of CD26⁺ TDPCs relative to previously identified tendon stem/progenitor cells, we detected the expression levels of signature genes of Tppp3⁺Pdgfra⁺ TSC [1], classic TSPC [2], Nestin⁺ TSPC [3], OCN⁺ TSPC [4], and α SMA⁺ TSPC [5] in CD26⁺ TSPCs, Apoe⁺ progenitors, Sema5a⁺ progenitors, and Tenocytes from the scRNA dataset. The results showed that CD26⁺ TPCs were enriched in signature genes of Tppp3⁺Pdgfra⁺ TSC (Tppp3, Pdgfra, Ly6e, Plin2, Lama4) and classic TSPC (Ly6a, Cd44, and Cd90, but not Scx and Comp). However, approximately 25.6% of CD26⁺ TSPCs cluster did not express Tppp3, indicating that CD26⁺ TSPCs were not identical to Tppp3⁺Pdgfra⁺ TSC. In addition, marker genes of Nestin⁺ TDPC (Nestin and Mcam), OCN⁺ TDPC (Bglap), and α SMA⁺ TSC (Acta2) were not highly expressed in CD26⁺ TPCs, suggesting that CD26⁺ TPCs were distinct from Nestin⁺ TDPCs, OCN⁺ TDPCs, and α SMA⁺ TSCs (Figure S2A). [Page 8, Line 135-146]

Figure S2A violin plot and UMAP plot of signature gene expression level of Tpp3+Pdgfra+ TSC, classic TSPC, Nestin+ TSPC, OCN+ TSPC and αSMA+ TSPC in CD26+ tendon progenitor cell (TPC), Apoe+ TDPC, Sema5a+ progenitors and Tenocyte from the scRNA dataset.

Reference:

- 1 Harvey T, Flamenco S, Fan CM. A Tpp3⁺Pdgfra⁺ tendon stem cell population contributes to regeneration and reveals a shared role for PDGF signalling in regeneration and fibrosis. *Nat Cell Biol.* 2019 Dec;21(12):1490-1503. doi: 10.1038/s41556-019-0417-z. Epub 2019 Nov 25. PMID: 31768046.
2. Bi Y, Ehrchiou D, Kilts TM, Inkson CA, Embree MC, Sonoyama W, Li L, Leet AI, Seo BM, Zhang L, Shi S, Young MF. Identification of tendon stem/progenitor cells and

the role of the extracellular matrix in their niche. *Nat Med.* 2007 Oct;13(10):1219-27. doi: 10.1038/nm1630. Epub 2007 Sep 9. PMID: 17828274.3. Yin Z, Hu JJ, Yang L, Zheng ZF, An CR, Wu BB, Zhang C, Shen WL, Liu HH, Chen JL, Heng BC, Guo GJ, Chen X, Ouyang HW. Single-cell analysis reveals a nestin⁺ tendon stem/progenitor cell population with strong tenogenic potentiality. *Sci Adv.* 2016 Nov 18;2(11):e1600874. doi: 10.1126/sciadv.1600874. PMID: 28138519; PMCID: PMC5262457.

4. Wang Y, Zhang X, Huang H, Xia Y, Yao Y, Mak AF, Yung PS, Chan KM, Wang L, Zhang C, Huang Y, Mak KK. Osteocalcin expressing cells from tendon sheaths in mice contribute to tendon repair by activating Hedgehog signaling. *Elife.* 2017 Dec 15;6:e30474. doi: 10.7554/eLife.30474. PMID: 29244023; PMCID: PMC5731821.

5. Dyment NA, Hagiwara Y, Matthews BG, Li Y, Kalajzic I, Rowe DW. Lineage tracing of resident tendon progenitor cells during growth and natural healing. *PLoS One.* 2014 Apr 23;9(4):e96113. doi: 10.1371/journal.pone.0096113. PMID: 24759953; PMCID: PMC3997569.

2. Figure 1 nicely presents the existence of 3 clusters of MSCs and 1 cluster of tenocytes. The authors also show convincing data that Cd26 was unique to one of the MSC clusters.

Response:

Thank you for your careful review and constructive comments.

3. Figure 2 includes human scRNAseq data that the authors compare to mouse data.

Unlike the mouse data, this data is not very convincing. Are there enough cells to

analyze in a meaningful way? The clusters look sparse and overlapping. Panel J does not show evidence of Cd26's uniqueness.

Response:

Thank you for your careful review and constructive comments. To confirm the uniqueness of CD26⁺ TSPC in human scRNAseq data, we included another human sample in the scRNAseq data and re-performed the analysis. The result showed that cluster MSC-1 expressed high level of Dpp4, CD55, Pi16, CD248 and Sema3c, which was also highly expressed in CD26⁺ TSPC in mice scRNAseq data (Figure 2I--K).

However, although we found another two populations of MSC in human dataset (MSC-2 and MSC-3), MSC-2 and MSC-3 showed few expressions of markers of Apoe⁺ progenitors and Sema5a⁺ progenitors. Due to the differences between human and murine species, the signature genes of these two cell populations may exhibit disparities. We could not identify analogues populations of Apoe⁺ progenitors and Sema5a⁺ progenitors in human scRNA-seq dataset. In human, MSC-2 is enriched in expression of Apoe, Sfrp1, Vcam1 and Pla2g2a, indicating that these cells were associated with lipid metabolism, which were partially similar to Apoe⁺ progeniros found in murine scRNA-seq dataset. As for MSC-3, Lcp1, Lyz, s100a12 and LST1 were enriched, indicating that these cells were associated with immune response. Nonetheless, we found that the signature genes of CD26⁺ TSPCs were analogous in both murine and human scRNA-seq datasets. The identification of analogues populations of Apoe⁺ progenitors and Sema5a⁺ progenitors in human required larger samples to confirm.

Therefore, we have rephrased the statement and demonstrate that the signature genes of CD26⁺ TSPCs were analogous in both murine and human scRNA-seq datasets in the result section in the revised manuscript.

The above information has been added in the discussion section in revised manuscript [Page 10, Line 173-180; Page 23, Line 451-462].

Figure 2. I. UMAP plot of MSC-1, MSC-2, MSC-3, osteoblast and chondrocyte/tenocyte. J. UMAP plot of marker genes in MSC-1, MSC-2, MSC-3, osteoblast and chondrocyte/tenocyte.

Figure 2K. Bubble plot of marker genes of each MSC cluster.

4. a. Figure 3 demonstrates increased proliferative capacity and trilineage differentiation capacity for CD26+ cells. This data strongly supports the stem cell characteristics of the CD26+ MSCs.

Response:

Thank you for your careful review and constructive comments.

b. The mouse reporter model is useful for lineage tracing, but the authors must first show the efficiency and specificity of the labeling?

Response:

Thank you for your careful review and constructive comments. To validate the efficiency and specificity of the labeling, tamoxifen was injected and tendon tissues were collected 5 days later. The results showed that approximately 73.25% of CD26+ cells were tdT positive (Figure S3G-H).

Figure S3G-H. Immunofluorescent staining and quantification of CD26 and TdTomato (tdT) in tendons post injection of tamoxifen.

c. Panel S3A should be added to this main document figure.

Response:

Thank you for your careful review and constructive comments. Panel S3A has been added to the main document figure as suggested.

Figure 3C. Proliferation of CD26+ tendon stem cells and Sema5a+ progenitors through CCK8 assay.

5. a. Figure 4 presents data from the injury model. The authors should justify the relevance of the punch injury, which is not a scenario seen clinically. Would the response be different in a relevant tendon injury model. For example, an Achilles tendon rupture with a repair presents a very different scenario. In the punch defect, a void is filled with infiltrating cells and remodeled. In a typical tendon injury, the torn ends heal back together.

Response:

Thank you for your careful review and constructive comments. Experimental tendon injury models offer a framework for investigating the mechanisms underlying tendon injury and healing [1]. These models include chemically-induced models, mechanical load-induced models, and mechanical injury models (partial injury or total transection) [1]. Chemically-induced models are attractive because they require less time and resources. Mechanical load-induced models have been used to elucidate the molecular and cellular mechanisms underlying tendinopathy and tendon healing in different

scenarios, such as tendon immobilization or overuse [2]. Tendon partial injury models, established through punch injury or partial transection, provide insight into the observation of the tendon healing process at specific regions and allow for the investigation of the effects of various factors and biomaterials on tendon repair, as well as complications such as fibrosis and heterotopic ossification [3-6]. The tendon total transection model has been used to study the molecular processes of tendon repair, identify growth factors with potential use in treating tendon disorders, and investigate different strategies for surgical repair [1, 7]. However, in this model, tendon torn ends healing would be disturbed without surgical intervention [1, 7]. In this study, we aimed to investigate the cell populations and underlying mechanisms during tendon healing and subsequent heterotopic ossification. Therefore, we chose the tendon punch injury model, in which both tendon healing and heterotopic ossification occur [3-6]. Since no animal models can perfectly mimic the clinical scenario of tendon injury, the role of CD26 TSPCs in other tendon injury models remains uncertain and worth further investigation to validate its contributions to tendon healing in different scenarios.

The above section has been added to the discussion [Page 28-29, Line 562-582; Page 30, Line 587-590].

References:

1. Zhang Q, Zhou D, Wang H, Tan J. Heterotopic ossification of tendon and ligament. *J Cell Mol Med.* 2020 May;24(10):5428-5437.
2. Nourissat G, Berenbaum F, Duprez D. Tendon injury: from biology to tendon repair. *Nat Rev Rheumatol.* 2015 Apr;11(4):223-33. doi: 10.1038/nrrheum.2015.26. Epub

2015 Mar 3. PMID: 25734975.

3. Hast MW, Zuskov A, Soslowsky LJ. The role of animal models in tendon research. *Bone Joint Res.* 2014 Jun;3(6):193-202. doi: 10.1302/2046-3758.36.2000281. PMID: 24958818

4. Harvey T, Flamenco S, Fan CM. A $Tppp3^+Pdgfra^+$ tendon stem cell population contributes to regeneration and reveals a shared role for PDGF signalling in regeneration and fibrosis. *Nat Cell Biol.* 2019 Dec;21(12):1490-1503. doi: 10.1038/s41556-019-0417-z. Epub 2019 Nov 25. PMID: 31768046

5. Donderwinkel I, Tuan RS, Cameron NR, Frith JE. Tendon tissue engineering: Current progress towards an optimized tenogenic differentiation protocol for human stem cells. *Acta Biomater.* 2022 Jun;145:25-42. doi: 10.1016/j.actbio.2022.04.028. Epub 2022 Apr 22. PMID: 35470075.

6. Tachibana N, Chijimatsu R, Okada H, Oichi T, Taniguchi Y, Maenohara Y, Miyahara J, Ishikura H, Iwanaga Y, Arino Y, Nagata K, Nakamoto H, Kato S, Doi T, Matsubayashi Y, Oshima Y, Terashima A, Omata Y, Yano F, Maeda S, Ikegawa S, Seki M, Suzuki Y, Tanaka S, Saito T. RSPO2 defines a distinct undifferentiated progenitor in the tendon/ligament and suppresses ectopic ossification. *Sci Adv.* 2022 Aug 19;8(33):eabn2138. doi: 10.1126/sciadv.abn2138. Epub 2022 Aug 19. PMID: 35984875.

7. Wang X, Li F, Xie L, Crane J, Zhen G, Mishina Y, Deng R, Gao B, Chen H, Liu S, Yang P, Gao M, Tu M, Wang Y, Wan M, Fan C, Cao X. Inhibition of overactive TGF- β attenuates progression of heterotopic ossification in mice. *Nat Commun.* 2018 Feb

7;9(1):551. doi: 10.1038/s41467-018-02988-5. PMID: 29416028.

8. Voleti PB, Buckley MR, Soslowsky LJ. Tendon healing: repair and regeneration. *Annu Rev Biomed Eng.* 2012;14:47-71. doi: 10.1146/annurev-bioeng-071811-150122. PMID: 22809137.

b. Panels S5F and S5G should be in main document figure; these panels are important, as they show early expression and expansion of CD26+ cells.

Response:

Thank you for your careful review and constructive comments. As suggested, Panel S5F to S5H have been added to the main document figure (Figure 4D-F).

Figure 4. D. Immunofluorescent staining of CD26 and EdU in tendon from sham and punched group. Scale bar: 100 μ m. E. Quantification of CD26 positive cell rate in tendon from sham and punched group. F. Quantification of CD26⁺EdU⁺ cell rate in tendon from sham and punched group.

c. The paper would be strengthened if the authors added scRNAseq of FACS-sorted CD26⁺ lineage traced cells from multiple timepoints. This would provide a cleaner isolation and analysis of the cells of interest.

Response:

Thank you for your careful review and constructive comments. Indeed, scRNAseq of FACS-sorted CD26⁺ lineage traced cells from multiple timepoints would strengthen the conclusion that CD26⁺ TSPC could develop into tenocytes. Nonetheless, to achieve this goal, two principal challenges have been indentified. First, due to the diminutive size of the tendon, further compounded by reduced part of injured region, a very large number of samples (estimated around 80 genetic -engeineered mice) would be needed to obtain high quality cells for scRNA-seq at multiple timepoint. Second, the robust and fibrous composition of ligamentous structures demands prolonged enzymatic digestion and FACS-sorted process, which might significantly affect the number of living cells and induce alterations in the gene expression profile, making it difficult to implement.

To better illustrated the differentiation route of CD26⁺ MSCs, we performed cytotrace analysis to strengthen the conclusion that CD26⁺ MSCs could developed into tenocytes and chondrocytes/osteoblasts after tendon injury (Figure 4B, G and S5J).

The above limitation has been added in the revised manuscript. [Page 29 Line 583-587]

Figure 4B. Umap plot of CD26+ TSPCs, Apoe+ proenitors, Sema5a+ progenitors, Chondrocyte/Tenocyte, Tenocytes and Chondrocyte from the merged group. G, Cytotrace analysis on CD26+ TSPCs, Apoe+ proenitors, Sema5a+ progenitors, Chondrocyte/Tenocyte, Tenocytes and Chondrocyte from the merged group.

Figure S5J. Cytotrace score among CD26+ TSPCs, Apoe+ proenitors, Sema5a+ progenitors, Chondrocyte/Tenocyte, Tenocytes and Chondrocyte

References:

1. Merrick D, Sakers A, Irgebay Z, Okada C, Calvert C, Morley MP, Percec I, Seale P. Identification of a mesenchymal progenitor cell hierarchy in adipose tissue. *Science*. 2019 Apr 26;364(6438):eaav2501. doi: 10.1126/science.aav2501. PMID: 31023895
2. Harvey T, Flamenco S, Fan CM. A $Tppp3^+Pdgfra^+$ tendon stem cell population contributes to regeneration and reveals a shared role for PDGF signalling in regeneration and fibrosis. *Nat Cell Biol*. 2019 Dec;21(12):1490-1503. doi: 10.1038/s41556-019-0417-z. Epub 2019 Nov 25. PMID: 31768046
6. a. Figure 5 lacks functional assessment. The authors should add gait and/or tendon biomechanics to assess healing.

Response:

Thank you for your careful review and constructive comments. To evaluate the mechanical properties of the tendon, we utilized a mechanical testing system to measure the maximum force and stress at failure. Maximum force refers to the greatest amount of force a material can withstand before it begins to experience permanent deformation or failure. Stress at failure is the stress a material is under when it fails or breaks, calculated by dividing the applied force at the point of failure by the cross-sectional area of the material. This value indicates the strength of the material. Mechanical testing was performed using a universal testing machine (DR-507AS). The hind limbs were collected, and tissues spanning the ankle, except for the Achilles tendon, were transected. The muscle-tendon-calcaneus complex was rigidly fixed to clamps, and a load-to-failure test was performed at an elongation rate of 10 mm/min. The results showed that mechanical properties, including maximum force and stress at failure, were decreased with the depletion of CD26+ cells (Figure 5I-J).

Figure 5. I. Quantification of Maximum Force in Veh. and DTx group. J. Quantification of Stress at failure in Veh. and DTx group. Data shown are mean \pm SEM.

b. What is the efficiency of DTR for cell ablation?

Response:

Thank you for your careful review and constructive comments. Immunofluorescent staining revealed that the CD26⁺ cell rate decreased from 29.29±0.64% in the vehicle group to 6.80±0.82% in the diphtheria toxin group, indicating successful depletion of CD26⁺ cells (Figure R3).

Figure R3 Immunofluorescent staining and quantification of CD26⁺ cells in Veh. and DTx group.

n=3. Data shown are mean ± SEM.

c. The authors cannot claim these cells are "essential" for tendon healing based solely on histologic evidence.

Response:

Thank you for your careful review and constructive comments. We apologize for the overstated description of the results. To better demonstrate the important role of CD26⁺ TSPCs in tendon healing, we utilized a mechanical testing system to measure the maximum force and stress at failure of the tendon, as mentioned above. The results

showed that mechanical properties, including maximum force and stress at failure, were decreased with the depletion of CD26+ cells (Figure 5I-J). Collectively, we evaluated the expression of tenogenic markers, collagen fibril diameter, and mechanical properties of the tendon to illustrate the tendon healing process [1,2]. To precisely describe the findings of the current study, we have rephrased the overstated statement in the revised manuscript [Page 16, Line 304-307].

References:

1. Zhang C, Zhang E, Yang L, Tu W, Lin J, Yuan C, Bunpetch V, Chen X, Ouyang H. Histone deacetylase inhibitor treated cell sheet from mouse tendon stem/progenitor cells promotes tendon repair. *Biomaterials*. 2018 Jul;172:66-82. doi: 10.1016/j.biomaterials.2018.03.043. Epub 2018 Mar 26. PMID: 29723756.
2. Harvey T, Flamenco S, Fan CM. A Tppp3+Pdgfra+ tendon stem cell population contributes to regeneration and reveals a shared role for PDGF signalling in regeneration and fibrosis. *Nat Cell Biol*. 2019 Dec;21(12):1490-1503. doi: 10.1038/s41556-019-0417-z. Epub 2019 Nov 25. PMID: 31768046
7. Figures 6-7: How clinically relevant is the HO model? This level of ossification not typical in human tendon injury responses.

Response:

Thank you for your careful review and constructive comments. Heterotopic ossification of tendon is common in populations with high risk of tendon injury such as athletes and workers with repetitive tendon overuse [1,2]. Additionally, HO formation has been

reported in 14-62% of cases following surgical repair after tendon injury [3,4]. There is no effective pharmacological therapy for HO formation. Identification of the specific cells responsible for HO during tendon healing is crucial for elucidating the cellular and molecular mechanisms underlying HO, which could provide insights for therapeutic strategies. [Page 5-6, Line 80-86].

As mentioned above, in this study, we aim to investigate the cell populations and underlying mechanism during tendon healing and subsequent heterotopic ossification, therefore, we chose the tendon punch injury model in which both tendon healing and heterotopic ossification would occur. In this model, the level of ossification increases over time and remain stable at late stage. In addition, trauma-induced heterotopic ossification is prone to develop in mice after tendon injury and these animal models are widely used. The variation of final ossification level between human and animal model might due to different observation timepoint and the genetic variation between different species [2, 7-8].

References:

1. Brinkman JC, Zaw TM, Fox MG, Wilcox JG, Hattrup SJ, Chhabra A, Neville MR, Hartigan DE. Calcific Tendonitis of the Shoulder: Protector or Predictor of Cuff Pathology? A Magnetic Resonance Imaging-Based Study. *Arthroscopy*. 2020 Apr;36(4):983-990. doi: 10.1016/j.arthro.2019.11.127. Epub 2019 Dec 7. PMID: 31816365.

2. Zhang Q, Zhou D, Wang H, Tan J. Heterotopic ossification of tendon and ligament. *J Cell Mol Med*. 2020 May;24(10):5428-5437. doi: 10.1111/jcmm.15240. Epub 2020

Apr 15. PMID: 32293797.

3. O'Brien EJ, Frank CB, Shrive NG, Hallgrímsson B, Hart DA. Heterotopic mineralization (ossification or calcification) in tendinopathy or following surgical tendon trauma. *Int J Exp Pathol*. 2012 Oct;93(5):319-31. doi: 10.1111/j.1365-2613.2012.00829.x. PMID: 22974213

4. Ateschrang A, Gratzner C, Weise K. Incidence and effect of calcifications after open-augmented Achilles tendon repair. *Arch Orthop Trauma Surg*. 2008 Oct;128(10):1087-92. doi: 10.1007/s00402-007-0441-5. Epub 2007 Sep 15. PMID: 17874248.

5. Tachibana N, Chijimatsu R, Okada H, Oichi T, Taniguchi Y, Maenohara Y, Miyahara J, Ishikura H, Iwanaga Y, Arino Y, Nagata K, Nakamoto H, Kato S, Doi T, Matsubayashi Y, Oshima Y, Terashima A, Omata Y, Yano F, Maeda S, Ikegawa S, Seki M, Suzuki Y, Tanaka S, Saito T. RSPO2 defines a distinct undifferentiated progenitor in the tendon/ligament and suppresses ectopic ossification. *Sci Adv*. 2022 Aug 19;8(33):eabn2138. doi: 10.1126/sciadv.abn2138. Epub 2022 Aug 19. PMID: 35984875

6. Wang X, Li F, Xie L, Crane J, Zhen G, Mishina Y, Deng R, Gao B, Chen H, Liu S, Yang P, Gao M, Tu M, Wang Y, Wan M, Fan C, Cao X. Inhibition of overactive TGF- β attenuates progression of heterotopic ossification in mice. *Nat Commun*. 2018 Feb 7;9(1):551. doi: 10.1038/s41467-018-02988-5. PMID: 29416028

7. Docheva D, Müller SA, Majewski M, Evans CH. Biologics for tendon repair. *Adv Drug Deliv Rev*. 2015 Apr;84:222-39. doi: 10.1016/j.addr.2014.11.015. Epub 2014 Nov 21. PMID: 25446135; PMCID: PMC4519231.

8. Sorkin M, Huber AK, Hwang C, Carson WF 4th, Menon R, Li J, Vasquez K, Pagani C, Patel N, Li S, Visser ND, Niknafs Y, Loder S, Scola M, Nycz D, Gallagher K, McCauley LK, Xu J, James AW, Agarwal S, Kunkel S, Mishina Y, Levi B. Regulation of heterotopic ossification by monocytes in a mouse model of aberrant wound healing. *Nat Commun.* 2020 Feb 5;11(1):722. doi: 10.1038/s41467-019-14172-4. PMID: 32024825; PMCID: PMC7002453.

8. Figure 7: The authors cannot claim "TNC-Hippo pathways dependent", as only descriptive *in vivo* experiments were performed. Even *in vitro*, only a limited set of mechanistic experiments were done.

Response:

Thank you for your careful review and constructive comments.

Hippo signaling is critical for cell proliferation and differentiation. In the current study, we found that depletion of CD26⁺ cells led to reduced HO formation, indicating that these cells are responsible for ectopic bone formation (Figure 7B-C). Furthermore, we observed that during the process of tendon healing, approximately 90.00% at 30 dpi and 86.54% at 60 dpi of p-YAP⁺ cells were CD26⁺ cells, indicating that inhibition of the Hippo pathway mainly affects the CD26⁺ cell population at these stages (Figure S7A-B).

Indeed, the claim that CD26⁺ TSPCs contribute to HO formation in a TNC-Hippo pathway-dependent manner was overstated. Therefore, we have rephrased the claim to demonstrate that TNC-Hippo signaling is involved in trauma-induced heterotopic

ossification. [Page 17, Line 330]

To further investigate the role of TNC in HO formation, tendon tissues at different time points were collected, and we observed that the expression of TNC was aberrantly increased at the stage of cartilage template formation and decreased at 90 dpi when ectopic bone was prominent (Figure R2). To further investigate whether targeting TNC would affect the tendon healing process, anti-TNC antibody was administered to neutralize TNC protein. The results showed that collagen fibril diameter and mechanical properties were not significantly affected in the anti-TNC group compared to the IgG control, indicating that targeting TNC did not significantly affect the process of tendon healing (Figure S87C-G). Collectively, these results indicate that TNC is a potential target for heterotopic ossification. [Page 19, Line 373-378]

Figure S7. A-B Immunofluorescent staining and quantification of CD26 and p-YAP of Achilles tendon from Sham, 30 dpi and 60dpi group. $n=3$. One way ANOVA with

Turkey's test. Scale bar: 100 μ m. Data shown are mean \pm SEM.

Figure R2. Histological analysis and quantification of TNC in Sham, 30 dpi and 90 dpi group. C: cartilage formation site; B: Bone. n=5. Scale bar: 100 μ m.

Figure S8. C, TEM images of cross section of Achilles tendons from IgG, and anti-TNC group. Scale 500nm. D-E, Bar graph presents the aggregated distribution of collagen fibril diameters using high-magnification TEM images from n = 3 mice per group. F-G. Mechanical properties evaluation

of Achilles tendons in IgG, and anti-TNC group. n = 3. Data shown are mean \pm SEM.

9. Please check for grammar and word choice - there are a number of errors in the paper.

Response:

Thank you for your careful review and constructive comments. The grammatical errors and word choices have been corrected by native English speakers.

Reviewer #3 (Remarks to the Author):

The manuscript by Chen and colleagues have set out to test the mechanisms underlying tendon regeneration and healing. Setting out from a single cell analysis, they identified a cell population, CD26 positive, which serve as tendon progenitor cells. Using bioinformatics analyses, in vitro differentiation assays, in vivo lineage tracing and ablation assays they show this cell population plays a critical role in tendon healing as well as in heterotopic bone formation, a debilitating process that occurs in multiple cases of tendon healing. The authors finally claim that this heterotopic bone formation and the regulation of the CD26 population is TNC and hippo pathway dependent.

Major Comments

1. Overall, hardly any rationale for carrying out the experiments and methodology is given rendering reading the manuscript and following the work very hard. For example, they generate the Cd26-CreERT2; hDTR mouse line yet do not show how it was done, what is the definition of DTR. It is also not clear when the toxin was given and when

just tamoxifen. Further, in the injury experiment - what age were the mice used? It is only mentioned adult. Also, why 14 dpi were chosen? Likewise, the EdU regime – when and why were the specific times chosen.

Response:

Thank you for your careful review and constructive comments. To better illustrate the findings of the current study, we have revised the Results section to provide a clearer rationale for carrying out the experiments. Detailed information on the methodology has been added to the Methods section, which we hope will enhance the readability, logic, and rationality of the manuscript.

For the generation of CD26-creERT2-hDTR mice, the CRISPR/Cas9 technology was utilized to precisely integrate the Dpp4-creERT2-P2A-hDTR-PolyA gene cassette into the Rosa26 locus of mice. The brief procedure is as follows: sgRNA was transcribed in vitro, and a donor vector was constructed. Cas9, donor vector, and sgRNA were co-injected into fertilized eggs of C57BL/6JGpt mice. F0 generation mice were obtained and validated for correct integration via PCR and sequencing. Positive F0 mice were then bred with C57BL/6JGpt mice to generate a stable F1 generation of mice carrying the genetic modification. [Page 30-31, Line 607-614] In this genetic model, CD26⁺ cells express both creERT and hDTR (human diphtheria toxin receptor). Ablation of CD26⁺ cells can be induced by administering diphtheria toxin, while the function of creERT can be activated by administering Tamoxifen.

For inducible ablation of CD26⁺ cells at the injury sites, each CD26-creERT2-hDTR mouse was locally injected with 50ng diphtheria toxin (List Labs) daily for 5

days (Figure 5A). [Page 31, Line 617-619]

Figure 5A. Scheme for the DTx induction.

For CD26 lineage tracing, CD26-creERT2-hDTR; mTmG mice were injected intraperitoneally with 100 mg/kg daily tamoxifen for 5 days (Figure 3H). [Page 31, Line 616-617]

Figure 3H. Scheme for the TMX induction.

In the current study, we aim to identify the tendon stem/progenitor cells and the underlying molecular mechanisms that contribute to tendon healing and heterotopic ossification in adulthood. To achieve this, we used mice aged 9 weeks as our tendon injury model. [Page 35, Line 691-693]

Furthermore, we aim to investigate the cell population and molecular mechanisms involved in tendon healing and heterotopic ossification following tendon injury. Cartilage template formation and subsequent endochondral ossification are important

processes in the formation of heterotopic ossification. Tachibama et al. found that Sox9⁺ or Col2⁺ chondrocytes peak at 14 days post-injury (dpi), indicating that progenitor cells for cartilage template formation may be prominent at this stage [1]. Therefore, we chose 14 dpi to perform single-cell RNA sequencing (scRNA-seq). [Page 16, Line 309-311]

Additionally, Harvey et al. found that tendon progenitor cells proliferate at 7 dpi in a tendon injury model [2]. Therefore, we chose 7 dpi to investigate the expansion of CD26⁺ cells in vivo. [Page 14, Line 254-257]

Detailed information related to the rationale for carrying out the experiments and methodology has been added to the manuscript.

References:

1. Tachibana N, Chijimatsu R, Okada H, Oichi T, Taniguchi Y, Maenohara Y, Miyahara J, Ishikura H, Iwanaga Y, Arino Y, Nagata K, Nakamoto H, Kato S, Doi T, Matsubayashi Y, Oshima Y, Terashima A, Omata Y, Yano F, Maeda S, Ikegawa S, Seki M, Suzuki Y, Tanaka S, Saito T. RSPO2 defines a distinct undifferentiated progenitor in the tendon/ligament and suppresses ectopic ossification. *Sci Adv.* 2022 Aug 19;8(33):eabn2138. doi: 10.1126/sciadv.abn2138. Epub 2022 Aug 19. PMID: 35984875.
2. Harvey T, Flamenco S, Fan CM. A Tppp3⁺Pdgfra⁺ tendon stem cell population contributes to regeneration and reveals a shared role for PDGF signalling in regeneration and fibrosis. *Nat Cell Biol.* 2019 Dec;21(12):1490-1503. doi: 10.1038/s41556-019-0417-z. Epub 2019 Nov 25. PMID: 31768046.

2. Throughout the manuscript, no or hardly any quantifications are shown and it is therefore extremely hard to validate the data.

Response:

Thank you for your careful review and constructive comments. We have now included quantifications for the experiments, specifically for the differentiation assays, cell proliferation assay, immunofluorescent staining, and flow cytometry presented in Figure 3C-E and 4K-L, as well as in Supplementary Figure S3G-H and S6E-G.

Figure 3C, CCK8 assay of CD26+ MSCs and Sema5a+ progenitors. n=3. Two-tailed unpaired Student's t test.

Figure 4K-L. Immunofluorescent staining and quantification of Fmod and tdTomato (tdT) in tendon from Sham and 30 dpi group. n=5. Scale bar: 100 μ m.

Figure S3G-H, Immunofluorescent staining and quantification of CD26 and tdT after tamoxifen induction in *CD26-CreER^{T2}-hDTR* mice. Data shown are mean \pm SEM.

Figure 3. D, Alizarin red, alcian blue and oil red O staining of CD26⁺ MSCs and Sema5a⁺ progenitors under osteogenesis, chondrogenesis or adipogenesis. Scale bar: 100 μ m. E, Quantification of CD26⁺ MSCs and Sema5a⁺ progenitors under osteogenesis, chondrogenesis or adipogenesis. n=3. Two-tailed unpaired Student's t test.

Figure S6. E-G, Flow cytometry analysis of CD45⁺CD11b⁺ myeloid cells in Veh. and DTx group. n=3. Two-tailed unpaired Student's t test.

3. The authors fail to describe in detail how they carried out pseudotime trajectory analysis of MSC clusters in their scRNAseq data. As CD26⁺ are shown to be at the origin of the pseudotime trajectory, was this cluster chosen manually as such or was it chosen by the package's algorithm? If it was chosen manually, what was the rationale behind this choice? Additionally, the authors use Monocle 3 package for their first trajectory analysis, then switch over to slingshot algorithm in Figure 4E - what is the reason for this change of practice?

Response:

Thank you for your careful review and constructive comments. We apologize for the unclear demonstration of the pseudotime trajectory analysis in Figure 1 and Figure 4. To avoid further confusions related to pseudotime trajectory analysis, we decided to utilize CytoTRACE and Monocle algorithm to perform this analysis as suggested. To objectively determine the origin of pseudotime trajectory, we first performed

CytoTRACE analysis on CD26⁺ TSPCs, Apoe⁺ progenitors and Sema5a⁺ progenitors from the uninjured group and CD26⁺ TSPCs, Apoe⁺ progenitors, Sema5a⁺ progenitors, Chondrocyte/Tenocyte, Tenocytes and Chondrocyte from the merged group. The result showed that CD26⁺ TSPCs presented the highest cytotrace score in both the uninjured and merged single-cell RNA sequencing (scRNA-seq) datasets, indicating that CD26⁺ TSPCs were the least differentiated among these populations (Figure 1E, 4G and S5J). Next, pseudotime trajectory analysis was performed with Monocle algorithm and the results showed that CD26⁺ TSPCs was the beginning of the trajectory (Figure S1F and S5F-I). [Page 9, Line 149-153; Page 14, Line 262-268]

Figure 1. E, CytoTRACE analysis of CD26⁺ TSPCs, Apoe⁺ progenitors and Sema5a⁺ progenitors from the uninjured group.

Figure 4B. Umap plot of CD26⁺ TSPCs, Apoe⁺ progenitors, Sema5a⁺ progenitors,

Chondrocyte/Tenocyte, Tenocytes and Chondrocyte from the merged group. G, Cytotrace analysis on CD26+ TSPCs, Apoe+ progenitors, Sema5a+ progenitors, Chondrocyte/Tenocyte, Tenocytes and Chondrocyte from the merged group.

Figure S5J. Cytotrace score among CD26+ TSPCs, Apoe+ progenitors, Sema5a+ progenitors, Chondrocyte/Tenocyte, Tenocytes and Chondrocyte.

Figure S1F, Pseudotime analysis of CD26+ TSPCs, Apoe+ progenitors and Sema5a+ progenitors in uninjured group.

Figure S5F-I, Pseudotime analysis of CD26+ TSPCs, Apoe+ progenitors, Sema5a+ progenitors, Chondrocyte/Tenocyte, Tenocytes and Chondrocyte in merged group.

4. The authors show that ~40% of the CD26 cells are EdU positive. What about the other 60%? What are they and where are they found?

Response:

Thank you for your careful review and constructive comments. The EdU (5-ethynyl-2'-deoxyuridine) assay is a method commonly used for detecting cellular DNA synthesis, particularly in the study of cell proliferation and DNA replication. We found that 40% of CD26⁺ cells were EdU⁺ after tendon injury, which was significantly higher than the percentage before injury. This indicates that the proliferation of these cells was activated. CD26⁺EdU⁻ cells were considered not to be undergoing DNA replication. These cells may not participate in the process of tendon healing or have stopped proliferating. It is reasonable to observe that the process of DNA replication and cell proliferation does not occur in all cells at the same time. In addition, we found that these cells were located both at the peritendon and in the tendon midsubstance after tendon injury.

Figure 4D-F. Immunofluorescent staining and quantification of EdU and CD26 in tendon

from Sham and 30 dpi group. n=5. Scale bar: 100 μ m. Data shown are mean \pm SEM.

5. In Figure 2C, Cd26 and Lama4 show almost complete overlap in the tendon sheath, however in Figure S2A, significantly less overlap between these stainings is observed. Moreover, signal for both Cd26 and Lama4 is detected in these images in the middle of the tail tendon, which the authors do not address. Please clarify.

Response:

Thank you for your careful review and constructive comments. The less overlap and staining in the middle of the tai tendon may have been caused by uneven sectioning and staining. To address this, we re-performed the fluorescent co-staining for CD26 and Lama4, and the results showed that CD26+Lama4+ cells were located surrounding the tendon. Approximately 90.71% of CD26+ cells were Lama4 positive, indicating that most of the CD26+ cells reside within the tendon sheath (Figure S2B).

Figure S2B. immunofluorescent analysis and quantification of CD26 and Lama4 in cross section of tail tendon from adult mice. n=3. Scale bar: 100 μ m.

6. The authors claim the mouse and human MSC clusters are analogous. They back this

claim by presenting expression of very few genes in the human data. Claiming such a claim, requires they show expression of a larger number of genes, or compute the correlation between cluster identities between species. Following such additions, their claim would be much stronger.

Response:

Thank you for your careful review and constructive comments. We included a larger number of marker gene and the result showed that cluster MSC-1 expressed high level of Dpp4, CD55, Pi16, CD248 and Sema3c, which was also highly expressed in CD26⁺ TSPC in mice scRNAseq data (Figure 2I-K). These results indicate that CD26⁺ TSPC cluster is analogous between mouse and human species.

However, although we found another two populations of MSC in human dataset (MSC-2 and MSC-3), MSC-2 and MSC-3 showed few expressions of markers of Apoe⁺ progenitors and Sema5a⁺ progenitors. Due to the differences between human and murine species, the signature genes of these two cell populations may exhibit disparities. We could not identify analogues populations of Apoe⁺ progenitors and Sema5a⁺ progenitors in human scRNA-seq dataset. In human, MSC-2 is enriched in expression of Apoe, Sfrp1, Vcam1 and Pla2g2a, indicating that these cells were associated with lipid metabolism, which were partially similar to Apoe⁺ progeniros found in murine scRNA-seq dataset. As for MSC-3, Lcp1, Lyz, s100a12 and LST1 were enriched, indicating that these cells were associated with immune response. Nonetheless, we found that the signature genes of CD26⁺ TSPCs were analogous in both murine and human scRNA-seq datasets. The identification of analogues populations of Apoe⁺

progenitors and *Sema5a*⁺ progenitors in human required larger samples to confirm.

Therefore, we have rephrased the statement and demonstrate that the signature genes of *CD26*⁺ TSPCs were analogous in both murine and human scRNA-seq datasets in the result section in the revised manuscript.

The above information has been added in the discussion section in revised manuscript [Page 10, Line 173-180; Page 23, Line 451-462].

Figure 2. I. UMAP plot of MSC-1, MSC-2, MSC-3, osteoblast and chondrocyte/tenocyte. J. UMAP plot of marker genes in MSC-1, MSC-2, MSC-3, osteoblast and chondrocyte/tenocyte.

Figure 2K. Bubble plot of marker genes of each MSC cluster.

7. Quantifications of the differentiation assays shown in Figure 3C must be added.

Moreover, from the image it seems as if the Sema5a+ MSC show stronger Alizarin red staining than Cd26+ MSC, which could suggest better differentiation. Results of the quantifications should be addressed.

Response:

Thank you for your careful review and constructive comments. Quantifications of the differentiation assays have been added in Figure 3E. The results showed that the osteogenic potential of CD26+ MSCs is lower than that of Sema5a+ progenitors, while

Sema5a⁺ progenitors hardly differentiate into adipocytes. The chondrogenic capacity of these two cell populations showed no difference. The difference in the differentiation potential between CD26⁺ MSCs and Sema5a⁺ progenitors indicates that CD26⁺ MSCs are more primitive than Sema5a⁺ progenitors, while Sema5a⁺ progenitors are more committed to the osteogenic lineage. [Page 11, Line 191-193]

Figure 3. D. Alizarin red, alcian blue and oil red O staining of CD26⁺ TSPCs and Sema5a⁺ progenitors under osteogenesis, chondrogenesis or adipogenesis. n=3. Scale bar: 100 μ m. E. Quantification of osteogenesis, chondrogenesis and adipogenesis in Figure 3D. Data shown are mean \pm SEM.

8. Leinroth et al.,2022 (PMID 35545045) carried out similar experiments with FAPs isolated from skeletal muscles, where they demonstrated higher adipogenic potential for Cd26+ FAPs, but low osteogenic potential. As connective tissue lineage plasticity has been a subject of extensive research, the authors should address Leinroth et al.,2022 findings and the differentiation potential of FAP subpopulations compared to their tendon MSC?

Response:

Thank you for your careful review and constructive comments. The CD26+ mesenchymal cell population has been found in muscle and adipose tissues in previous studies, as well as in tendon tissues in the current study [1,2]. This cell population expresses high levels of marker genes, including Dpp4, Pi16, and Anxa3 [1,2]. However, the differentiation potential of mesenchymal cells collected from different tissues is not identical. Leinroth et al. found that CD26+ FAPs collected from muscles adjacent to tendons showed poor osteogenic capacity [1]. Merrick et al. found that CD26+ progenitor cells collected from subcutaneous adipose tissue displayed enhanced competence for differentiation into osteocytes compared to ICAM+ cells [2]. In the current study, CD26+ TSPCs collected from tendons displayed triple-lineage differentiation capacity. These results indicate that CD26+ mesenchymal cells collected from different tissues display different differentiation capacities. In addition, we found that CD26+ TSPCs could differentiate into tenocytes, chondrocytes, and osteoblasts in vivo. Whether CD26+ FAPs contribute to tendon healing remains uncertain.

References:

1. Leinroth AP, Mirando AJ, Rouse D, Kobayahi Y, Tata PR, Rueckert HE, Liao Y, Long JT, Chakkalakal JV, Hilton MJ. Identification of distinct non-myogenic skeletal-muscle-resident mesenchymal cell populations. *Cell Rep.* 2022 May 10;39(6):110785. doi: 10.1016/j.celrep.2022.110785. PMID: 35545045
2. Merrick D, Sakers A, Irgebay Z, Okada C, Calvert C, Morley MP, Percec I, Seale P. Identification of a mesenchymal progenitor cell hierarchy in adipose tissue. *Science.* 2019 Apr 26;364(6438):eaav2501. doi: 10.1126/science.aav2501. PMID: 31023895

9. Along these lines, the authors claim also in the Abstract, that the CD26 population is a novel population. However, this population has been shown in multiple other settings therefore the authors should rephrase their claim.

Response:

Thank you for your careful review and constructive comments. As described above in response to comment 8, CD26⁺ TSPCs collected from tendon tissues displayed different differentiation capacities, indicating that this cell population was not identical to previously identified CD26⁺ populations in other tissues. Furthermore, according to our findings, CD26⁺ TSPCs residing within tendon tissues and their role in tendon healing and heterotopic ossification have not been reported in previous studies. Since some of the marker genes among different CD26⁺ populations were similar, we have rephrased the claim in the Abstract to "we identified a CD26⁺ tendon stem/progenitor cell population residing in peritendon," as suggested. [Page 3, Line 29-31]

10. In Figure 3D, the authors describe an in-vitro differentiation experiment of MSC subpopulation identities, using only two markers (one for each subpopulation). This claim is not supported by sufficient data. Additional markers for each subpopulation (e.g., using FACS or RT-PCR for example) is required to claim this change of identity, especially when using ex-vivo assays where the cells are not in their natural environment.

Response:

Thank you for your careful review and constructive comments. To validate whether CD26⁺ TSPCs could develop into Sema5a⁺ progenitors, we cultured FACS-sorted CD26⁺ TSPCs and Sema5a⁺ progenitors separately in tentative tendon stem/progenitor cell (TSPCs) medium for 14 days. RT-PCR results showed increased expression of Sema5a⁺ progenitor marker genes (Sema5a, Enpp1, Hmcn1, Smoc2) and reduced expression of CD26⁺ TSPCs marker genes (Dpp4, Sbsn, Stmn4) in the CD26⁺ TSPCs group (Figure S3E). However, the Sema5a⁺ progenitors' group did not show increased expression of CD26⁺ TSPCs marker genes (Figure S3F). These results indicate that CD26⁺ TSPCs can develop into Sema5a⁺ progenitors. [Page 11, Line 201-205]

Figure S3D. mRNA expression level changes of marker genes of CD26+ TSPC and Sema5a+ progenitor in CD26+ TSPCs cultured in tentative culture medium for 14 days.

Figure S3E. mRNA expression level changes of marker genes of CD26+ TSPC and Sema5a+ progenitor in Sema5a+ progenitors cultured in tentative culture medium for 14 days.

11. The section from line 186 to line 197 is very poorly written. It has insufficient background and requires citation and references to justify the genes used by the authors for their claim. Labeling of the figures should be clearer to enable the reader understand when were human or mouse cells used, and what subpopulation of MSC was examined.

Response:

Thank you for your careful review and constructive comments. To determine the marker gene expression profile of CD26⁺ TSPCs, we isolated murine CD26⁺ TSPCs using fluorescence activated cell sorting (FACS) and examined their expression of various stem cell markers. These markers include CD90, CD44, and CD105, which have been previously found to be expressed in identified tendon stem/progenitor cells (TSPCs) and tendon-derived progenitor cells (TDPCs) [1, 2]. Additionally, the leptin receptor has been reported as a bone marrow-derived MSC marker gene [3].

The above information has been added to the revised manuscript. [Page 12, Line 219-221]

Furthermore, as suggested, labels for cell sources and subpopulations have been added to Figure 3J and Figure S4A.

Figure 3J. Flow cytometry analysis of stem cell and tenocyte markers in CD26⁺ TSCs.

Figure S4A. Flow cytometry analysis of stem cell markers from human tendons (CD44, CD105, CD90 and LepR) in CD26⁺ cells.

References:

1. Feng H, Xing W, Han Y, Sun J, Kong M, Gao B, Yang Y, Yin Z, Chen X, Zhao Y, Bi Q, Zou W. Tendon-derived cathepsin K-expressing progenitor cells activate Hedgehog signaling to drive heterotopic ossification. *J Clin Invest.* 2020 Dec 1;130(12):6354-6365. doi: 10.1172/JCI132518. PMID: 32853181
2. Bi Y, Ehirchiou D, Kilts TM, Inkson CA, Embree MC, Sonoyama W, Li L, Leet AI, Seo BM, Zhang L, Shi S, Young MF. Identification of tendon stem/progenitor cells and the role of the extracellular matrix in their niche. *Nat Med.* 2007 Oct;13(10):1219-27. doi: 10.1038/nm1630. Epub 2007 Sep 9. PMID: 17828274.
3. Zhou BO, Yue R, Murphy MM, Peyer JG, Morrison SJ. Leptin-receptor-expressing mesenchymal stromal cells represent the main source of bone formed by adult bone marrow. *Cell Stem Cell.* 2014 Aug 7;15(2):154-68. doi: 10.1016/j.stem.2014.06.008. Epub 2014 Jun 19. PMID: 24953181
12. Throughout the manuscript, the authors hardly describe what were ages the mice used and why were these ages chosen.

Response:

Thank you for your careful review and constructive comments. In the current study, our aim is to identify the tendon progenitor cells and the underlying molecular mechanisms that contribute to tendon healing and heterotopic ossification. Therefore, mice aged 9 weeks at adulthood were chosen for the tendon injury model.

The above information has been added to the revised method section [Page 7, Line 110-112; Page 35, Line 691-693].

13. In Figure 4, the Apoe+ cluster seems significantly bigger than in the previous scRNAseq data presented. Could the authors address what were the differences between experiments?

Response:

Thank you for your careful review and constructive comments. The Apoe+ cluster presented in Figure 4B represents a combined cluster of both the uninjured and injured groups. In Figure 1C, the Apoe+ cluster was specifically presented in the uninjured group. Therefore, the cell number of the Apoe+ cluster in Figure 4B seems larger than that in Figure 1C.

Figure 1C. UMAP plot of cluster 1-4.

Figure 4B. UMAP plot of CD26+ TSPCs, Apoe+ progenitors, Sema5a+ progenitors, Chondrocyte/Tenocyte, Tenocytes and Chondrocyte from the merged group.

14. Figure S6D- stiffness and modulus are poorly explained both in meaning and methods.

Response:

Thank you for your careful review and constructive comments. Stiffness refers to the measure of a material's resistance to deformation. A more specific term that quantifies stiffness is the effective Young's modulus, which represents the slope of the stress-strain curve over a particular range of deformation.

Detailed information regarding the mechanical properties testing has been added to the revised manuscript. (Page 38, Line 771-774)

15. The authors need to better introduce the process of HO as part of tendon healing. Is HO a naturally occurring process in healthy tendon healing? Is it always pathological? How long does it take for the HO to disappear from the tendon in normal healing?

Response:

Thank you for your careful review and constructive comments. Heterotopic ossification (HO) is not a naturally occurring process in healthy tendon healing [1]. The healthy tendon healing process includes the inflammatory stage, proliferative stage, and remodelling phase. However, HO of the tendon is common in populations with a high risk of tendon injury, such as athletes and workers with repetitive tendon overuse or severe tendon injury [2-5]. Furthermore, HO formation has been reported in 14-62% of cases following surgical repair after tendon injury [6, 7]. HO is a pathological process after tendon injury that causes pain and restricts range of motion [3, 8]. Once HO develops in the tendon, it does not disappear [1].

Some of the above information has been added to the revised manuscript [Page 5-

6, Line 80-86]. Due to space limitations, in the section introducing heterotrophic ossification (HO), we have referred to relevant literature.

References:

1. Voleti PB, Buckley MR, Soslowky LJ. Tendon healing: repair and regeneration. *Annu Rev Biomed Eng.* 2012;14:47-71. doi: 10.1146/annurev-bioeng-071811-150122. PMID: 22809137.
2. Brinkman JC, Zaw TM, Fox MG, Wilcox JG, Hattrup SJ, Chhabra A, Neville MR, Hartigan DE. Calcific Tendonitis of the Shoulder: Protector or Predictor of Cuff Pathology? A Magnetic Resonance Imaging-Based Study. *Arthroscopy.* 2020 Apr;36(4):983-990. doi: 10.1016/j.arthro.2019.11.127. Epub 2019 Dec 7. PMID: 31816365.
3. Zhang Q, Zhou D, Wang H, Tan J. Heterotopic ossification of tendon and ligament. *J Cell Mol Med.* 2020 May;24(10):5428-5437. doi: 10.1111/jcmm.15240. Epub 2020 Apr 15. PMID: 32293797.
4. Bleakney RR, Tallon C, Wong JK, Lim KP, Maffulli N. Long-term ultrasonographic features of the Achilles tendon after rupture. *Clin J Sport Med.* 2002 Sep;12(5):273-8. doi: 10.1097/00042752-200209000-00003. PMID: 12394198.
5. Magnusson SP, Agergaard AS, Couppé C, Svensson RB, Warming S, Krogsgaard MR, Kjaer M, Eliasson P. Heterotopic Ossification After an Achilles Tendon Rupture Cannot Be Prevented by Early Functional Rehabilitation: A Cohort Study. *Clin Orthop Relat Res.* 2020 May;478(5):1101-1108. doi: 10.1097/CORR.0000000000001085.

PMID: 31913154; PMCID: PMC7170668.

6. O'Brien EJ, Frank CB, Shrive NG, Hallgrímsson B, Hart DA. Heterotopic mineralization (ossification or calcification) in tendinopathy or following surgical tendon trauma. *Int J Exp Pathol*. 2012 Oct;93(5):319-31. doi: 10.1111/j.1365-2613.2012.00829.x. PMID: 22974213

7. Ateschrang A, Gratzner C, Weise K. Incidence and effect of calcifications after open-augmented Achilles tendon repair. *Arch Orthop Trauma Surg*. 2008 Oct;128(10):1087-92. doi: 10.1007/s00402-007-0441-5. Epub 2007 Sep 15. PMID: 17874248.

8. Cong Q, Liu Y, Zhou T, Zhou Y, Xu R, Cheng C, Chung HS, Yan M, Zhou H, Liao Z, Gao B, Bocobo GA, Covington TA, Song HJ, Su P, Yu PB, Yang Y. A self-amplifying loop of YAP and SHH drives formation and expansion of heterotopic ossification. *Sci Transl Med*. 2021 Jun 23;13(599):eabb2233. doi: 10.1126/scitranslmed.abb2233. PMID: 34162750; PMCID: PMC8638088.

16. In Figure S7D-H, why did the authors focus on chondrogenic and not osteogenic differentiation, given they describe an ossification process? Further, could the authors repeat these experiments with Sema5a+ MSC, given they did show differentiation potential to both relevant lineages?

Response:

Thank you for your careful review and constructive comments. Most cases of injury-induced heterotopic ossification (HO) develop through endochondral ossification [1].

This process involves several stages: chondrogenic differentiation of mesenchymal

stromal cells, formation of a cartilage template, hypertrophy of chondrocytes, vascular invasion, and subsequent bone formation. Targeting the stage of chondrogenic differentiation could suppress the process of endochondral ossification at an early stage. Therefore, we focused on chondrogenic differentiation in this study.

To validate whether TNC promotes chondrogenic differentiation of *Sema5a*⁺ progenitors, we cultured *Sema5a*⁺ progenitors with TNC stimulation or a vehicle control under chondrogenic induction conditions. Alcian blue staining showed that TNC promotes chondrogenic differentiation of *Sema5a*⁺ progenitors (Figure R4A-B). Furthermore, XMU-MP-1 suppressed the TNC-induced chondrogenic differentiation of *Sema5a*⁺ progenitors (Figure R4C-D). These results indicate that TNC promotes chondrogenic differentiation of *Sema5a*⁺ progenitors through Hippo signaling.

Figure R4. A. Alcian blue staining and quantification in Ctrl, Ctrl+CI and TNC+CI group.

CI: chondrogenic induction. n=3. B, Western blot analysis of Sox9 and Gapdh in Ctrl, Ctrl+CI and TNC+CI group. C, Alcian blue staining and quantification in Ctrl, TNC+Veh. and TNC+XMU-MP1 group. n=3. Data shown are mean \pm SEM. D, Western blot analysis of p-YAP, YAP and Gapdh in Ctrl, TNC+Veh. and TNC+XMU-MP1 group.

References:

1. Zhang Q, Zhou D, Wang H, Tan J. Heterotopic ossification of tendon and ligament. *J Cell Mol Med.* 2020 May;24(10):5428-5437. doi: 10.1111/jcmm.15240. Epub 2020 Apr 15. PMID: 32293797; PMCID: PMC7214162.

17. TNC is a highly expressed gene in tendons, and specifically along the muscle tendon junction. Complete knockout of TNC is not a minor mutation and could have widespread effects. The authors should provide description of the mutant mice, specifically tendon and muscle tendon junction development, and describe the tendon regeneration process regardless of the CD26 population in these mutants compared to wild-type mice?

Response:

Thank you for your careful review and constructive comments. Saga et al. found that TNC KO mice develop normally and no abnormalities were found in myotendinous junctions [1]. Consistent with previous reports, we found that TNC KO mice showed no physiological abnormalities [1,2]. There was no significant difference in the weight of TNC KO mice compared with WT mice. However, as previously reported, we observed abnormal behavior in TNC KO mice, which exhibited incessant movement regardless of the dark-light cycle [2].

To further investigate whether targeting TNC would affect the tendon healing process, we administered an anti-TNC antibody to neutralize TNC protein. The results showed that collagen fibril diameter and mechanical properties, including maximum force and stress at failure, were not significantly affected in the anti-TNC group compared to the IgG control group. This indicates that targeting TNC did not significantly affect the process of tendon healing (Figure S8C-G).

Figure S8. C, TEM images of cross section of Achilles tendons from IgG, and anti-TNC group. Scale 500nm. D-E, Bar graph presents the aggregated distribution of collagen fibril diameters using high-magnification TEM images from n = 3 mice per group. F-G. Mechanical properties evaluation

of Achilles tendons in IgG, and anti-TNC group. n = 3. Data shown are mean \pm SEM.

References:

1. Saga Y, Yagi T, Ikawa Y, Sakakura T, Aizawa S. Mice develop normally without tenascin. *Genes Dev.* 1992 Oct;6(10):1821-31. doi: 10.1101/gad.6.10.1821. PMID: 1383086.
2. Mackie EJ, Tucker RP. The tenascin-C knockout revisited. *J Cell Sci.* 1999 Nov;112 (Pt 22):3847-53. doi: 10.1242/jcs.112.22.3847. PMID: 10547346.
3. Järvinen TA, Jozsa L, Kannus P, Järvinen TL, Kvist M, Hurme T, Isola J, Kalimo H, Järvinen M. Mechanical loading regulates tenascin-C expression in the osteotendinous junction. *J Cell Sci.* 1999 Sep;112 Pt 18:3157-66.
4. Xu K, Shao Y, Xia Y, Qian Y, Jiang N, Liu X, Yang L, Wang C. Tenascin-C regulates migration of SOX10 tendon stem cells via integrin- α 9 for promoting patellar tendon remodeling. *Biofactors.* 2021 Sep;47(5):768-777.

18. Does YAP inhibition improve tendon healing or just eliminates HO? Comparison of the regeneration and healing process in the inhibited mice and control wild-type should be carried out.

Response:

Thank you for your careful review and constructive comments. To validate whether inhibition of the Hippo pathway could affect the tendon healing process, mice with tendon injuries received treatment intraperitoneally three times a week with XMU-MP-1 (2mg/kg) or a vehicle control. At day 30 post-tendon injury, tendon tissues were

collected, and transmission electron microscopy (TEM) analysis and mechanical testing were performed to evaluate the tendon healing process. TEM analysis showed that the XMU-MP-1 group contained collagen fibrils with a shifted distribution towards sizes >80 nm in diameter compared to the injured controls (Figure S8H-J). In addition, mechanical properties, including maximum force and stress at failure, were increased in the XMU-MP-1 group (Figure S8K-L). These results indicate that inhibition of the Hippo pathway improves the tendon healing process.

Figure S7. H, TEM images of cross section of Achilles tendons from Vehicle, and XMU-MP-1 group. Scale 500nm. I-J, Bar graph presents the aggregated distribution of collagen fibril diameters using high-magnification TEM images from $n = 3$ mice per group. K-L. Mechanical properties evaluation of Achilles tendons in Vehicle, and XMU-MP-1 group. $n = 3$. Data shown are mean \pm SEM.

19. Why were TNC and Hippo pathway chosen? Clearly, the KEGG pathway analysis shown other pathways which are more significant.

Response:

Thank you for your careful review and constructive comments. The rationale for choosing TNC/Hippo signalling as our research focus is as follows: (1) The Hippo signalling pathway has been widely reported to participate in the regulation of osteogenesis and chondrogenesis, and we also observed enrichment of this pathway in CD26+ TSPCs through KEGG enrichment analysis (Figure 7O) [1-3]. (2) In our previous study, we found that TNC plays an important role in pathological new bone formation in ankylosing spondylitis [4]. (3) Interestingly, we found that TNC was aberrantly upregulated upon tendon injury (Figure 7P). Administration of an anti-TNC antibody significantly suppressed heterotopic ossification (HO) while not significantly affecting the process of tendon healing, as mentioned above in comment 17 (Figure S8C-G). Collectively, these results indicate that TNC is a potential target for heterotopic ossification.

Indeed, the molecular mechanisms underlying chondrogenic differentiation of CD26+ TSPCs are complex, and it is not surprising that multiple signaling pathways are involved, including the PI3K-Akt pathway and MAPK pathway. Nonetheless, these pathways also play essential roles under physiologic conditions, and targeting these pathways might lead to severe side effects.

References:

1. Zhao X, Tang L, Le TP, Nguyen BH, Chen W, Zheng M, Yamaguchi H, Dawson B,

You S, Martinez-Traverso IM, Erhardt S, Wang J, Li M, Martin JF, Lee BH, Komatsu Y, Wang J. Yap and Taz promote osteogenesis and prevent chondrogenesis in neural crest cells in vitro and in vivo. *Sci Signal*. 2022 Oct 25;15(757):eabn9009. doi: 10.1126/scisignal.abn9009. Epub 2022 Oct 25. PMID: 36282910; PMCID: PMC9938793.

2. Karystinou A, Roelofs AJ, Neve A, Cantatore FP, Wackerhage H, De Bari C. Yes-associated protein (YAP) is a negative regulator of chondrogenesis in mesenchymal stem cells. *Arthritis Res Ther*. 2015 May 30;17(1):147. doi: 10.1186/s13075-015-0639-9. PMID: 26025096; PMCID: PMC4449558.

3. Pan JX, Xiong L, Zhao K, Zeng P, Wang B, Tang FL, Sun D, Guo HH, Yang X, Cui S, Xia WF, Mei L, Xiong WC. YAP promotes osteogenesis and suppresses adipogenic differentiation by regulating β -catenin signaling. *Bone Res*. 2018 Jun 1;6:18. doi: 10.1038/s41413-018-0018-7. PMID: 29872550; PMCID: PMC5984632.

4. Li Z, Chen S, Cui H, Li X, Chen D, Hao W, Wang J, Li Z, Zheng Z, Zhang Z, Liu H. Tenascin-C-mediated suppression of extracellular matrix adhesion force promotes enthesal new bone formation through activation of Hippo signalling in ankylosing spondylitis. *Ann Rheum Dis*. 2021 Jul;80(7):891-902. doi: 10.1136/annrheumdis-2021-220002. Epub 2021 Apr 15. PMID: 33858850

20. The authors should address what happens to the CD26+ cells in the TNC KO and following Hippo-pathway inhibition? Is it only their osteogenic differentiation that is effected? Are they proliferating at the same rate? Are they invading the tendon

midsubstance at the same rate?

Response:

Thank you for your careful review and constructive comments. To investigate whether inhibition of the Hippo pathway with XMU-MP-1 could affect the osteogenic differentiation, cell proliferation, and cell invasion properties of CD26⁺ TSPCs, mice received XMU-MP-1 (2mg/kg) or vehicle treatment three times a week after tendon injury. At 7 days post-injury (dpi), injured tendon tissues were collected, and CD26⁺ tendon progenitor cells were isolated using fluorescence-activated cell sorting. The results showed that osteogenic differentiation and cell proliferation slightly increased in the XMU-MP-1 group, while the invasion of CD26⁺ tendon progenitor cells into the tendon midsubstance showed no difference (Figure R5A-E).

To further investigate whether knockout of TNC would affect the osteogenic differentiation capacity, cell proliferation, and cell invasion properties of CD26⁺ TSPCs, injured tendon tissues at 7 dpi were collected, and CD26⁺ TSPCs were isolated using fluorescence-activated cell sorting from wildtype and TNC knockout mice. The results showed that the osteogenic differentiation capacity, cell proliferation, and cell invasion properties of CD26⁺ TSPCs were not affected in TNC knockout mice (Figure R6A-E).

Figure R5. A-B. Alizarin red staining and quantification of CD26⁺ TSPCs under osteogenic induction from vehicle and XMU-MP-1 group. C. Proliferation of CD26⁺ TSPCs from vehicle and XMU-MP-1 group. through CCK8 assay. D-E. Immunofluorescent staining of CD26 in tendon from vehicle and XMU-MP-1 group. Scale bar: 100 μ m. Veh., vehicle; M, midsubstance. n = 3. Data shown are mean \pm SEM.

Figure R6. A-B. Alizarin red staining and quantification of CD26⁺ TSPCs collected from wildtype (WT) and TNC knockout (KO) mice under osteogenic induction. C. Proliferation of CD26⁺ TSPCs collected from WT and TNC KO mice through CCK8 assay. D-E. Immunofluorescent staining of CD26 in tendon from WT and TNC KO mice. Scale bar: 100 μ m. Veh., vehicle; M, midsubstance. n = 3. Data shown are mean \pm SEM.

21. As shown now, with the lack of explanation and insufficient data, the data in Figure 7 (TNC and hippo pathway) seem irrelevant for this manuscript and could fit in a separate manuscript.

Response:

Thank you for your careful review and constructive comments. Based on the

supplementary experimental data currently available, we have identified significant roles for TNC and the Hippo pathway in heterotopic ossification (HO) formation, which are likely related to the chondrogenic differentiation of CD26⁺ TSPCs. Furthermore, we observed that during the process of tendon healing, approximately 90.00% at 30 days post-injury (dpi) and 86.54% at 60 dpi of p-YAP⁺ cells were CD26⁺ cells, indicating that inhibition of the Hippo pathway primarily affects the CD26⁺ cell population at these stages (Figure S7A-B). These findings suggest their potential as therapeutic targets. Therefore, we have included these findings within the scope of this study. Additionally, we have emphasized the limitations in clarifying the molecular mechanisms underlying the contribution of CD26⁺ TSPCs to HO formation. Future investigations will focus on confirming whether the differentiation of CD26⁺ TSPCs is Hippo signaling-dependent, utilizing YAP conditional knockout mice as suggested.

Figure S7. A-B Immunofluorescent staining and quantification of CD26 and p-YAP of Achilles tendon from Sham, 30 dpi and 60dpi group. $n=3$. One way ANOVA with

Turkey's test. Scale bar: 100 μ m. Data shown are mean \pm SEM.

22. Please expand the discussion on the role of the Sema5a⁺ and Apoe⁺ MSC clusters. Are they transient or permanent cell states found in the tissue? What are their biological roles? Do they differentiate into other lineages? Based on what exactly did the authors term Apoe⁺ cells as intermediate progenitors?

Response:

Thank you for your careful review and constructive comments. Since both Sema5a⁺ and Apoe⁺ progenitors could be detected under normal and tendon healing conditions, we initially considered these two populations as permanent cell states. However, we could not rule out the possibility that these two populations might be transient cell states, as no evidence of lineage tracing was provided. GO term analysis showed enrichment of immune system processes, neuron differentiation, blood vessel development, skeletal system development, angiogenesis, and chemotaxis in Apoe⁺ progenitors, indicating that these cells may participate in the above biological processes after tendon injury (Figure S5B). Additionally, GO term analysis showed enrichment of skeletal system development, extracellular matrix organization, ossification, cartilage development, bone development, and osteoblast differentiation in Sema5a⁺ progenitors, indicating that these cells participate in these biological processes after tendon injury (Figure S5C).

Unfortunately, as no flow cytometry antibody is available for the marker genes of Apoe⁺ progenitors, we were unable to isolate this population to investigate its differentiation capacity in vitro. Apoe or Sema5a lineage tracing mice would provide a

useful tool to investigate the differentiation fate of these cells during tendon healing and heterotopic ossification (HO) formation in the future. Based on pseudotime trajectory analysis with the Monocle 3 algorithm, which showed that Apoe⁺ progenitors reside in the middle between CD26⁺ TSPCs and Sema5a⁺ progenitors, we initially considered Apoe⁺ cells as intermediate progenitors. However, as no evidence of lineage tracing was provided, we decided to rename these cells as Apoe⁺ progenitors.

The above information has been added to the revised manuscript [Page 22-23, Line 440-457; Page 30, Line 590-593].

23. The authors mention generating several mice lines for this study, however there is no description of the generation process, or references to other papers detailing the generation and relevant phenotypes of these mutants.

Response:

Thank you for your careful review and constructive comments. To generate CD26-creERT2-hDTR mice, we utilized CRISPR/Cas9 technology to precisely integrate the Dpp4-creERT2-P2A-hDTR-PolyA gene cassette into the Rosa26 locus of mice. The brief procedure is as follows: sgRNA was transcribed in vitro, and a donor vector was constructed. Cas9, the donor vector, and sgRNA were co-injected into fertilized eggs of C57BL/6JGpt mice. F0 generation mice were obtained and validated for correct integration via PCR and sequencing. Positive F0 mice were then bred with C57BL/6JGpt mice to generate a stable F1 generation of mice carrying the genetic modification.

For the generation of TNC KO mice, exon 2-4 of the Tnc-201 transcript was selected as the knockout region. In this project, we used CRISPR/Cas9 technology to modify the Tnc gene. The brief process is as follows: gRNA was transcribed in vitro. Cas9 and gRNA were microinjected into the fertilized eggs of C57BL/6JGpt mice. Fertilized eggs were transplanted to obtain positive F0 mice, which were confirmed by PCR and sequencing. A stable F1 generation mouse model was obtained by mating positive F0 generation mice with C57BL/6JGpt mice.

The details of the above information have been added to the revised manuscript [Page 30-31, Line 607-614; Page 31, Line 618-626].

24. In Figure S6I, the comparison between PBS and DTX groups is not reliable due to the wide spread of the control group compared to the narrow spread of the DTX group. The authors would need to increase the n in these experiments to make them reliable. Also, this experiment is not detailed in the methods at all, the authors need to detail when and from where was the blood collected, along with other experimental procedures.

Response:

Thank you for your careful review and constructive comments. We have added another four samples to each group to validate whether inducible ablation of CD26⁺ cells at the injury sites would affect the white blood cell profiles in peripheral blood. For inducible ablation of CD26⁺ cells at the injury sites, each mouse was locally injected with 50 ng

of diphtheria toxin (DTx) daily for 5 days. On the third day after the last DTx injection, peripheral blood from both the PBS and DTx groups was collected. White blood cell profiles were detected using an automated hematology analyzer (BC-5000 Vet). The results showed that local administration of diphtheria toxin near the Achilles tendon did not affect the circulating white blood cell profile in CD26-hDTR mice (Figure S6H).

In addition, detailed information on the detection of white blood cell profiles has been added to the revised manuscript [Page 39-40, Line 794-799].

Figure S6H. Peripheral blood routine examination of PBS and DTx group in CD26-hDTR mice.

Veh., vehicle, DTx, Diphtheria toxin. Scale bar: 100 μ m. Data shown are mean \pm SEM.

25. To get a sense of the healing properties of the tendon, images of a non-injured control next to the images and quantifications in Fig. 5 should be added.

Response:

Thank you for your careful review and constructive comments. Sham control images and quantifications of the tendon have been added to Figure 5.

Figure 5B Picrosirius red staining of Achilles tendons from Sham, Vehicle (Veh.) and diphtheria toxin (DTx) group. Scale bar: 100 μ m.

Figure 5E-F, Histological analysis and quantification of tenogenic genes including Col1a1, Scx and Fmod in sham or injured tendon treated with diphtheria toxin or

vehicle control. n=6. Scale bar: 100µm.

Minor comments

1. Fig.1 A-B show 14 clusters, line 108 in the text describes 13 clusters.

Response:

Thank you for your careful review and constructive comments. We have re-analysed the scRNA-seq dataset and corrected the description to 9 clusters in Figure 1A. (Page 7, Line 112-115)

2. From line 113 to line 120, the authors detail highly expressed marker genes for three MSC subpopulations in the scRNAseq dataset. They reference Figure S2A-B, which relates to the tail tendon structure and a tSNE map of its scRNAseq. Could it be possible the authors meant to refer to Figure S1B-D instead? (as these figures show expression of the genes discussed in the text)

Response:

Thank you for your careful review and constructive comments. We apologize for any confusion caused by the mislabeling of the figure panels. We have corrected the arrangement and description of the figure panels.

3. In Figure 3A, the authors use the Ly6a gene name in the FACS plot. Ly6a encodes for the Sca1 protein, which would be more appropriate to use in this context.

Response:

Thank you for your careful review and constructive comments. We have revised the description and now use Sca1 to demonstrate the FACS plot.

Figure 3A, Fluorescence activated cell sorting (FACS) of Sca1⁺CD26⁺ MSCs and Sca1⁺Sema5a⁺ progenitors.

4. In Figure 3F, the framing of the tendon regions do not seem consistent (comparing spread of cell nuclei). Additionally, adding labels for the tissue regions would greatly help understanding the images.

Response:

Thank you for your careful review and constructive comments. The difference in the spread of cell nuclei may have been due to inconsistent transection of the tissue, and we have replaced it with a consistent, representative immunofluorescent staining image. To better illustrate the framing of the tendon region, we have added the label "tendon midsubstance (M)" in Figure 3H.

Figure 3. H, Immunofluorescent images of tdTomato and Sema5a. Scale bar: 100 μ m.

5. The slingshot algorithm is missing entirely from methods section.

Response:

Thank you for your careful review and constructive comments. To avoid confusions caused by too many pseudotime analysis results, we decided to utilize CytoTRACE and Monocle algorithm to perform these analyses as suggested. Therefore, the part of pseudotime analysis slingshot analysis was removed in the revised manuscript.

6. In line 238, the authors refer to Figure S5A-C when discussing gene expression over pseudotime, however this figure deals with GO analysis. Could it be possible the authors meant to refer to Figure S6A-C, which seems more fitting?

Response:

Thank you for your careful review and constructive comments. We apologize for any confusion caused by the mislabeling of the figure panels. We have now corrected the arrangement and description of the figure panels.

7. Lines 249-250 are incoherent and their meaning unclear.

Response:

Thank you for your careful review and constructive comments. We apologize for any confusion caused by the misleading phrases. We have revised the rationale for this description. Since some hematopoietic lineage cells express CD26 and inflammatory responses play a role in the tendon healing process [1], we first investigated the existence of CD45+CD26+ immune cells under normal conditions. The results showed that few CD45+CD26+ immune cells were found in uninjured tendons (Figure S6A and S6B) [Page 15, Line 282-286].

Reference:

1. Nourissat G, Berenbaum F, Duprez D. Tendon injury: from biology to tendon repair. *Nat Rev Rheumatol.* 2015 Apr;11(4):223-33. doi: 10.1038/nrrheum.2015.26. Epub 2015 Mar 3. PMID: 25734975.

8. The authors did not specify in the differentiation assay what was the cell concentration used (cells per well), nor what was the medium used before switching to the differentiation mediums.

Response:

Thank you for your careful review and constructive comments. The cell concentration used for osteogenic and adipogenic differentiation was 0.9×10^5 cells/well, and for chondrogenic differentiation, it was 1×10^5 cells/well. Before switching to the differentiation mediums, DMEM medium supplemented with 10% fetal bovine serum

(FBS) was used.

Detailed information about the differentiation assay has been added to the revised manuscript. [Page 37, Line 737-752]

9. The authors did not specify what medium they were using for the proliferation assay, or in what dish they cultured the cells (what was the cell concentration).

Response:

Thank you for your careful review and constructive comments. For proliferation detection, different sorted cells were plated at a concentration of 1000 cells per 10 cm dish and cultured in DMEM medium supplemented with 10% FBS for 9 days.

Detailed information about the proliferation assay has been added to the revised manuscript. [Page 38, Line 753-756]

10. In the scRNAseq section in the methods, did the authors control for mitochondrial genes, nCount or nFeatur data to eliminate doublets or low-quality cells as commonly used pipelines instructs? If not, what was their QC strategy for these issues?

Response:

Thank you for your careful review and constructive comments. We filtered out cells with less than 300 genes per cell, more than 10% mitochondrial read content, and nCount over 15000. Detailed methods for quality control of scRNAseq analysis have been added to the method section [Page 42, Line 837-838].

11. In the tendon punch experiment, the authors did not specify what package or algorithm were used to integrate the two datasets for analysis.

Response:

Thank you for your careful review and constructive comments. We used the merge, NormalizeData, and ScaleData functions in the Seurat package to integrate the two datasets for analysis.

Detailed methods for quality control of scRNAseq analysis have been added to the method section. [Page 42, Line 840-841]

12. The authors do not detail how they carried out the KEGG pathway analysis.

Response:

Thank you for your careful review and constructive comments. A differential gene expression analysis was initially conducted on the CD26+ TSPCs of the uninjured and injured groups to identify a set of genes that are highly expressed in the injured group ($p < 0.05$). Subsequently, all peak-related genes were mapped to the KEGG (Kyoto Encyclopedia of Genes and Genomes) database, and a KEGG enrichment analysis was performed on this gene set. The calculated p-values were subjected to FDR correction, with an $FDR \leq 0.05$ used as the threshold. [Page 44, Line 881-887]

13. The authors refer a couple of times to Figure 8 however this MS has only 7 figures.

Response:

Thank you for your careful review and constructive comments. We apologize for the

misleading labeling of the figure panels. We have corrected the arrangement and description of the figure panels.

14. Lower magnifications of immunostained regions should be shown in order to better understand what regions were these images taken from (in a like manner to that shown in Fig. 2L).

Response:

Thank you for your careful review and constructive comments. Lower magnification images of histological staining have been added to guide the audience to the specific location within the tendon where the images in Figure 2C, Figure 6D, and Figure 6F were taken.

Figure 2C. SOFG staining and immunofluorescent staining of CD26 and Lama4 of of Achilles tendons. n=3. Scale bar: 100 μ m.

Figure 6D. SOFG staining and immunofluorescent staining of Sox9 and tdTomato (tdT) of Achilles tendon from Sham, 30 dpi and 60dpi group. n=5. Scale bar: 100 μ m.

Figure 6F. SOFG staining and immunofluorescent staining of Sp7 and tdTomato (tdT) of Achilles

tendon from Sham, 30 dpi and 60dpi group. n=5. Scale bar: 100 μ m.

15. It is not clear why immune CD26⁺ cells are not ablated in the DTR experiment.

Please explain.

Response:

Thank you for your careful review and constructive comments. We observed a low number of CD26⁺CD45⁺ immune cells in uninjured tendons, while a significant increase in CD26⁺CD45⁺ immune cells was detected in injured tendons, suggesting that these cells may migrate from the peripheral blood after tendon injury (Figure S6A and S6B). Furthermore, both immunofluorescent and flow cytometry analyses demonstrated that the infiltration of CD45⁺ immune cells was not altered after diphtheria toxin injection (Figure S6C-D and S6E-G). To specifically deplete CD26⁺ TSPCs in the peritendinous region without affecting circulating CD26⁺ immune cells, diphtheria toxin was locally injected around the Achilles tendon for 5 days. Peripheral blood was collected three days after the final diphtheria toxin injection, and a white blood cell profile analysis was performed. The results showed that local depletion of CD26⁺ cells did not affect the circulating white blood cell profile (Figure S6H). Furthermore, our scRNA-seq data revealed that only 5.19% of monocyte-macrophages and few neutrophils and lymphocytes expressed CD26 (Figure R1), which may explain why diphtheria-toxin-induced ablation of CD26⁺ cells did not alter the populations of neutrophils and monocyte-macrophages (Figure R1). Overall, these results indicate that local depletion of CD26⁺ cells did not affect the infiltration of

immune cells after tendon injury.

Figure S6. A-B, Immunofluorescent analysis of CD45 and CD26 of uninjured and injured Achilles tendons. $n=3$. Two-tailed unpaired Student's t test. C-D, Immunofluorescent analysis of CD45 and CD26 of Veh. and DTx group. $n=3$. Two-tailed unpaired Student's t test. E-G, Flow cytometry analysis of CD45⁺CD11b⁺ myeloid cells in Veh. and DTx group. $n=3$. Two-tailed unpaired Student's t test. H, Peripheral blood routine examination of PBS and DTx group in CD26-hDTR mice. $n=7$.

Two-tailed unpaired Student's t test. Veh., vehicle, DTx, Diphtheria toxin. Scale bar: 100µm. Data shown are mean ± SEM.

Figure R1 A. UMAP plot and keys to the clustering numbers and cell types are shown to the right. B. UMAP plot of expression of CD26. C. Bubble plot of expression of CD26.

16. In the figures describing the Cre construct, instead of PolyA it is written as PlolyA.

Response:

Thank you for your careful review and constructive comments. We apologize for the incorrect spelling of "PolyA" and this mistake has been corrected.

Reviewer #4 (Remarks to the Author):

I co-reviewed this manuscript with one of the reviewers who provided the listed reports.

This is part of the Nature Communications initiative to facilitate training in peer review

and to provide appropriate recognition for Early Career Researchers who co-review manuscripts.

Response:

Thank you for your careful review and contribution.

REVIEWER COMMENTS

Reviewer #1 (Remarks to the Author):

Thank you for the opportunity to review the manuscript entitled “A CD26⁺ tendon stem cell population contributes to tendon repair and heterotopic ossification.” The authors have adequately addressed all comments raised in the first round of reviews with additional experiments and explanations. Including additional images, single-cell RNA-sequencing analyses, and quantifications have strengthened their claims. A minor addition should be included in the manuscript to solidify the single-cell RNA-sequencing results. Please address the comment below before publication:

1. The CytoTrace analysis supports your findings by showing high CytoTrace scores in the “Cd26+” group. It would be even more convincing to show a feature plot (expression overlaid on dimensional reduction) of the Cd26 expression on the CytoTrace UMAP, to correlate the regions of the UMAP with high CytoTrace scores to potentially high expression of Cd26.

Response:

Thank you for your careful review and constructive comments. We have added the feature plot of CD26 expression on the CytoTRACE UMAP as suggested. The result showed that CD26 expression correlate the regions of UMAP with high CytoTRACE scores (Figure 1E and S1E).

Figure 1. E, CytoTRACE analysis of CD26⁺ MSCs, Apoe⁺ progenitors, Sema5a⁺ progenitors.

Figure S1. E, UMAP plot of CD26 in CD26⁺ MSCs, Apoe⁺ progenitors, Sema5a⁺ progenitors.

Reviewer #2 (Remarks to the Author):

The authors have addressed all comments.

Response:

Thank you for your careful review and constructive comments.

Reviewer #3 (Remarks to the Author):

The authors have nicely carried out the majority of the concerns however a few have remained that need corections.

Previous main comment 20: The authors claim that the CD26⁺ cells have proliferated and invaded the tendon midsubstance to the same extent in the TNC KO. While proliferation is shown in vitro, the fact that a similar number of cells is observed in the tissue does not conclude that invation was not affected. In order to claim this, lineage analysis must be done. It should either be carried out or simply to moidfy the claim.

Response:

Thank you for your careful review and constructive comments. Indeed, immunofluorescent staining showed similar proportion of CD26⁺ cells within the tendon midsubstance, does not necessarily conclude that invasion was not affected. We would modify this claim as “The results implies that the osteogenic differentiation capacity, cell proliferation and cell invading property of CD26⁺ TSPCs might not affected in TNC KO mice. However, lineage analysis of CD26 in TNC-KO mice would help to further confirm the invasion property of CD26⁺ cells.”

Previous minor comment 9: The authors claim they plated 1000 cells per 10 cm tissue culture plate. This does not seem realistic. Could it be a typo?

Response:

Thank you for your careful review and constructive comments. We are sorry for the misunderstanding of the description of the CFU assay and CCK8 assay.

For CFU assay, according previous study, about 100-200 cells are recommended to seed in a well of 6-well plate and the cell density is about 11-20 cells /cm² [1]. In the current study, we seeded 1000 cells in a dish with 10 cm diameter, therefore the cell density was about 16 cells/cm², which was consistent with previous study.

For CCK8 assay, different sorted cells were seeded at the concentration of 1200 cells per well in a 96-well plate and cultured in DMEM medium supplemented with 10% FBS. The CCK8 assay was performed according to instruction manual.

Detailed information of the CFU assay and CCK8 assay has been added in the revised manuscript [Page 38, Line 754-759].

References:

1. Franken NA, Rodermond HM, Stap J, Haveman J, van Bree C. Clonogenic assay of cells in vitro. Nat Protoc. 2006;1(5):2315-9. doi: 10.1038/nprot.2006.339. PMID: 17406473.

Previous minor comment 11: the methods used by the authors to integrate datasets seem inadequate to the commonly used pipelines. Seurat offers specific integration tools for this purpose, and not just by merging datasets.

Rseponse:

Thank you for your careful review and constructive comments. We are sorry for the inadequate description of the integration of datasets. We did use the Seurat to perform the integration of datasets. In detail, we first merged datasets using Merge() function and then normalize data using NormalizeData() and then identified the top 2,000 highly variable genes by FindVariableFeatures(). Next, we scaled the seuratobject using ScaleData() and RunPCA() with informative features. We used package Harmony to correct the batch effects.

Detailed information has been added in the revised manuscript [Page 42, Line 844-848].

Reviewer #4 (Remarks to the Author):

Response:

Thank you for your careful review and constructive comments.